# Genome-wide mapping of cancer dependency genes and genetic modifiers of chemotherapy in high-risk hepatoblastoma

Jie Fang[1,18], Shivendra Singh[1,18], Changde Cheng[2,18], Sivaraman Natarajan [2,18], Heather Sheppard [3,4,18], Ahmed Abu-Zaid[1], Adam D. Durbin[5], Ha Won Lee [6], Qiong Wu [1], Jacob Steele [7], Jon P. Connelly [7], Hongjian Jin[8], Wenan Chen [8], Yiping Fan[8], Shondra M. Pruett-Miller [7], Jerold E. Rehg[3], Selene C. Koo [4], Teresa Santiago[4], Joseph Emmons[9], Stefano Cairo[10], Ruoning Wang [11], Evan S. Glazer[12], Andrew J. Murphy [1,12], Taosheng Chen [6], Andrew M. Davidoff [1,12,13,14], Carolina Armengol [15,16,17], John Easton [2], Xiang Chen [2,13] ✉ & Jun Yang [1,13,14] ✉

A lack of relevant genetic models and cell lines hampers our understanding of hepatoblastoma pathogenesis and the development of new therapies for this neoplasm. Here, we report an improved MYC-driven hepatoblastoma-like murine model that recapitulates the pathological features of embryonal type of hepatoblastoma, with transcriptomics resembling the high-risk gene signatures of the human disease. Single-cell RNA-sequencing and spatial transcriptomics identify distinct subpopulations of hepatoblastoma cells. After deriving cell lines from the mouse model, we map cancer dependency genes using CRISPR-Cas9 screening and identify druggable targets shared with human hepatoblastoma (e.g., CDK7, CDK9, PRMT1, PRMT5). Our screen also reveals oncogenes and tumor suppressor genes in hepatoblastoma that engage multiple, druggable cancer signaling pathways. Chemotherapy is critical for human hepatoblastoma treatment. A genetic mapping of doxorubicin response by CRISPR-Cas9 screening identifies modifiers whose loss-of-function synergizes with (e.g., PRKDC) or antagonizes (e.g., apoptosis genes) the effect of chemotherapy. The combination of PRKDC inhibition and doxorubicin-based chemotherapy greatly enhances therapeutic efficacy. These studies provide a set of resources including disease models suitable for identifying and validating potential therapeutic targets in human high-risk hepatoblastoma.

Hepatoblastoma and hepatocellular carcinoma (HCC) are the most common primary liver malignancies in children and adolescents/young adults. While primary liver cancers account for only 1–2% of all pediatric tumors[1], the largest incidence increase has been observed for hepatoblastoma in children under 5 years in nearly all regions of the world[2]. The rate is increasing at more than 4.3% annually in the US[3]. Hepatoblastoma is an embryonal neoplasm that likely arises from hepatic cell precursors[4,5]. Genetically, hepatoblastoma has the fewest somatic mutations among all human cancers[6], suggesting that hepatic precursor cells during the early stage of liver development may be

particularly susceptible to fundamental events resulting in oncogenic transformation. In line with previous findings as reviewed[7,8], genomic sequencing studies have confirmed that mutations in the Wnt-β-catenin signaling pathway are the most common genetic event in hepatoblastoma[9–17]. The gene for the antioxidant transcription factor, *NFE2L2*, is also altered in a subpopulation of high-risk hepatoblastomas[9,10,15,17], suggesting that liver cells undergo oxidative stress during cellular transformation or disease progression.

The Hippo signaling pathway plays a critical role in liver organogenesis and cancer[18–20]. The dysregulated downstream effector molecule of Hippo signaling, YAP1, is involved in hepatoblastoma tumorigenesis[21–25]. Combination of the activated form of YAP1 (YAP$^{S127A}$) with either hepatoblastoma relevant NFE2L2 mutant or CTNNB1 mutant promotes liver tumorigenesis although either alone is unable to transform normal liver cells in these mouse models[26]. Hepatic developmental pathways may determine the differentiation capacity of mutated liver progenitor/stem cells, and differentiation status may determine the aggressiveness of hepatoblastoma[9]. Hepatoblastomas with high expression levels of stem/progenitor cell markers (*EpCAM, LIN28B, SALL4, HMGA2, AFP*) are usually associated with poor prognosis[9]. Such liver stem/progenitor cells have the ability to accumulate mutations following chemotherapy, leading to the development of post-treatment residual disease resulting in relapse and metastasis[17].

The *MYC* oncogenes are involved in many cancers including hepatoblastoma[11,27–31]. Gain of chromosome 2 and 8 (with *MYCN* and *MYC* oncogenes, respectively) is common (25–50%) in hepatoblastoma[7,11,32,33]. While β-catenin mutation alone (*CTNNB1*) is usually insufficient to transform liver progenitor cells into hepatoblastoma[26], MYC cooperates with β-catenin and YAP to sustain tumorigenesis[29] and is an essential requirement for tumor maintenance in a β-catenin-based hepatoblastoma mouse model[28]. β-catenin drives MYC expression[30] and MYC silencing prevents tumor growth in human hepatoblastoma cancer cell line-based xenograft models[11]. These data indicate that MYC plays an essential role in hepatoblastoma growth.

Due to a lack of targetable somatic mutations and a paucity of genetic animal disease models and cell lines[34,35], identification of therapeutic targets in hepatoblastoma remains challenging. Conventional chemotherapy is critical for most hepatoblastoma treatment. However, the genetic response of hepatoblastoma cells to chemotherapy is not well defined, which impedes development of more effective therapies because of an incomplete understanding of the mechanism of therapeutic response and resistance.

Here we generate a hepatocyte-specific MYC-driven multifocal hepatoblastoma-like tumor model that resembles high-risk human hepatoblastoma. The transcriptomics of this transgenic hepatoblastoma-like model are characterized by bulk RNA-seq, single cell RNA-seq, and pathology-based spatial transcriptomics, all of which confirm its similarity with human hepatoblastoma. Cell lines generated from this model are readily passaged in vitro. Cancer dependency genes are mapped by a genome-wide CRISPR-Cas9 screening approach. We also perform genetic mapping of cellular responses to doxorubicin, a commonly used chemotherapeutic for hepatoblastoma treatment, with a genome-wide CRISPR-Cas9 screen and identify genes that synergize with and antagonize the effect of chemotherapy. Based upon this screen, a combination therapy is developed which shows better efficacy than doxorubicin treatment alone. Our studies characterize hepatoblastoma disease models (mouse and cell lines) that recapitulate pediatric hepatoblastoma and identify potential therapeutic targets of hepatoblastoma that are conserved across mouse and human species.

## Results

### Hepatocyte-specific MYC overexpression drives rapid hepatic oncogenesis

Previous genetic hepatoblastoma mouse models have provided invaluable information toward our understanding of the role of oncogenic drivers in this cancer. However, most of these models have only addressed well differentiated hepatoblastoma, which has a relatively good clinical outcome, or they do not align with the onset of liver development in children. To overcome these limitations, we generated a model in a C57BL/6J genetic background by crossing hepatocyte-specific transgenic Alb-Cre mice (Cre recombinase under the control of the mouse albumin enhancer/promoter hybrid)[36] with CAG-STOP$^{flox/flox}$-Myc mice (CAG promoter-driven human c-MYC, whose expression is prevented by a LoxP site flanked STOP cassette)[37] (Fig. 1a). Hepatocyte-specific, Cre-mediated excision of the floxed STOP cassette allows expression of the CAG promoter-driven human *Myc*, leading to a typical phenotype with hepatomegaly and para-neoplastic alopecia in double transgenic Alb-Cre;CAG-Myc mice (ABC-Myc, Figs. 1b and S1a). Strikingly, activation of one allele of the *Myc* oncogene led to rapid onset of liver tumors in neonatal mice; all these mice died within 1–10 weeks after birth (Fig. 1c). which is consistent with the known role for MYC in sustaining hepatoblastoma growth. Embryonic lethality was not induced by *Myc* activation as all possible genotypes were recovered at the expected Mendelian ratio (Table S1). Western blot and immunohistochemistry validated MYC overexpression in non-neoplastic hepatocytes and liver tumor tissues at fetal (E17.5) and different postnatal stages (Figs. 1d, S1b), suggesting that MYC is activated in the fetal stage. In parallel, we also developed an ABC-Myc;TdTomato model that had a similar tumor penetrance and lethality but that also expressed TdTomato as a lineage reporter (Fig. 1c). Together the data demonstrate that the introduction of human MYC alone is sufficient to quickly drive tumorigenesis in susceptible murine hepatic stem/progenitor cells in the fetal livers of transgenic mice.

### Pathological analyses define the ABC-Myc-driven liver neoplasm as a hepatoblastoma-like malignancy

Hepatoblastoma is histologically heterogenous, with two main histologic patterns (epithelial, and epithelial mixed with mesenchymal components). Epithelial patterns are further delineated into fetal, embryonal, mixed fetal and embryonal, cholangioblastic, small cell undifferentiated, macrotrabecular, mixed and others[38,39]. Tumors arising in the ABC-Myc model effaced most of the sampled liver tissues and had a highly resembling human hepatoblastoma histology with embryonal or combinations of both fetal and embryonal morphologies comprising the bulky tumors, as well as scattered foci of extra-medullary hematopoiesis (Fig. 1e, Table 1). These multifocal tumors involving all liver lobes correspond to human PRE-Treatment EXTent of tumor (PRETEXT) stage IV disease[40]. Clinically, 35% of patients with hepatoblastoma present as multifocal tumors at diagnosis, and 43% of these are PRETEXT stage IV[41], a poor prognostic factor that usually requires high-intensity, dose-dense cisplatin and doxorubicin-based chemotherapy, and often liver transplantation[42]. Samples at time points E14.5, E17.5, and postnatal day 7 (P7) were evaluated to assess the presence of pre-neoplastic lesions. Neoplastic transformation was first observed in E17.5 livers in low numbers of scattered developing hepatocytes with abnormal nuclear morphologies (Fig. S1c). Nuclear changes consisted of karyomegaly, marginalization of chromatin, and a single, centralized, and prominent nucleolus that is consistent with other cancers where constitutive MYC activation is present. These dysplastic cells were interpreted as pre-neoplastic lesions based on the biological time course of the ABC-Myc mouse model described in this paper.

Neoplastic nodules were grossly visible in all liver sections of ABC-Myc mice starting at P7. Multifocal to coalescing neoplastic foci with an embryonal morphology could be observed in the livers of P7 ABC-Myc mice (Fig. 1e), consistent with the hypothesis that hepatoblastoma-like neoplasia may arise from epithelial-lineage committed hepatic stem progenitor cells with the introduction of human oncogenic Myc signaling resulting in impaired differentiation. Further evaluation of the

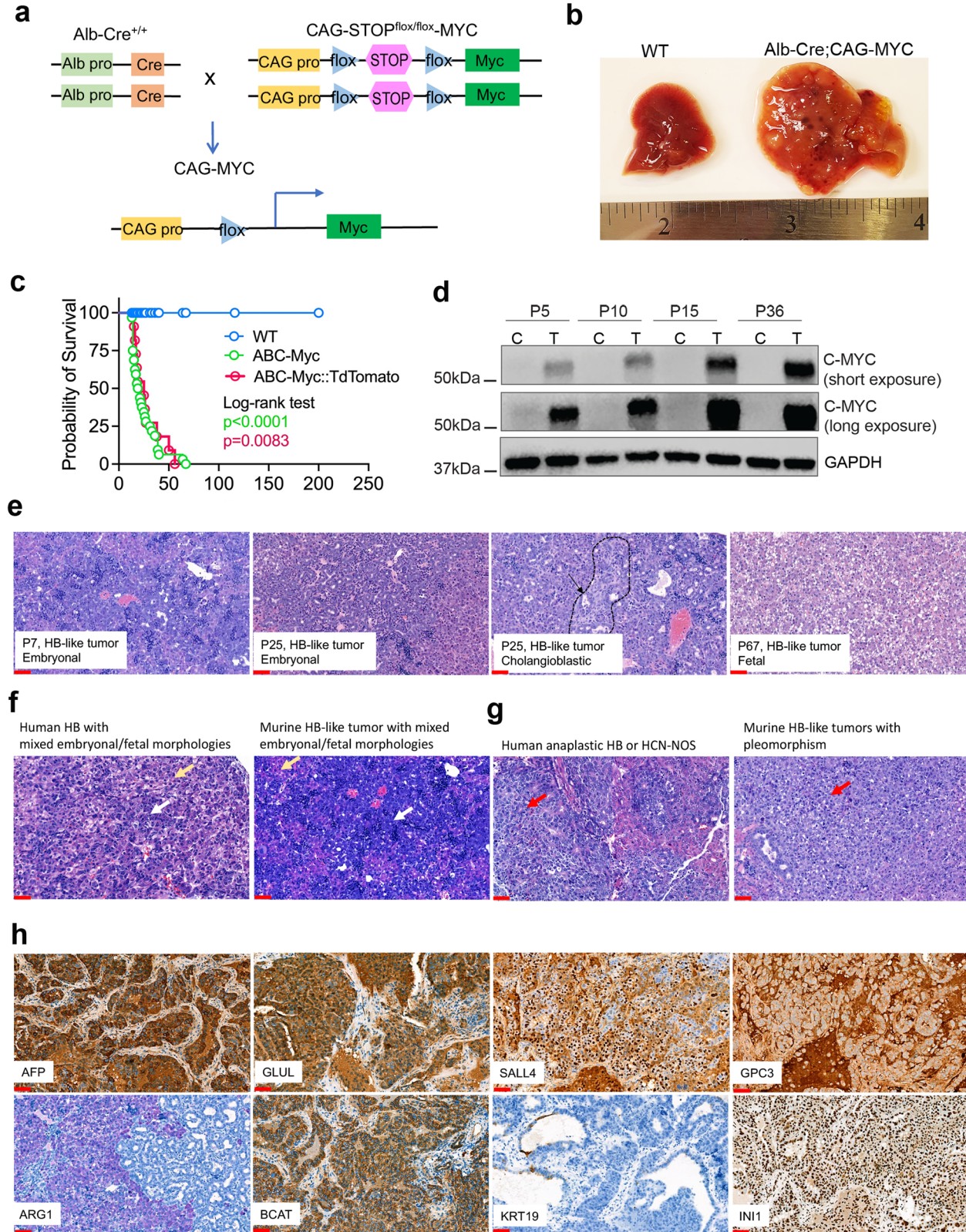

hepatoblastoma-like tumors from time points P25 to P67 showed a coexistence of distinct subpopulations of neoplastic cells with embryonal, fetal, and rarer cholangioblastic-like morphologies (Fig. 1e). The co-existence of these morphologies in advanced hepatoblastoma-like tumors is most consistent with human pediatric hepatoblastoma with a mixed epithelial phenotype. Small cell undifferentiated, rhabdoid, teratoid, and mesenchymal morphologies were not observed. There were no definitive well-differentiated fetal morphologies identified in the sections except within P67 tumors (Fig. 1e). All tumors had combinations of primitive morphologies comparable to the previously described C2 morphologic phenotype as described by Cairo et al.[11]. Hepatoblastomas characterized as C2 are documented to have aggressive biological behavior and an unfavorable prognosis, which is observed in this model.

**Fig. 1 | ABC-Myc drives hepatoblastoma-like tumor development. a** Breeding strategy to generate Alb-Cre;CAG-Myc (ABC-Myc) compound mice. **b** Hepatomegaly with tumor nodules in ABC-Myc liver in comparison with age matched normal mouse liver. **c** Inferior overall survival of ABC-Myc (green color, $n = 33$), and ABC-Myc/TdTomato (pink, $n = 11$) mice, respectively. Log-rank (Mantel-Cox) test used for statistical analysis in Kaplan-Meier survival. **d** Western blot showing overexpression of C-MYC in ABC-Myc livers at postnatal day 5 (P5), 10 (P10),15 (P15) and 36 (P36) in comparison with the normal controls. The blots are representative of three independent experiments. **e** Hematoxylin and Eosin (H&E) shows histology of ABC-Myc tumors at postnatal day 7 (P7), 25 (P25), 67 (P67).

Sample number for each image $n = 1$. Scale bar = 25 μm. **f** Hematoxylin and Eosin (H&E) shows mixed histology of human and ABC-Myc tumors. Sample number for each image $n = 1$. Scale bar = 25 μm. **g** Hematoxylin and Eosin (H&E) shows pleomorphism of human and ABC-Myc tumors. Sample number for each image $n = 1$. Scale bar = 25 μm. **h** Immunostaining of alpha fetoprotein (AFP), glutamine synthetase (GLUL), Spalt Like Transcription Factor 4 (SALL4), Glypican 3 (GPC3), arginase (ARG1), β-catenin (BCAT), cytokeratin 19 (KRT19) and integrase interactor 1 (INI1). Sample number for each image n = 1. Scale bar = 25 μm. Source data are provided as a Source Data file.

The Myc-driven murine hepatoblastoma-like tumors demonstrate phenotypic plasticity of hepatocyte lineage committed stem/progenitor cells. While the co-existence of embryonic and fetal histological features of ABC-Myc tumors resemble the human hepatoblastoma (Fig. 1f), it is important to differentiate hepatoblastoma from hepatocellular carcinoma in pediatric patients, because of differing treatment and prognosis[38]. While the poorly differentiated histology is consistent with the pediatric C2 phenotype, some tumor areas also contain histologic features of the subclassification of pediatric hepatoblastomas with hepatocellular carcinoma features that were previously called transitional liver cell tumors (TLCT)[9] (Fig. 1g), indicating that some ABC-Myc tumor cells have features of HCN-NOS (Hepatocellular Malignant Neoplasm, Not Otherwise Specified) that frequently presents phenotypic plasticity.

We further determined the pathological features of this hepatoblastoma-like malignancy using immunohistochemical markers of human pediatric hepatoblastoma, and observed overexpression of hepatic stem/progenitor cell markers documented in C1 and C2 human pediatric hepatoblastomas[11] (Fig. 1h). Murine hepatoblastoma-like neoplasms had diffuse immunopositivity for alpha fetoprotein (AFP) and glypican 3 (GPC3), two stem cell markers used to distinguish neoplastic hepatocellular cells[39,43,44], as well as immunoreactivity for glutamine synthetase (GLUL or named as GS), a β-catenin target and a marker of β-catenin activated hepatocytes[38,39], SALL4, another embryonal type of hepatoblastoma marker[45,46], and Arginase-1 (ARG-1), a marker used to distinguish primary hepatocellular tumors from metastatic tumors[47]. Immunoreactivity was visually observed in greater than 75% of the bulky hepatoblastoma-like neoplasms and staining intensity for all markers was visually graded as moderate to strong in staining intensity for all markers (Table 1). Rare subpopulations of poorly differentiated neoplastic cells, visually quantified at less than 1% of the neoplasm, had immunoreactivity for cytokeratin 19 (KRT19), a marker for biliary cancer or small-cell undifferentiated type hepatoblastoma[38], as well as non-neoplastic, entrapped bile ducts. INI1 (SMARCB1) was retained in all neoplasms, further demonstrating the hepatocellular origin of these cells. The strong cytoplasmic staining of β-catenin may suggest an activation of the Wnt/β-catenin signaling pathway in these tumors (Fig. 1h). Ki67 staining showed that 3–6% of cells were positive (Fig. S1d, e). In summary, the ABC-Myc hepatoblastoma-like model overall recapitulates the embryonal or mixed fetal and embryonal histologic features of human hepatoblastoma, with some bearing HCN-NOS features, and has anatomic and molecular characteristics of human disease highly associated with the high-risk C2 subtype[11].

C1 and C2 hepatoblastoma subclasses were initially defined by gene expression profiling and can be delineated by epithelial cell type, proliferation differences, and expression of stem cell markers that may be assayed by IHC. C1 and C2 features have been further correlated with the phase of hepatic differentiation in which susceptible lineage-committed subpopulations may undergo tumorigenesis. C2 tumors have molecular features of non-neoplastic murine liver at E11.5 and E12.5, while C1 tumors have features of hepatic differentiation in late and postnatal stages[11]. Some retention of both C1 and C2 characteristics by our murine hepatoblastoma-like neoplasms may result from

differences in transgene copy number expression in the embryonal liver; this difference may affect the timing of malignant transformation in susceptible hepatocyte specified stem-progenitor populations starting at E9.5, when albumin expression can first be detected[48]. Neoplastic transformation was first observable in E17.5 livers in small subpopulations of atypical appearing cells by histology, suggesting that tumorigenesis is occurring along a continuum of time in this model that is based on the increasing expression of albumin into adulthood. Therefore, hepatoblastoma-like neoplasms with hybrid features of C2 and C1 may be expected in the model. Thus, the ABC-Myc hepatoblastoma-like model recapitulates the morphologic features of human hepatoblastoma, histologically most similar to the high-risk C2 class of hepatoblastoma, with some bearing HCN-NOS features, but also retains some immunohistochemical and molecular characteristics of low-risk C1 neoplasms. These subtypes may occur sequentially or randomly. Nevertheless, we only observed well differentiated hepatoblastoma in P67 while embryonal and cholangioblastic subtypes occur at an early time (P7, P25) (Fig. 1e), suggesting that there could be a sequential event during MYC-mediated cellular transformation that coopts with liver developmental program. C2 aggressive type may be derived from stem/progenitor cells in hepatoblast, and thus appeared at an early developmental stage, while the C1 type may be derived from a more differentiated cells at late developmental stage.

## Serum chemistry panel analysis reveals liver dysfunction of ABC-Myc mice similar to that of human hepatoblastoma

To assess the liver function of ABC-Myc mice, we performed serum chemical analysis (Fig. 2a, b). Not surprisingly, ABC-Myc mice showed abnormal elevation of AFP (Fig. 2a), alkaline phosphatase (ALP), alanine transaminase (ALT), and total bilirubin (Fig. 2b), the three commonly used biomarkers of liver function, indicating that the livers in ABC-Myc mice are damaged. One clinical study showed that 80% of hepatoblastoma patients had abnormal levels of ALP and 12.5% had increased ALT[49]. As liver is the major organ that produces glucose, liver cancer can cause hypoglycemia. Indeed, the serum glucose levels in ABC-Myc mice were remarkably reduced (Fig. 2b). The serum levels of creatinine and blood urea nitrogen (BUN) in ABC-Myc mice were also declined in comparison with the age-matched normal mice although not statistically different. While creatinine and BUN are the commonly used chemical markers to assess kidney function, liver cancer can lead to less production of creatinine, a break-down product of creatine in liver through transamination of amino acids. Low levels of BUN may indicate liver disease in the clinic due to less production of urea. However, the albumin and globulin levels seemed to be in the normal range, and no abnormal levels of common electrolytes (Sodium, Potassium, Calcium and Phosphorous) were observed (Fig. 2b). We further performed complete blood count (CBC) measurements to assess if ABC-Myc mice had developed additional complications. While white blood cell counts showed no difference between normal mice and ABC-Myc mice, the absolute number of circulating eosinophils tended to be increased although the difference was not statistically significant (Fig. S2). However, the ABC-Myc mice developed microcytic anemia, as indicated by a reduction in the proportion of red blood cells

**Table 1 | Pathological characterization of ABC-Myc tumors**

| Accession Number | Fetal | Embryonal | Crowded | Macrotrabecular | Mesenchymal | Cholangioblastic | INI1 | SALL4 | GPC3 | AFP | GLUL | ARG | Bcat | Grouping |
|---|---|---|---|---|---|---|---|---|---|---|---|---|---|---|
| RS19-2057 | + | + | - | - | - | +/15-20% | +/100%; retained | +/90% | +/100% | +/100% | +/100% | + | +/100%; membraneous | C2 |
| RS22-578 | - | + | - | - | - | +/<5% | +/100%; retained | + | + | + | +/patchy & strong | + | +/100%; membraneous | C2 |
| RS22-575 | + | + | - | + | - | +/<5% | +/100%; retained | +/15-20% | + | + | +/87% | + | +/100%; membraneous | C2 |
| AP21-532 | + | + | + | + | - | +/<5% | +/100%; retained | + | + | + | + | + | +/100%; membraneous | C2 |
| RS22-576 | - | + | - | - | - | - | +/100%; retained | + | + | + | + | + | +/100%; membraneous | C2 |
| RS22-577 | + | + | - | - | - | +/<5% | +/100%; retained | + | + | + | + | + | +/100%; membraneous | C2 |

(hematocrit, HCT%), amount of hemoglobin (HB), mean corpuscular volume (MCV) and mean corpuscular hemoglobin (MCH), increase in size variation (percentage of red cell distribution width, RDW%), but normal range of total number of red blood cells and mean corpuscular hemoglobin concentration (MCHC) (Fig. 2c). Thrombocytosis also occurred in ABC-Myc mice, as indicated by an increase in total platelet counts, plateletcrit (PCT), and mean platelet volume (MPV) (Fig. 2d). One study reported that among hepatoblastoma patients, 75% had thrombocytosis and 37.5% had microcytic anemia, whereas only 23.1% of pediatric patients with hepatocellular carcinoma had thrombocytosis and none had microcytic anemia[50]. These chemistry and CBC parameters are consistent with the presence of hepatoblastoma-like disease in ABC-Myc mice.

### Signaling pathways in ABC-Myc tumor cells resemble those in human hepatoblastoma with a poor outcome

To understand the molecular mechanisms of ABC-Myc hepatoblastoma-like tumors, we identified the differentially expressed genes in tumors versus age-matched normal murine livers using bulk RNA-seq (Fig. 3a and Supplementary Data 1), followed by signaling pathway analysis. Interestingly, the *Igf2* oncogene ranked first (log2 fold change = 11.6) among the upregulated genes in tumors (Supplementary Data 1). In humans, *IGF2* is located in the 11p15.5 imprinted locus, which is the second most frequently altered locus in hepatoblastomas and hepatocellular carcinomas, mostly through copy-neutral loss of heterozygosity[17]. *IGF2* induction by 11p15.5 alterations is likely the first genetic event in hepatoblastoma[17]. The most significantly downregulated genes in tumors were cytochrome P450 (CYP) family genes related to metabolic functions of mature hepatocytes (Fig. 3a). Gene set enrichment analysis (GSEA)[51] showed that the genes upregulated and downregulated in ABC-Myc tumors were significantly associated with the corresponding human hepatoblastoma gene signatures reported by Cario[11] (Fig. 3b). Since Cario gene sets consisted of hepatoblastoma tissue RNA samples including those resected after preoperative chemotherapy, we compared ABC-Myc gene expression with the gene datasets generated from biopsy or surgery prior to any chemotherapy (Ikeda dataset, GSE131329)[52], which included 14 noncancerous liver tissues and 53 tumor tissues. We used the top 200 differentially expressed genes from Ikeda genset for GSEA analysis, and again, we obtained very similar results (Fig. S3a), which further strengthened our conclusion. In agreement with the immunostaining findings, GSEA demonstrated that the β-catenin pathway was also significantly upregulated in ABC-Myc tumors as indicated by its association with gene signatures derived from β-catenin transgenic liver tumors[29] and β-catenin knockdown in HepG2 cells[53] (Fig. 3c). To further determine if ABC-Myc induces transcriptomes similar to those in human hepatoblastoma, we performed a comparative analysis using the VENN diagram showing the number of deregulated genes (and their %) in the comparison between tumor vs. non-tumor liver samples from hepatoblastoma patients and from the ABC-Myc model at FDR < 0.05 (Fig. 3d). Briefly, we integrated the RNA-seq from ABC-Myc tumors and control livers with the RNA-seq from Carrillo-Reixach's study that included tumor and non-tumor samples from 32 patients with hepatoblastoma[15]. As a result, we obtained a matrix of 11,393 ortholog genes. Then, we performed a supervised analysis by comparing tumor vs. non-tumor samples using human and mouse samples. The results showed that 50.1% and 42.5% of the up- and downregulated genes in the ABC-Myc tumors vs. control liver samples were also deregulated in human hepatoblastoma in comparison with nontumor samples, respectively (Fig. 3d). The statistical analysis clearly showed a significant overlapping in upregulated ($p = 1.6 \times 10^{-96}$) and downregulated ($p = 2.1 \times 10^{-153}$) genes and clearly supports the high similarity of our ABC-Myc tumor model with human hepatoblastoma on the transcriptomic level. To further confirm the high similarity of molecular features between human and mouse tumor samples, we

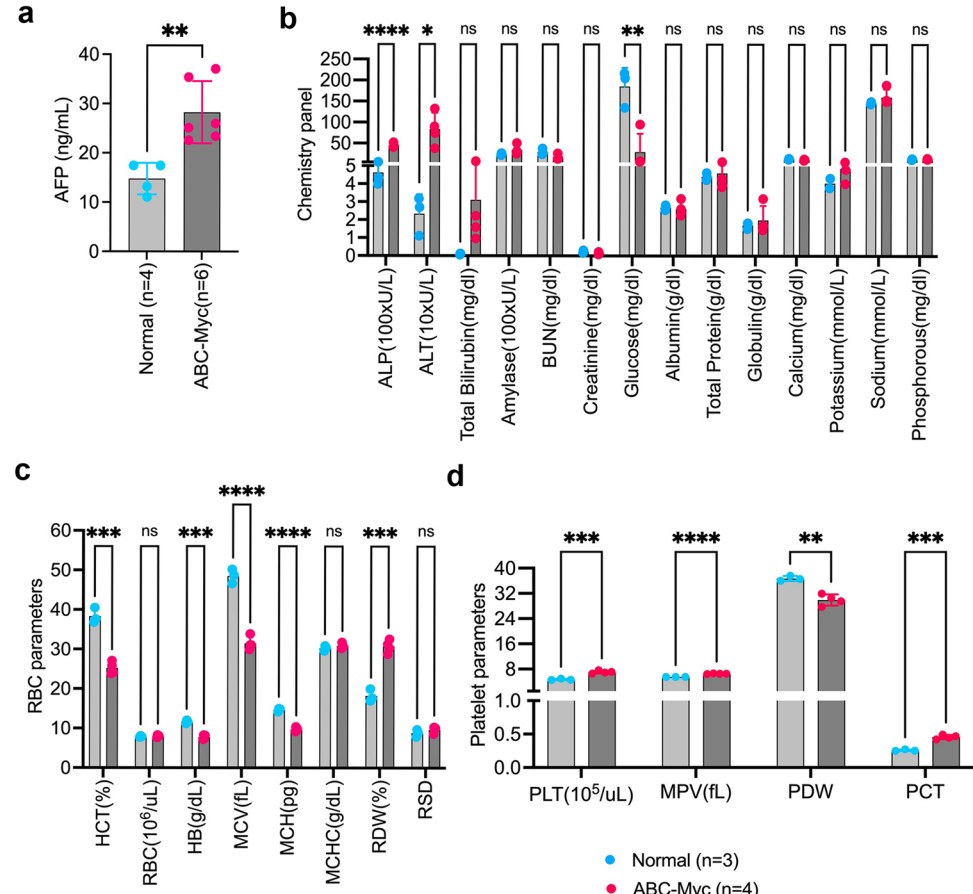

**Fig. 2 | Clinical chemistry analysis of serum from ABC-Myc mice. a** Quantification of serum AFP levels in normal (*n* = 4 biologically independent animals) and ABC-Myc (*n* = 6 biologically independent animals) mice by ELISA. Data are presented as mean ± SD. Unpaired two-sided t-test,**\*\****p* = 0.0046. **b** Chemistry panel markers in determination of liver function, kidney function and electrolytes in serum from normal (*n* = 3 biologically independent animals) and ABC-Myc (*n* = 4 biologically independent animals) mice. Data are presented as mean ± SD. Unpaired two-sided t-test,\*\*\*\**p* < 0.0001, \*\**p* = 0.0056, \**p* = 0.0171, ns not significant. ALP Alkaline phosphatase, ALT Alanine transaminase, BUN Blood urea nitrogen. **c** Complete blood count to determine the changes in red blood cells in blood from normal (*n* = 3 biologically independent animals) and ABC-Myc (*n* = 4 biologically

independent animals) mice. Data are presented as mean ± SD. Unpaired two-sided t-test,\*\*\*\**p* < 0.0001, \*\*\**p* < 0.001, ns not significant. HCT hematocrit, RBC red blood cell, HB hemoglobin, MCV mean corpuscular volume, MCH mean corpuscular hemoglobin, MCHC mean corpuscular hemoglobin concentration, RDW red cell distribution width, RSD red cell standard deviation. **d** Complete blood count to determine the changes in platelets in blood from normal (*n* = 3 biologically independent animals) and ABC-Myc (*n* = 4 biologically independent animals) mice. Data are presented as mean ± SD. Unpaired two-sided t-test,\*\*\*\**p* < 0.0001, \*\*\**p* < 0.001, \*\**p* = 0.0017. PLT platelet, MPV mean platelet volume, PDW platelet distribution width, PCT Plateletcrit. Source data are provided as a Source Data file.

used the integrative human and mouse ortholog genes to perform a Principal Component Analysis of RNA-seq. The results showed that tumor and non-tumor samples were clearly grouped into two categories independent of the species from which samples were obtained (Fig. 3e). Specifically, mouse tumor samples were grouped with human tumor samples and control mouse liver samples were grouped with adjacent non-tumor samples from patients with hepatoblastoma. Additionally, we cross-referenced our RNA-seq results for the top 500 genes upregulated and downregulated in ABC-Myc tumors with human hepatoblastoma RNA-seq analysis reported by Hooks et al.[54] (Fig. S3b). The results again revealed that the top differentially expressed genes in ABC-Myc tumors were similarly altered in human hepatoblastomas (Fig. S3b), further supporting that the murine hepatoblastoma-like model resembles human disease at the transcriptomic level. Further comparison of the gene pathways between ABC-Myc tumors and human hepatoblastomas revealed that both shared altered metabolic pathways and those regulating the cell cycle, DNA replication and repair, and RNA splicing (Table S2). Altogether, our data clearly indicate the high similarity of our ABC-Myc model and human hepatoblastoma and support its use as an experimental model for this extremely rare disease.

The outcomes of hepatoblastoma can be distinguished by two molecular signatures, C1 and C2, which represent better and worse outcomes, respectively[11]. We cross-referenced C1 and C2 signatures to our RNA-seq data and found that ABC-Myc tumors expressed higher levels of C2 and lower levels of C1 signatures (Fig. S3c). To further validate that ABC-Myc tumors resemble the C2 class, we applied seven different prediction algorithms[11], and all of which showed that ABC-Myc hepatoblastomas were classified as C2 (Table S3). Hirsch et al. reported that hepatoblastoma can be further classified into 4 molecular subtypes, 'Hepatic differentiation', 'Liver progenitor', 'Mesenchymal' and 'Proliferation'[17]. ABC-Myc tumors exhibited low expression of 'Hepatic differentiation' signature, but high expression of the 'Liver progenitor' and 'Proliferation' signatures (Fig. S3d). We also found three out of six 'Mesenchymal' markers were expressed in ABC-Myc tumors (Fig. S3d), albeit to a lesser degree (Supplementary Data 1). While the 'Hepatic differentiation' group overlaps with C1, the 'Liver progenitor' signature is associated with a subclass of hepatoblastoma that has the worst outcome[15,17]. Consistent with the primarily embryonal histological features, GSEA results showed that ABC-Myc tumors had significant upregulation of cancer stem cell signatures including "liver cancer with upregulated EpCAM" and "undifferentiated cancer"

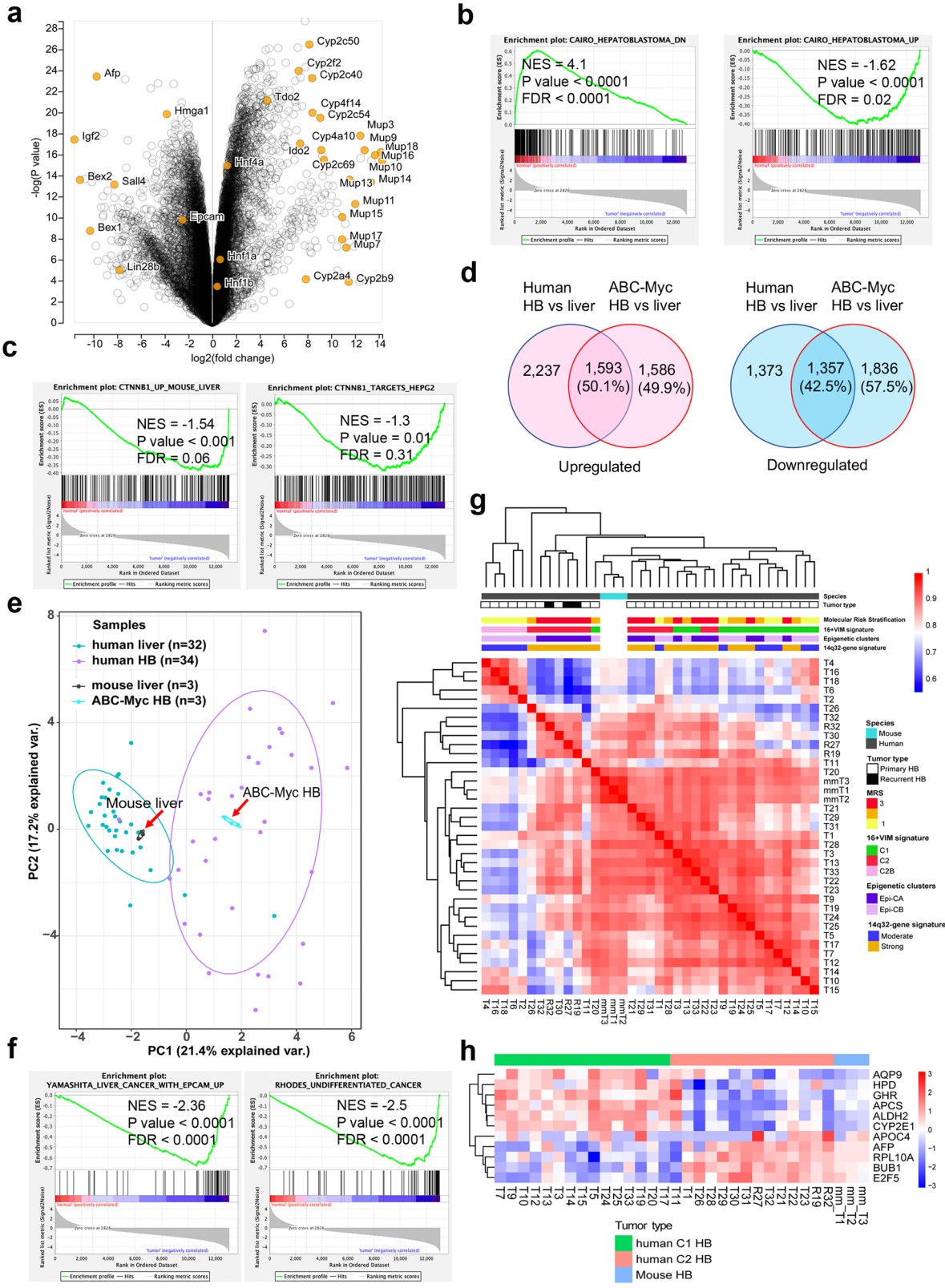

(Fig. 3f). Compared with age-matched normal livers, the hepatoblastoma embryonic gene markers (i.e., *Lin28b, Sall4, EpCAM, Hmga, Afp*) in ABC-Myc tumors, which are usually associated with a poor outcome[9], were increased over 4–250 fold (Fig. 3a, Supplementary Data 1). Notably, *Lin28b* is an oncogene that can drive hepatoblastoma in a transgenic mouse model[55], and is highly expressed in high-risk hepatoblastoma[31]. Clustering analysis of the correlation between the

gene expression of human hepatoblastoma in Carrillo-Reixach's study and the mouse samples showed four main groups of tumor samples (Fig. 3g). In line with the results in Fig. 3e, the gene expression profile of ABC-Myc tumor samples was highly correlated with that of the human primary hepatoblastoma and specifically, with tumors in the proliferative "C2- Pure subclass" ($p = 0.019$) and with a strong "14q32-gene signature" overexpression ($p = 0.027$). These molecular features have

**Fig. 3 | Signaling pathways in ABC-Myc tumor cells resemble those in human hepatoblastoma with poor outcome. a** Volcano plot showing differentially expressed genes in ABC-Myc tumors ($n = 3$) vs normal mouse livers ($n = 3$). X-axis represents the expression changes in log2 (fold). Y axis represents the significance of expression change for each gene in -log10 ($p$ value). **b** GSEA showing genes highly downregulated and upregulated in ABC-Myc hepatoblastoma are significantly associated with the signatures downregulated (left panel) and upregulated (right panel) in human hepatoblastoma reported by Cario et al.[11]. $P$ Value calculated by one-sided Fisher's exact test. The FDR is calculated by comparing the distribution of normalized enrichment scores from many different genesets. **c** GSEA showing genes highly upregulated in ABC-Myc hepatoblastoma are significantly associated with the β-catenin signatures derived from mouse livers overexpressing β-catenin in dataset (GSE79084)[29] (left panel), and β-catenin knockdown in HepG2 cells from dataset (GSE94858)[53] (right panel). **d** Proportional VENN diagrams of the up-regulated (top) and down-regulated genes (bottom) in the human HB ($n = 34$) vs. adjacent non-tumor liver (NL, $n = 32$) samples (left) and mice Myc-ABC tumor ($n = 3$) vs. control liver (CL, $n = 3$) samples. The numbers in the Venn diagrams represent the number of significant genes at FDR < 0.05. The

comparisons were performed considering the total of 11,393 ortholog genes. RNA-seq database from patients with HB was obtained from Carrillo-Reixach et al. (GSE133039)[15]. The $P$ values (upregulated $p = 1.6 \times 10^{-96}$, downregulated $p = 2.1 \times 10^{-153}$) of the overlaps are calculated by the hypergeometric distribution. **e** Principal Component Analysis using the integrated dataset consisting in 11393 genes present in mouse and human tumor (HB, GSE133039)[15], non-tumor liver (NL) and control liver (CL) samples. **f** GSEA showing stem cell gene signatures highly upregulated in ABC-Myc hepatoblastoma. $P$ Value calculated by one-sided Fisher's exact test. The FDR is calculated by comparing the distribution of normalized enrichment scores from many different genesets. **g** Pearson correlation heatmap using the dendrogram of bootstrapping hierarchical clustering from tumoral samples including the 11,393 ortholog genes present in mouse and human tumor samples. Human hepatoblastomas were annotated with molecular features obtained from Carrillo-Reixach et al. (GSE133039)[15]. **h** Heatmap of the 11 ortholog genes of gene the 16-gene signature in C1 and C2 human hepatoblastomas (GSE133039)[15] and mouse ABC-Myc tumors. Source data are provided as a Source Data file.

been already reported to be associated with clinical features of poor prognosis[11,15]. In line with these findings, *Dlk1* was highly expressed in ABC-Myc tumors (log2 fold change = 8.85) (Supplementary Data 1). *DLK1* is a well-known hepatoblast marker and is highly expressed in hepatoblastoma[56]. Carrillo-Reixach et al. recently identified the *DLK1-DIO3* locus genes on 14q32 as a new hallmark of human hepatoblastoma that is associated with Wnt/β-catenin signaling, and high expression of 14q32 gene signature being associated with a poor outcome[15], supporting that hepatoblasts could be the cells of origin of ABC-Myc hepatoblastoma. Finally, integrative analysis of the expression profile of the 11 ortholog genes of the 16-gene signature further confirmed that the mouse ABC-Myc tumors had a similar profile to the human C2 hepatoblastomas in the Carrillo-Reixach cohort[11] (Fig. 3h). Taken together, these data demonstrate that ABC-Myc tumors resemble human hepatoblastoma with molecular signatures of aggressive disease.

## scRNA-seq analysis of ABC-Myc tumors reveals the heterogeneity of hepatoblastoma-like cells

Single cell RNA sequencing (scRNA-seq) studies have shown that the mammalian liver is composed of multiple cell lineages in addition to hepatocytes and cholangiocytes[57–59]. The heterogeneity of liver cells is further complicated by the anatomical structure of liver zonation[60,61], which shows a distinct expression pattern of metabolic genes distributed from the central vein to the portal vein along the lobule axis[62]. While scRNA-seq analysis has provided insight into adult hepatocellular carcinoma and its tumor microenvironment[63,64], tumor heterogeneity in hepatoblastoma at the single cell level has just been recently appreciated[65,66]. To investigate if the transcriptomic ecosystem of ABC-Myc-driven tumors recapitulates human hepatoblastomas, we performed scRNA-seq to define the distinct cellular populations of cells dissected from 4 tumors and 3 healthy livers. Cell Ranger Single-Cell Software Suite (version 6, 10X Genomics) was used to quality control and quantify the single-cell expression data to generate filtered gene-barcode matrices for 90,715 cells with an average of 3037 mRNA molecules (UMIs, median = 1963, range: 302–32,765). First, we characterized the transcriptional differences between tumor samples and the control group by applying the NBID algorithm[67] that we developed for differential analysis of scRNA-seq. The top 10 genes upregulated in tumor samples, ordered by the fold change (adjusted $P < 6.428\text{e-}323$, log2FC > 5), were *Camp*, *Ngp*, *Igf2*, *Ltf*, *Prtn3*, *Afp*, *Ermap*, *Rhd*, *Elane*, and *Mpo* (Fig. S4a). The high levels of *Igf2* and *Afp* further verified the hepatoblastoma-like tumors arising from ABC-Myc mice. Highly elevated expression of granule genes such as *Camp*, *Ngp*, and *Ltf* are correlated with neutrophil development. In addition, genes *Prtn3*, *Elane*, and *Mpo* are functional activation markers of neutrophils

involved in inflammation, infection, and tumor invasion. Activated tumor-associated neutrophils release enzymes, mainly proteinase 3 (encoded by *Prtn3*), neutrophil elastase (encoded by *Elane*), and myeloperoxidase (coded by *Mpo*), destroying surrounding tissues, which may lead to tumor invasion. Numerous studies have shown that neutrophils in the tumor microenvironment can promote rapid tumor development and growth[68]. We also noted that erythroid genes *Ermap* and *Rhd* are among the top 10 genes, and erythroid-like signature is present in human pediatric hepatoblastomas[65]. Pathway analysis with Hallmark genes showed that DNA repair, cell cycle, MYC targets, E2F targets, and heme metabolism were enriched in tumor samples (adjusted P < 4.5e-08, Fig. S4b).

After correction for batch effect, 16 clusters of cells in normal and tumor tissues were generated by unsupervised clustering of the global single-cell transcriptomic datasets with Latent Cellular State Analysis (Fig. 4a, b). The frequencies of low-quality cells (cells with low UMI counts ($\leq 500$) or more than 20% UMI of mitochondrial genes) in clusters 5 were greater than 50%, which was therefore removed, and the rest of the 15 clusters were kept for further analysis. Among the 15 clusters that remained, six (clusters 2, 3, 7, 9, 12, and 16) were dominated by cells from tumor tissues, with more than 97.5% of cells in each cluster from a tumor sample (Fig. 4a, b). However, the cells in four clusters (clusters 4, 13, 14 and 15) were predominantly from the control group, with more than 93.6% of cells in each cluster from a normal liver sample. The remaining five clusters (clusters 1, 6, 8, 10, and 11) were largely shared by both tumor and control groups (with 23.5–60.6% of cells in each cluster from the tumor group). To define the biological functions of each cluster, we used Seurat (version 4.3.0) to compare the average single-cell expression profiles from each cluster with previously annotated reference datasets[69]. After determining the similarity of each cluster to previously defined cell types[70], we generated top 10 markers for each cell cluster (Supplementary Data 2), and then performed gene set enrichment analysis to associate the marker genes of each group to known functional pathways in KEGG (Supplementary Data 2). The clusters shared between the tumor and control groups (clusters 1, 6, 8, 10, and 11) were all enriched with immunity related genes. Genes involved in DNA damage, heme biosynthetic process, and oxidative phosphorylation were enriched in tumor-specific clusters (clusters 2, 3, 7, 9, 12, and 16) (Supplementary Data 2). Clusters specific to the control group (clusters 13 and 15) were enriched for genes involved in the endoplasmic reticulum, essential for hepatocytes' protein synthesis function.

scRNA-seq analysis of human hepatoblastoma by Song et al. reported 6 tumor clusters (Tumor clusters 1–3, Tumor cluster 4 Erythroid, Tumor cluster 5 DCN high and Tumor cluster Neuroendocrine) and 6 hepatoblastoma-associated clusters (HB associated Erythroid,

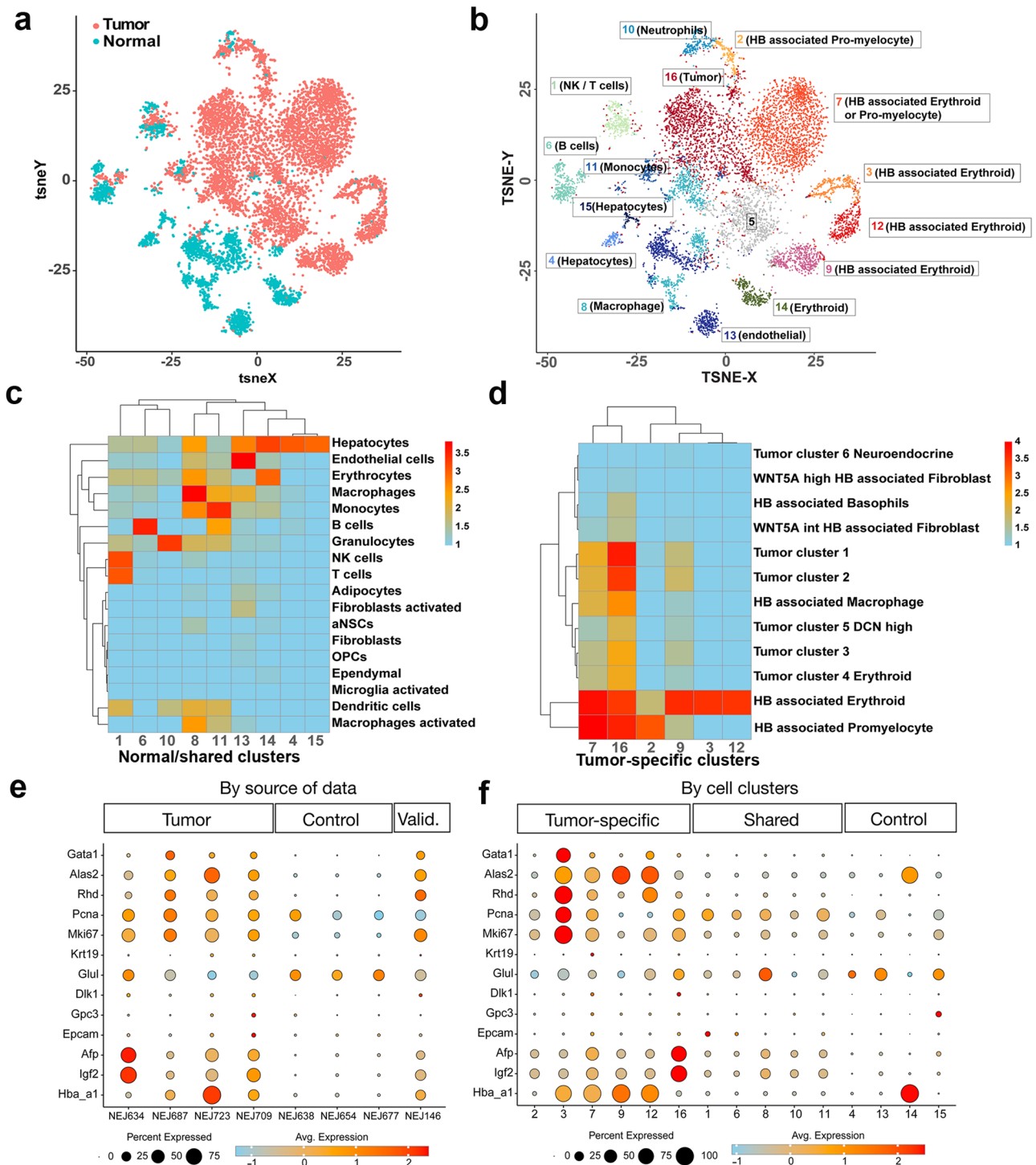

**Fig. 4 | scRNA-seq analysis of ABC-Myc tumors reveals the heterogeneity of hepatoblastoma-like cells. a** A t-SNE plot showing the source of cells partitioned by normal livers ($n = 3$) and ABC-Myc tumor ($n = 4$) samples. **b** A t-SNE plot showing the inferred cell clusters using Latent Cellular State Analysis for the normal livers ($n = 3$) and ABC-Myc tumor ($n = 4$) samples. Cluster 5 mainly consisted of low-quality cells with low ($\leq 500$) UMI counts or more than 20% UMI counts from mitochondrial genes. Therefore cluster 5 was not pursued further in our analysis. The cells in each cluster were colored and labeled with numbered annotations assigned by the Bioconductor package SingleR using normal mouse cell-type marker genes and orthologs of human hepatoblastoma tumor signature genes (Song et al.,)[65] as references. **c** Confusion matrices of cell clusters, shared between normal liver and tumor samples or specific to normal liver samples, aligned against

a reference expression profile consisting of normal mouse cell types from celldex. The color indicates the log10 transformed counts of labeled cells using SingleR. **d** Confusion matrices of tumor-specific cell clusters aligned against reference human HB tumor data from Song et al.[65]. The color indicates the log10 transformed counts of labeled cells. **e** Expression pattern of selected up-regulated genes between tumor and control groups. Each dot represents the expression profile for a gene in a sample. The size of a dot indicates the percent of expressed cells in a sample, and the darkness of the blue color indicates the strength of average expression. Data was grouped by tumor samples, control samples, and a tumor sample NEJ146 as the validation set. **f** Bubble plot of the expression pattern of selected up-regulated genes between tumor and control groups, with data grouped by inferred cell clusters. Source data are provided as a Source Data file.

HB associated Promyelocyte, HB associated Macrophage, HB associated Basophils, WNT5A high HB associated Fibroblast and WNT5A intermediate HB associated Fibroblast[65]. To map gene signatures in these 12 clusters generated from human hepatoblastoma, we used SingleR to compare the single-cell expression profiles with previously annotated reference cell types[70]. This method quantifies the similarity to reference cells based on the highest Spearman rank correlations, using a set of marker genes to focus on the relevant differences between cell types. We used annotated normal mouse cell-type marker genes from R package celldex[70] and orthologs of human hepatoblastoma signature genes[65] for normal cells and tumor cells, respectively. The clusters shared between the tumor and control groups (clusters 1, 6, 8, 10, and 11) were annotated as NK/T cells, B cells, macrophages, neutrophils, and monocytes, respectively (Fig. 4b, c, Supplementary Data 2). Cell clusters specific to the normal control group (clusters 13, 14, 15) were matched with endothelial, erythrocyte and hepatocyte, respectively (Fig. 4b, c). In comparison with the cell type-specific signatures identified in hepatoblastoma related clusters[65], we found that cluster 2 in ABC-Myc tumors is most likely the 'HB associated Promyelocyte', while clusters 3, 9 and 12 are most likely the 'HB associated Erythroid' (Fig. 4b, d), although these three clusters have different enrichment of biological functions (Supplementary Data 2). Cluster 16 is dominated by cells resembling the human hepatoblastoma clusters[65], and enriched with expression signatures of 'HB associated Promyelocyte and Erythroid' clusters (Fig. 4b, d). Like cluster 16, cluster 7 also bears gene signatures of human 'Tumor clusters' and 'HB associated Promyelocyte and Erythroid clusters' (Fig. 4d). The specific clusters from normal livers have low fraction of UMIs mapping to hemoglobin genes (except for Cluster 14, a small cluster of erythroid cells) (Supplementary Table 4). On the contrary, tumor specific clusters have overall elevated fractions of UMIs mapping to hemoglobin genes, especially for the clusters of HB-associated Erythroid that showed 33–83% of UMI from hemoglobin genes per cell. These results indicate that the high expression levels of erythroid genes are unlikely due to contamination.

To further characterize the transcriptomics of tumor-like Clusters 7 and 16, we tested the transcriptional differences between each of the clusters against the rest of the tumor-specific clusters (cluster 7 vs. clusters 2, 3, 9, 12, and 16; and cluster 16 vs. 2, 3, 7, 9, 12). For Cluster 7, genes *S100a4, H2-Eb1, Cstdc5, Ighm, H2-Aa, Cd74, Stfa3, F13a1 and Dcn* (logFC > 2.43, adjusted *P* < 2.1e-238, Supplementary Data 3) are among top upregulated genes. Upregulated genes also include interesting genes like *Dlk1, Epcam, Gpc3,* and *Krt19* (logFC 0.38, 1.13, 0.94, 2.00; adjusted *P* < 4.04e-6). KEGG analysis showed that cluster 7 is enriched with immunity- or inflammation-related pathways such as the chemokine signaling pathway, Cytokine-cytokine receptor interaction, B cell receptor signaling pathway, Th1 and Th2 cell differentiation and NF-kappa B signaling pathway (Fig. S4c). For Cluster 16, *Hamp2* and *Hamp* genes encoding liver produced hormone peptides that regulate iron absorption and distribution across tissues, liver injury biomarker *Cps1*, and bile salt export pump gene *Abcb11* are among the top upregulated genes (logFC > 3.92, adjusted *P* < 3.06e-322, Supplementary Data 4). Genes *Afp* (logFC = 2.41, adjusted *P* < 3.06e-322), *Igf2* (logFC = 2.7, adjusted *P* < 3.06e-322), and *Dlk1* (logFC = 0.82, adjusted *P* = 6.15e-27) are also among upregulated genes. KEGG analysis revealed that Cluster 16 is enriched with pathways involved in amino acid and lipid metabolism, citrate cycle, and Hippo signaling pathway gene sets (Fig. S4d).

Differential gene expression analysis of scRNA-seq validated that *Igf2* and *Afp* are two of the top 10 genes highly expressed in ABC-Myc tumors in comparison with normal controls, which are highly expressed in clusters 7 and 16 (Figs. 4e, f, S4e), supporting that clusters 7 and 16 represent heterogenous hepatoblastoma-like cells. Then, we specifically examined the expression of other hepatoblastoma cell markers and erythroid lineage markers (Fig. 4e, f). We found that *Epcam,*

*Gpc3* and *Dlk1* were highly expressed in a small percentage of cells (< 25%) in tumor samples compared to normal control livers although not as remarkably as the erythroid genes (*Gata1, Alas2, Rhd, Hba-a1*) (Fig. 4e). This was validated by performing scRNA-seq on one additional tumor sample, NEJ146, which was not performed together with the aforementioned samples. The cellular proliferation markers *Pcna* and *Mki67* were also highly expressed in ABC-Myc tumors (Fig. 4e). Together with *Afp* and *Igf2,* the expression of stem cell markers *Dlk1, Gpc3* and *Epcam* appeared to be higher in clusters 7 and 16 (Fig. 4f), which may represent the bona fide cancer cells. Nevertheless, we noticed the differential expression of these markers in each individual tumor, demonstrating both intra- and inter- tumor heterogeneity of ABC-Myc tumors.

To demonstrate the heterogeneity of tumor cells, we have highlighted the tumor cell specific clusters (Cluster 2, 3, 7, 9, 12, 16) in each tumor sample (Fig. S5a). The results showed that each tumor consisted of these clusters with different percentages (Fig. S5b), which demonstrated both intra-tumoral and inter-tumoral heterogeneity. For example, NEJ723 and NEJ634 were dominated by Clusters 7 and 16, respectively; while NEJ709 and NEJ687 showed multiple tumor clusters co-existed at substantial fractions. To further characterize the tumor cell heterogeneity, we only focused on the clusters 7 and 16, which expressed highest levels of *Afp* and *Igf2* and thus these clusters presumably represent bona fide tumor cells. We were able to partition these strong *Afp+Igf2+* clusters 7 and 16 into several subclusters (Fig. S5c). For each of them, we found significant variation in subcluster proportion across tumor cells (*P* < 0.0005). In addition, we determined the composition of different types of tumor cells in each tumor sample by mapping ABC-Myc tumor cells with the annotated human hepatoblastoma cluster genes[65]. Again, the murine tumors showed intra-tumoral heterogeneity (different tumor classes in individual tumors) and inter-tumoral heterogeneity (different composition of various tumor cell types) (Fig, S5d). There is a significant variation of proportion of tumor cell types among the four samples (Chi square test: *P* = 0.0005). We then used the 16-gene signature that differentiates C1 and C2 types to interrogate the subtype heterogeneity from our scRNA-seq (Fig. S6). Basically, in tumor samples, the expression levels of C2 signature were greatly higher than the C1 gene signature. However, in normal liver samples, the expression of C2 signature was negligible and the C1 signature was dominantly high (Fig. S6a). Next, we examined the expression of C1 and C2 in each cluster of all samples, and found that cluster 16 and cluster 7 expressed high levels of C2 signature while the hepatocytes (cluster 15) expressed highest levels of C1 signature (Fig. S6b). We then specifically determined the C1/C2 expression in tumor-specific clusters in each tumor sample. Again, in contrast to C1 signature expression, we found that C2 signature was highly expressed in cluster 16 and cluster 7 in tumor samples (NEJ634, NEJ687 and NEJ723) (Fig. S6c). However, we noticed that tumor sample NEJ709 expressed comparable levels of C1 and C2 signatures (Fig. S6c). Taken together, these data further support the tumor heterogeneity of ABC-Myc tumors.

Overall, these data support that ABC-Myc hepatoblastoma-like tumors (cluster 7 and 16) resemble human hepatoblastoma that bear the cellular heterogeneity consisting of tumor cells expressing high levels of erythroid genes[65], raising one possibility that a population of these cells could behave like the 'Tumor cluster 4 Erythroid' in human hepatoblastoma that is resistant to most chemotherapeutic agents[65].

## Spatial transcriptomic analysis of ABC-Myc tumors validates the heterogeneity of hepatoblastoma-like cells

To further understand the cellular heterogeneity of ABC-Myc tumors, we performed spatial transcriptomics analysis of 4 ABC-Myc tumors and 3 healthy mouse livers, using Visium Spatial gene expression from 10× Genomics. First, we examined the expression profiles of the 16 clusters from the scRNA-seq analysis. Consistent with the scRNA-seq

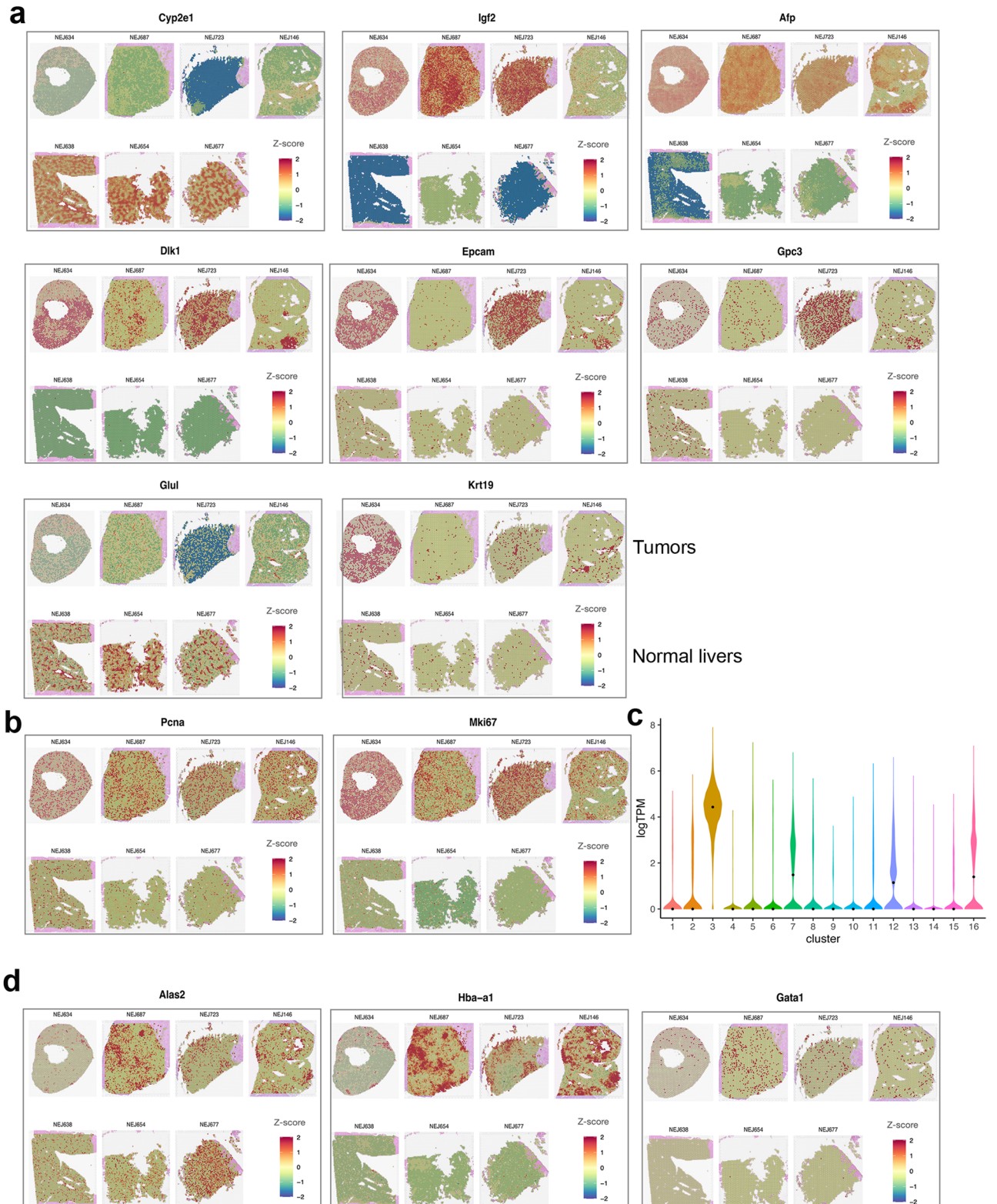

**Fig. 5 | Spatial transcriptomic analysis of ABC-Myc tumors validates the heterogeneity of hepatoblastoma-like cells. a**, **b**, **d** Spatial feature plots of selected marker genes Cyp2e1, Igf2, Afp, Dlk1, Epcam, Gpc3, Glul, Krt19 **a**, Pcna and Mki67 **b**, Alas2, Hba-a1, Gata1 **d**. In each plot, the top row is tumor samples, and the bottom row is normal liver samples. The gene-spot matrices were analyzed with the Seurat package (versions 3.0.0/3.1.3) in R. Spatial spots were colored by the z-transformed expression values across samples, showing extensive gene expression heterogeneity. **c** Violin plot for Mki67 expression levels across 16 clusters in single-cell RNA-seq data. Source data are provided as a Source Data file.

results, genes highly expressed in clusters (2, 3, 7, 12, 16) from tumor tissues were all highly expressed in the spatial gene expression profiles of tumor tissues (NEJ634, NEJ687, NEJ723) and one validation sample (NEJ146) (Fig. S7a, b). Then we specifically investigated the individual genes that indicate hepatocyte differentiation (*Cyp2e1*), hepatoblastoma markers (*Igf2*, *Afp*, *Glul*, *Krt19*) and embryonal hepatoblastoma stem cell markers (*Dlk1*, *Epcam* and *Gpc3*) (Fig. 5a). We observed overall reduction in expression of *Cyp2e1* in tumor tissues in comparison with normal livers but marked induction of *Igf2*, *Afp*, *Dlk1*, *Epcam* and *Gpc3* in tumor areas. While *Glul* was expressed in tumor tissues, its expression was higher in normal livers. *Krt19* was highly expressed in some areas in tumor tissues. Interestingly, the expression of stem cell markers *Dlk1*, *Epcam* and *Gpc3* were inter-tumor heterogeneous, being more highly expressed in NEJ634 and NEJ723 tumors in comparison with NEJ687 and NEJ146 tumors, in contrast to the expression profiles of *Cyp2e1* and *Glul*. These data indicate that NEJ634 and NEJ723 tumors are less differentiated. The intra-tumor heterogeneity was also demonstrated by spatial transcriptomics analysis. For example, the *Afp* and *Igf2* expression in NEJ634 was nearly uniformly expressed across the whole tissue section; however, the expression of stem cell markers (*Dlk*, *Epcam* and *Gpc3*) were heterogeneous (Fig. 5a).

Our bulk RNA-seq and scRNA-seq analyses classified ABC-Myc tumors as highly proliferative C2-Class (Figs. 3g, S3, S6). We therefore examined the C1 and C2 gene signatures in spatial transcriptomics. In our spatial expression analysis, we found that C1 expression was dominantly high in normal livers. However, the C1 expression in normal livers were not evenly distributed and heterogeneity was observed across the whole section. While C1 expression in tumor samples were greatly lower than in normal livers, spatial heterogeneity was present and samples NEJ687 and NEJ146 expressed higher levels of C1 than NEJ634 and NEJ723 (Fig. S7c). Correspondingly, C2 signature was highly expressed in all tumors present with heterogenous expression across the tumor sections. NEJ634 and NEJ723 expressed higher levels of C2 in comparison with NEJ687 and NEJ146 (Fig. S7c), indicative of intra- and inter-tumoral heterogeneity. The spatial gene expression profile validated that tumor tissues expressed high levels of Pcna and Mki67, two cellular proliferation markers (Fig. 5b). Nevertheless, the expression of Mki67 tended to be more prevalent in NEJ634 and NEJ723 tumors, which expressed high levels of stem cell markers and C2 signature. Our scRNA-seq analysis revealed Mki67 was highly expressed in clusters 3, 7, 12, and 16 from tumor samples, and cluster 3 (HB associated Erythroid) expressed the highest levels of Mki67 (Fig. 5c), in line with results from human hepatoblastoma that HB-associated erythroid cells were highly proliferative[65].

The erythroid lineage markers expressed in multiple tumor and tumor-associated clusters were also reflected in spatial gene expression profiles, as indicated by the high levels of *Alas2*, *Hba-a1* and *Gata1* across tumor areas (Fig. 5d). Nevertheless, the expression of these erythroid markers in tumor sample NEJ634 was much lower than other three tumor samples, indicating tumor heterogeneity. Notably, NEJ634 expressed the highest levels of *Krt19* among the 4 tumor samples.

Taken together, our spatial transcriptomics analysis supports the scRNA-seq and pathology results revealing the heterogeneity of ABC-Myc hepatoblastoma-like tumors.

### Genome-wide screen of cancer dependency genes in an ABC-Myc-derived hepatoblastoma-like tumor cell line

As a rare cancer and because of the lack of relevant disease models, drug-actionable targets in hepatoblastoma have only rarely been reported. To identify the dependency genes of hepatoblastoma, we established cell lines from ABC-Myc tumors, which can be readily passaged in vitro in standard DMEM media. We used one of these highly aggressive cell lines, NEJF10, to conduct a genome-wide, pooled CRISPR-Cas9 screening to uncover new therapeutic targets of hepatoblastoma (Fig. 6a). We identified 1583 essential genes that are

required for NEJF10 survival ($p < 0.02$, FDR < 0.25) (Fig. 6b, Supplementary Data 5), including 100 targets with inhibitors available (Supplementary Data 6), and 30 tumor suppressive genes ($p < 0.001$, FDR < 0.25) whose knockouts lead to increased proliferation (Fig. 6c, Supplementary Data 5). Pathway enrichment analysis of the essential genes using the 'Genetic and Chemical Perturbation database'[51] showed that they are enriched in class 2 hepatoblastoma genes (CAIRO_HEPATOBLASTOMA_CLASSES_UP, $n = 612$), and are targets of BMP2, DREAM complex, MYC and β-catenin (Fig. 6d). After compiling the essential genes and tumor suppressive genes, we found that classical cancer signaling pathways may exert important functions in the progression of hepatoblastoma, including the PI3K pathway (*Pten*), the p53 pathway (*Cdkn2a*, *Trp53*, *Myh9*, *Sox4*, *Dapk3*), and the RAS-RAF-MEK-mTOR pathway (*Grb2*, *Ptpn11*, *Kras*, *Raf*, *Map2k2*, *Mapk1*, *Rheb*, *mTor*, *Nf1*, *Lztr1*, *Rasa2*, *Dusp9*) (Fig. 6e).

Genomic sequencing analysis of thousands of human tumors demonstrates that Hippo signaling pathway is widely dysregulated[71], and plays a critical role in the tumorigenesis of liver cancers[18–20]. Our CRISPR screen identified key components of the Hippo pathway in hepatoblastoma (Fig. 6f), including the oncogenic transcription factor YAP and its interaction partner TAZ, both of which are required for cancer cell survival. The tumor suppressive genes (*Taok1*, *Lats1*, *Nf2*), upstream of the Hippo pathway that inhibit YAP through phosphorylation-induced cytoplasmic retention and degradation, are important for hepatoblastoma proliferation. *Amotl2*, which encodes a Motin family member, Angiomotin-like 2, is a tumor suppressor that negatively regulates the YAP and TAZ function via AMOT-mediated tight junction localization[72]. RhoA is a GTPase that controls YAP/TAZ translocation through promoting actin polymerization and stress fiber formation[73–75]. One study shows that the RhoA–YAP–MYC signaling axis promotes the development of polycystic kidney disease[76].

To determine if genetic context dependencies could exist, we included two additional ABC-Myc cell lines, NEJF1 and NEJF6, for genome-wide CRISPR screening using a similar approach. We identified 1346 and 846 essential genes ($P < 0.02$, FDR < 0.25), 37 and 24 tumor suppressive genes ($P < 0.02$, FDR < 0.25) in NEJF1 and NEJF6, respectively (Fig. S8a–d, Supplementary Data 7, 8). Venn analysis revealed that over 50% of essential genes overlapped among the three cell lines, but that each cell line also showed unique gene dependencies (Fig. 6g, Supplementary Data 9). However, there was less overlapping in tumor suppressive genes among the three cell lines (Fig. 6g, Supplementary Data 9). Nevertheless, like NEJF10, the cancer signaling pathways (PI3K, mTOR, and p53, Hippo) were also enriched in NEJF1 and NEJF6 cell lines (Fig. S9a–d), indicating that each of these pathways are shared and important for ABC-Myc hepatoblastoma-like cells.

To determine if murine and human hepatoblastoma share common cancer dependency genes, we analyzed the essential genes identified through genome-wide CRISPR-Cas9 screen in Huh6, a human hepatoblastoma cell line included in the screening of a first-generation pediatric cancer dependency map[77]. Huh6 bears *TP53(Asn239Asp, Ala159Asp)* and *CTNNB1(Gly34Val)* mutations (depmap.org). 1411 essential genes (CRISPR score threshold −0.7) and 22 anti-proliferative genes (CRISPR score threshold 0.4) were identified in Huh6 cells (Fig. S10a, Supplementary Data 10). Pathway enrichment analysis showed that the essential genes are similarly enriched with class 2 hepatoblastoma genes, targets of BMP2, DREAM complex, MYC and β-catenin (Fig. S10b), and all these pathways are commonly shared by the ABC-Myc cell lines. VENN analysis of essential genes in Huh6 and ABC-Myc cell lines showed that 72% (1020 out of 1411) of the essential genes in Huh6 are shared by the three ABC-Myc lines (Fig. S10c, Supplementary Data 11), including those targetable genes (i.e., *CDK7*, *CDK9*, *PRMT1*, *PRMT5*, *NEDD8*, *PLK1*) (Fig. S10a), which are involved in a variety of biological functions (Fig. S10d). We particularly compared the Huh6 with NEJF10 cell line by VENN analysis. 61.7% of the essential genes in Huh6 cells were shared by NEJF10 (Fig. S10e). Among the

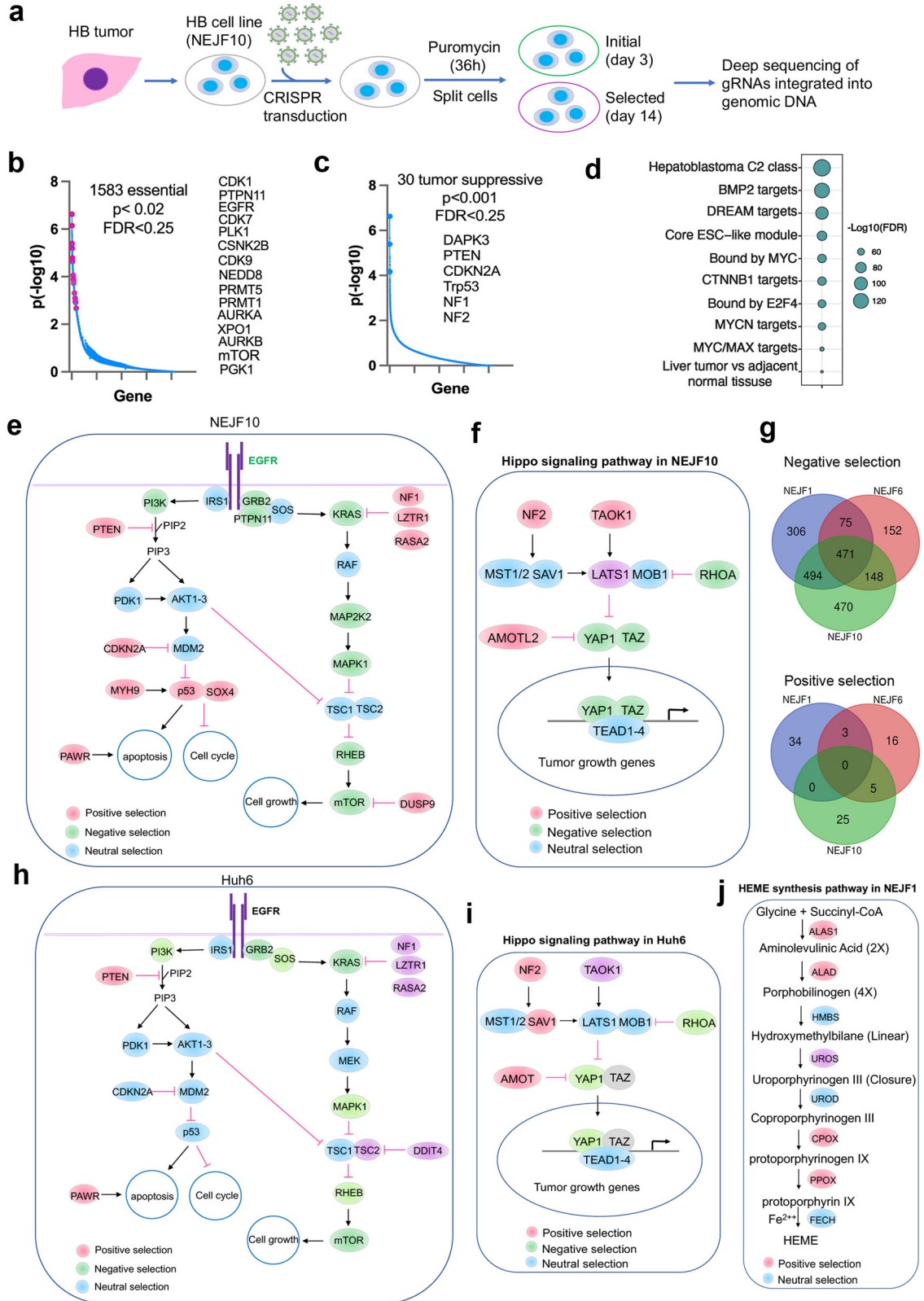

tumor suppressors, we found two (*NF2*, *PTEN*) were commonly shared between NEJF10 and HuH6 cells (Fig. S10f). Importantly, Huh6 shared common cancer pathways involved in PI3K, mTOR and Hippo signaling (Fig. 6h, i). We identified several p53 pathway genes in ABC-Myc cells (Figs. 6e, S9a, c). However, Huh6 has a *TP53* mutation and, therefore, no selective pressure was conferred in the CRISPR screen. In summary, the oncogenic pathways and the therapeutically targetable genes are

conserved both in our murine hepatoblastoma model and human hepatoblastoma.

Interestingly, in addition to the conserved oncogenic pathways identified in ABC-Myc cell lines as discussed above, the heme biosynthesis pathway appeared to perform tumor suppressive functions in NEJF1 but not in NEJF6 and NEJF10 (Fig. 6j). Among the 8 enzymes responsible for catalyzing heme biosynthesis from glycine and

**Fig. 6 | Cancer dependency genes and oncogenic pathways of hepatoblastoma cells. a** Diagram showing the procedure of genome-wide CRISPR screen of cancer dependency genes using ABC-Myc NEJF10 cell line. **b, c** Cancer essential genes and tumor suppressors identified in NEJF10 cell line with FDR cutoff <0.25. X-axis represents the total gene number. Y axis represents the p value in -log10. *P* value obtained by permutation test and FDR calculated from the empirical permutation *p*-values using the Benjamini-Hochberg procedure by MAGeCK. **d** Pathway enrichment analysis of cancer dependency genes identified in ABC-Myc cell line by using GSEA and CGP (chemical and genetic perturbations) dataset. **e** Canonical cancer pathways enriched in genes identified by CRISPR screen in NEJF10 cell line. **f** Hippo signaling pathway enriched in genes identified by CRISPR screen in NEJF10 cell line. **g** Venn analysis of essential genes (negative selection) and tumor suppressive genes (positive selection) identified from NEJF1, NEJF6, NEJF10. CRISPR FDR cutoff <0.25. **h** Cancer dependency genes identified in human hepatoblastoma Huh6 cell line from DepMap data (www.depmap.org). **i** Hippo signaling pathway enriched in genes identified by CRISPR screen in Huh6 cell line. **j** Heme biosynthesis pathway was enriched in positive selection in NEJF1 cells. Source data are provided as a Source Data file.

succinyl-CoA, 6 of them inhibited NEJF1 proliferation or survival. Inactivating mutations in heme synthesis genes define a group of diseases known as porphyria[78]. Recent studies revealed that acute hepatic porphyria is associated with increased risk of hepatocellular carcinoma (HCC). *HMBS*, although it was not enriched in our screening, has been shown to bear bi-allelic inactivating mutations in acute intermediate porphyria associated–HCC and sporadic HCC[79,80]. Our screening data suggest that the tumorigenesis and/or progression of a subgroup of hepatoblastomas could benefit from metabolic dysfunction due to inactivation of heme synthesis.

### Genetic mapping of chemotherapy response

Conventional chemotherapeutic agents (i.e., cisplatin, doxorubicin) play a critical role in hepatoblastoma treatment[81]. These chemotherapeutic agents have significant toxicities and, in some cases, limited anti-cancer efficacy. Thus, a better understanding of the genetic response of hepatoblastoma cells to chemotherapy may help to develop more effective and safer individual and combination therapies. To map the genetic response of chemotherapy, NEJF10 cells were transduced with a lentiviral pooled genome-wide sgRNA library and were divided into control and treatment groups (Fig. 7a). Cells were treated with doxorubicin at two doses, IC$_{20}$ (5 nM) and IC$_{90}$ (30 nM) (Fig. S11), for 7 and 14 days, respectively. We identified hits whose mutation caused sensitization (negative selection) and resistance (positive selection) to doxorubicin. At the sublethal IC$_{20}$ dose, we identified 315 genes of negative selection ($p < 0.01$, FDR $< 0.25$) and 20 genes of positive selection ($p < 0.01$, FDR $< 0.25$) (Fig. 7b, Supplementary Data 12). However, at the IC$_{90}$ dose, we only identified 70 positive selection genes ($p < 0.001$, FDR $< 0.3$) (Fig. 7d, Supplementary Data 13).

Functional protein association network analysis of negative selective genes at IC$_{20}$ revealed that most of them are physically and/or functionally connected, and function in DNA repair through non-homologous end-joining (NHEJ) (e.g., *Prkdc*, *Lig4*, *Xrcc4*) or homologous recombination (e.g., *Rad51*, *Rpa2*, *Xrcc2*), mitochondria (e.g., *Mtg2*, *Polg*, *Chchd3*), small nuclear RNA (snRNA) biogenesis through RNA polymerase II (e.g., *Ctu2*, *Snrnp40*, *Cstf1*), gene transcription (e.g., *Ints6*, *Ccnc*, *Asun*), and mitosis (e.g., *Aurka*, *Tpx2*) (Fig. 7c). These data suggest that loss of function of NHEJ or homologous recombination-mediated DNA repair may further worsen the DNA damage induced by doxorubicin, leading to enhanced cell death. *Prkdc*, which encodes DNA-PK to sense double strand DNA breaks and regulates DNA repair via NHEJ, has emerged as a new therapeutic target[82]. Interestingly, disruption of a dozen of snRNA biogenesis genes also promoted the effect of doxorubicin. Notably, loss of function of *Aurka*, which encodes Aurora kinase A (AURKA) that is implicated in the regulation of cellular mitosis, led to enhanced effect of doxorubicin, consistent with recent studies showing that AURKA inhibitors potentiate the cancer cell killing of doxorubicin[83,84].

20 positive selection genes at IC$_{20}$ were obtained, including classical tumor suppressor genes such as *Cdkn2a*, *Pten*, *TrpS3* (Fig. 7b). As discussed above, *Dapk3* and *Sox4* are involved in regulation of the p53 pathway while *Lztr1* inhibits Ras activity. *Rock2* and *Myl6* encode proteins functioning downstream of Rho GTPase activity[85]. Although the mechanism of this pathway in chemoresistance remains to be investigated, one previous study showed that pharmacological inhibition of ROCK signaling enhances cisplatin resistance in neuroblastoma cells[86]. *Wdr77*, encoding the non-catalytic component of the methylosome complex, composed of PRMT5, WDR77 and CLNS1A[87], has germ-line mutations in patients that predispose to familial papillary thyroid cancer[88]. Transcriptome changes in pathways were enriched in the processes of cell cycle promotion and apoptosis in *WDR77* mutated tumors[88]. These data indicate that loss of function of tumor suppressors blocks the effect of sublethal dose of doxorubicin. However, under the IC$_{90}$ lethal dose, the pathways conferring doxorubicin resistance were distinct from those under IC$_{20}$ dose selection (Fig. 7e). In addition to the genes involved in apoptosis, DNA replication and mitosis, the major components of these pathways are involved in regulation of homeostasis of RNA and protein, including pre-mRNA splicing (e.g., *Sf3b5*, *Hnrnpa1*, *Smu1*), protein translation and degradation (*Rpl7l1*, *Rpl3l*, *Psma1*, *Psma4*, *Cct5*) (Fig. 7e, f). MAPK1, APAF1 and CASP9 are engaged in cytochrome C-mediated apoptotic response. *Faf1* encodes FAS-Associated Factor 1 (FAF1) that acts as a tumor suppressor by regulation of apoptosis and NF-κB activity, and ubiquitination and proteasomal degradation[89]. *Topors* encodes topoisomerase I-binding RING finger protein, which is a coactivator of p53 in growth suppression induced by DNA damage[90]. While it is not surprising that inactivation of the apoptotic pathway leads to resistance to chemotherapy-mediated cancer killing, the mechanisms of RNA splicing and protein homeostasis in doxorubicin resistance are largely unknown. Nevertheless, these data provide a rationale to develop strategies to enhance efficacy of chemotherapy.

### Drug screening using ABC-Myc-derived hepatoblastoma cells to identify new therapies

To develop a high-throughput screen platform using our ABC-Myc cell lines, we optimized the NEJF10 cell line in 384-well plate and treated cells with drugs currently being used for clinical cancer treatment, including 125 FDA-approved cancer drugs. With the range of 0.7–2 μM of tested compounds, 51 of them inhibited >50% of cell viability (Fig. S12a), including conventional chemotherapeutic agents such as topoisomerase inhibitors, tubulin inhibitors, and nucleotide synthesis inhibitors (Fig. S12b). We also found that ABC-Myc cells were sensitive to mTOR and MEK inhibitors, tyrosine kinase inhibitors, HDAC inhibitors and proteasome inhibitors, consistent with our CRISPR screening data showing that mTOR, EGFR, HDAC3, and proteasome are essential, indicating that these inhibitors may have clinical potential to treat hepatoblastoma patients. mTOR is activated downstream of YAP/TAZ in a YAP/β-catenin hepatoblastoma mouse model[91]. mTOR inhibitors blocked hepatoblastoma growth in vitro and in xenograft models[91,92], and one clinical study showed that two hepatoblastoma patients treated with the mTOR inhibitor everolimus after liver transplantation did not develop any metastases[93]. The following clinical trial resulting from this study reported that 10 patients with liver malignancy received everolimus after liver transplantation, and none of these patients developed recurrence by the endpoint of the study[94]. These data indicate that mTOR inhibition may be useful for treating hepatoblastoma patients especially for those who need liver transplantation, by benefiting from its anti-tumorigenic and immunosuppressive properties. One clinical study revealed that EGFR expression was elevated in hepatoblastoma specimens[56], as a target of

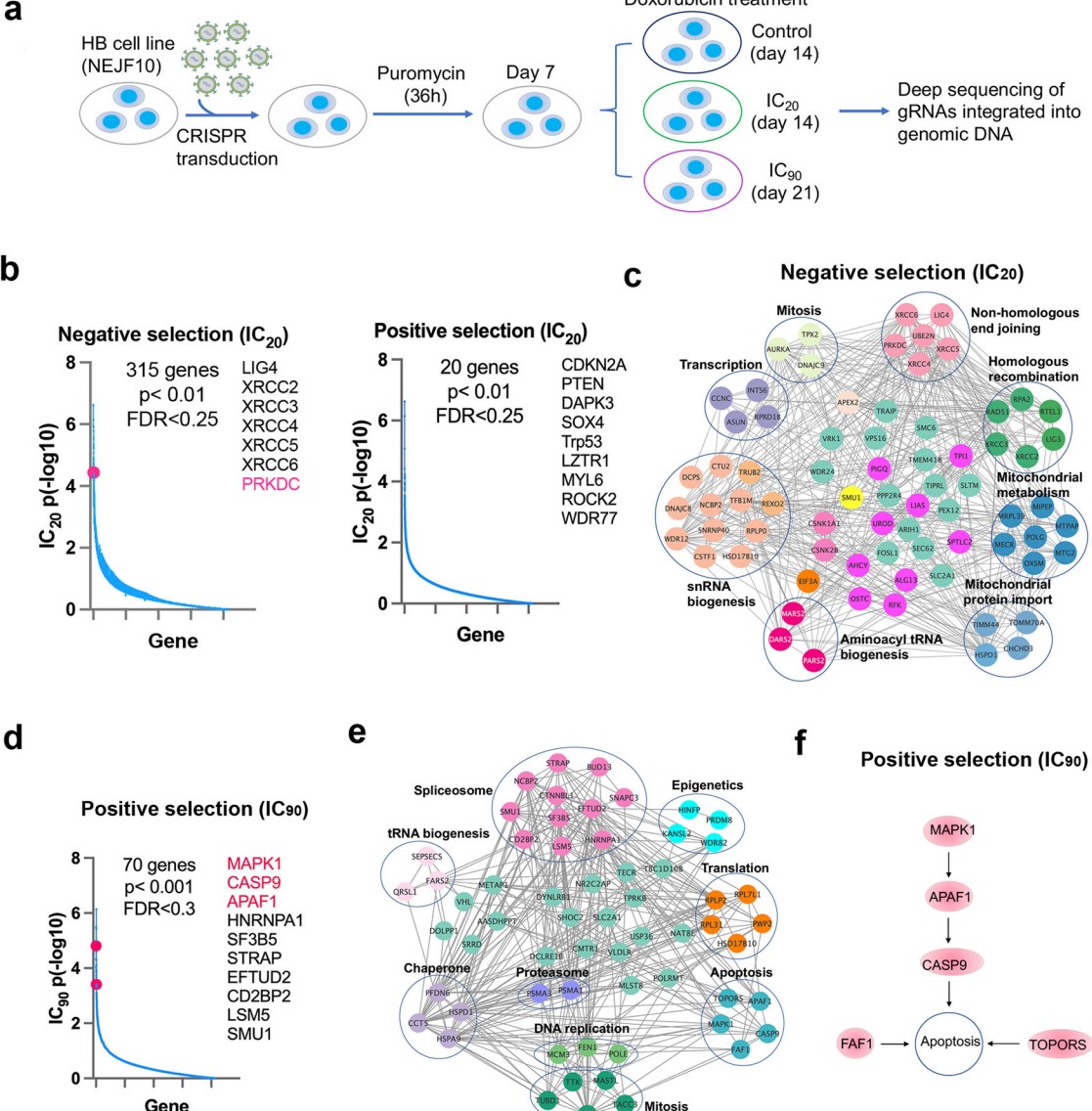

**Fig. 7 | Identification of genetic modifiers of chemotherapy. a** Diagram showing the procedure of genome wide CRISPR screening for the genetic modifiers of doxorubicin in NEJF10 cell line. **b** Negative selection and positive selection under IC$_{20}$ of doxorubicin. The red dot highlighted in the graph indicates *Prkdc* gene that is focused on in this study. CRISPR FDR cutoff <0.25. X-axis represents the total gene number. Y axis represents the p value in -log10. *P* value obtained by permutation test and FDR calculated from the empirical permutation *p*-values using the Benjamini-Hochberg procedure by MAGeCK. **c** Pathways within a protein–protein interaction network enriched in negative selection under IC$_{20}$ of doxorubicin.

**d** Positive selection under IC$_{90}$ of doxorubicin. CRISPR FDR cutoff <0.3. X-axis represents the total gene number. Y axis represents the *p* value in -log10. The genes highlighted in pink color indicates genes involved in apoptosis pathway. *P* value obtained by permutation test and FDR calculated from the empirical permutation *p*-values using the Benjamini-Hochberg procedure by MAGeCK. **e, f** Pathways within a protein-protein interaction network enriched in positive selection under IC$_{90}$ of doxorubicin€) and apoptotic pathway **f**. Network analysis performed using STRING program. Source data are provided as a Source Data file.

the Wnt/β-catenin pathway in liver[95], which may explain why our ABC-Myc cell line is sensitive to genetic and pharmacologic inhibition of EGFR.

Our CRISPR screen revealed that *Cdk7* and *Aurka* were essential to ABC-Myc cells. Although the functions of CDK7 in hepatoblastoma have yet to be explored, CDK7 inhibition disrupts the transcriptional dependency of MYC-driven cancer[96]. Several CDK7 inhibitors have been developed and two are in clinical trials (https://clinicaltrials.gov/). We treated five ABC-Myc cell lines with a selective CDK7 inhibitor, Samuraciclib[97], and found that this compound potently killed ABC-Myc cells (IC$_{50}$ < 100 nM) (Fig. S12c), and validated the drug killing (Fig. S12e). *AURKA* encodes aurora kinase A protein that is critical to G2/M phase progression during cell cycle. MYC and AURKA form a

complex that is a potentially actionable target in MYC-driven cancers[98,99]. Indeed, in comparison with a primary human fibroblast cell line, the ABC-Myc cell lines were at least 100-fold more sensitive to the AURKA inhibitor, Alisertib (Fig. S12d). These data indicate that ABC-Myc cell lines recapitulate the therapeutic vulnerability of human MYC-driven cancers. Our drug screening strategy allows the validation of the candidates obtained by the genomic screening through a different approach.

**PRKDC inhibition enhances efficacy of chemotherapy**
Loss of function of *Prkdc* synergized with doxorubicin effect in our CRISPR-Cas9 screen, providing a rationale to combine PRKDC inhibitors with chemotherapy to enhance efficacy. To validate the

role of PRKDC, we used RNAi to knock down *Prkdc* in NEJF10 cells (Fig. 8a), followed by doxorubicin treatment. Indeed, knockdown of *Prkdc* enhanced tumor cell killing by doxorubicin (Fig. 8b). We then tested this hypothesis by treating NEJF10 cells with doxorubicin and a selective PRKDC inhibitor, AZD7648, which shows >90-fold cellular selectivity over its structurally related members ATM, ATR, and mTOR[100], and has purity of >99% (Fig. S13). The colony formation assay demonstrated that AZD7648 synergized with doxorubicin to inhibit the cell survival of NEJF10 cells (Fig. 8c). PrestoBlue assay with BLISS index analysis further corroborated the synergistic effect of a PRKDC inhibitor and doxorubicin (Fig. S14a). Annexin V staining followed by flow cytometry analysis showed that the PRKDC inhibition and doxorubicin combination induced greater apoptosis (Fig. S14b). The synergistic effect of doxorubicin and AZD7648 was verified in additional ABC-Myc cell lines by colony formation assay (Fig. 8d). We further validated the synergistic effect of doxorubicin and AZD7648 in HepG2 cells, a human hepatoblastoma cell line[101], and obtained similar results (Fig. S14c, d). We also obtained similar results when combining doxorubicin with another PRKDC inhibitor (Fig. S15), NU7441 that has a distinct chemotype from AZD7648[102]. We then tested the combination therapy using our ABC-Myc mouse model. While monotherapy showed no benefit to ABC-Myc mice, the combination of AZD7648 and doxorubicin significantly extended mouse survival (Fig. 8e), reducing the liver weight significantly, comparable to the normal liver weight (Fig. 8f). In parallel, we tested the combination therapy using HepG2 xenografts. While doxorubicin showed modest anticancer effect, the combination of both led to a significant tumor growth delay in comparison with doxorubicin or AZD7648 alone (Fig. 8g). The greater efficacy of combination of doxorubicin and AZD7648 was further verified in a patient-derived xenograft hepatoblastoma model (Fig. 8h). In summary, the results from our CRISPR screen of the ABC-Myc model have identified therapeutic combinations that may be used in future clinical trials.

## Discussion

Hepatocellular malignancies have become a leading cause of cancer-related deaths in people of all ages[2,103,104]. Notably, the worldwide hepatoblastoma incidence has a greater rate of increase than other pediatric cancers[2]. Surgical resection is critical for curing hepatoblastoma (HB), yet two-thirds of patients have unresectable tumors at diagnosis[81], and so they need induction chemotherapy to enable surgical resection. Patients with resectable tumors have an event-free survival (EFS) of 80–90% and can be cured with surgical resection combined with conventional chemotherapy[81]. However, children with high-risk disease have poor outcomes with EFS under 50%[81,105,106]. New therapeutic approaches for high-risk patients remain desperately needed. Unfortunately, lack of cell lines and animal models that resemble high-risk human hepatoblastoma impedes our understanding of the pathogenesis of hepatoblastoma and identification of druggable targets. To meet this unmet clinical need, this study has (1) developed and validated the ABC-Myc hepatoblastoma-like model that closely resembles the histology of human hepatoblastoma and recapitulates high-risk human disease at transcriptional levels, (2) generated ABC-Myc cell lines based upon the genetic model which are suitable for genome-wide genetic screen and high-throughput drug screens, (3) mapped the cancer dependency genes in ABC-Myc cells and defined the key oncogenic pathways that are shared by human hepatoblastoma cell lines, (4) identified the genetic modifiers of chemotherapy by a genome-wide CRISPR screen, and (5) developed a combination therapy based upon the screening results that was translated to human hepatoblastoma models. Thus, this study has provided resources including disease models, targetable cancer dependency genes, and potentially more effective combination therapy approaches.

Previous approaches have been applied to establish hepatoblastoma models, including xenograft implantation[107–115], generation of transgenic mice[29,55,116] and hydrodynamic tail vein injection of oncogenes[22,24], and each of these models has its pros and cons[35]. Additionally, most of these models recapitulate the well-differentiated fetal type of hepatoblastoma, which usually has a good clinical outcome even without chemotherapy administration[117]. In this study, we created a hepatocyte-specific transgenic c-Myc model, ABC-Myc, which rapidly develops multifocal hepatic neoplasms with pathological features of mixed fetal and embryonal hepatoblastoma, the most common histologic subtype in human disease. Overall, the poorly differentiated histology is consistent with the pediatric C2 phenotype. Nevertheless, this murine model also contains histologic features of the subclassification of pediatric hepatoblastomas with hepatocellular carcinoma features that were previously called transitional liver cell tumors (TLCT)[9].The phenotypic plasticity that is observed in this Myc-driven murine model of hepatoblastoma is documented in pediatric hepatoblastoma where some hepatoblastomas can be classed into the transcriptomic subgroup "liver progenitor" differentiation state that appears to correlate with the C2A molecular for human hepatoblastomas. This differentiation state is highly proliferative, immune cold, composed of embryonal histologies, enriched for self-renewal and pluripotency transcription factors including MYCN and may represent a model for relapse or hepatoblastomas with metastatic potential[118], which fits with the histology and behavior of ABC-Myc hepatoblastoma-like tumors. The phenotype and molecular characterization of our model may also match with the C2A phenotype of which HCN-NOS can be included as well as a "liver progenitor" sub-grouping, and which also has been correlated with high-risk MRS-3B subgrouping[15].

Currently the prognostic significance of the presence of or relative proportions of different morphologic patterns that may arise in pediatric and adolescent hepatoblastomas remains unclear and there is still variability in the subclassifications of pediatric liver tumors because histology can be variable within a liver tumor and sampling may limit the ability to observe the variations in morphologic patterning that can be observed within one tumor. One exception is the presence of foci of neoplastic cells having a small-cell-undifferentiated morphology, which is not observed in this model. Additionally, INI-1 staining was retained in all sampled hepatoblastoma-like tumors indicating that the diagnosis of rhabdoid-like tumor is not appropriate for this model. In pediatric patients it is important to differentiate hepatoblastoma from hepatocellular carcinoma because of treatment and prognosis[38]. The Myc-driven murine hepatoblastoma-like tumors demonstrate phenotypic plasticity of hepatocyte lineage committed stem/progenitor cells, suggesting that some tumor components have HCN-NOS features.

Neither regionally invasive nor metastatic disease is a feature of this model, and this biological behavior is consistent with the known role for MYC to drive bulky tumor growth within the liver microenvironment[28]. Metastatic disease is not observed in this model from several reasons including genetics or reduced survival time from localized disease. Alternatively, it may be an extremely rare event in the model. In any of these scenarios it is reasonable to hypothesize that additional genetic or non-genetic drivers are probably important for the invasive and metastatic potential in this model and from what is known in the literature about other hepatoblastoma model systems and pediatric hepatoblastomas with demonstrated metastatic potential. Therefore, this model is considered as a hepatoblastoma-like model which represents multifocal aggressive tumors in liver without metastasis.

Mutation of CTNNB1 occurs in about 48–67% of pediatric hepatoblastoma cases, which is different from how liver tumors arise in our model. MYC overexpression is a significant genetic event in pediatric liver tumors including pediatric HCCs. In humans, mutational and

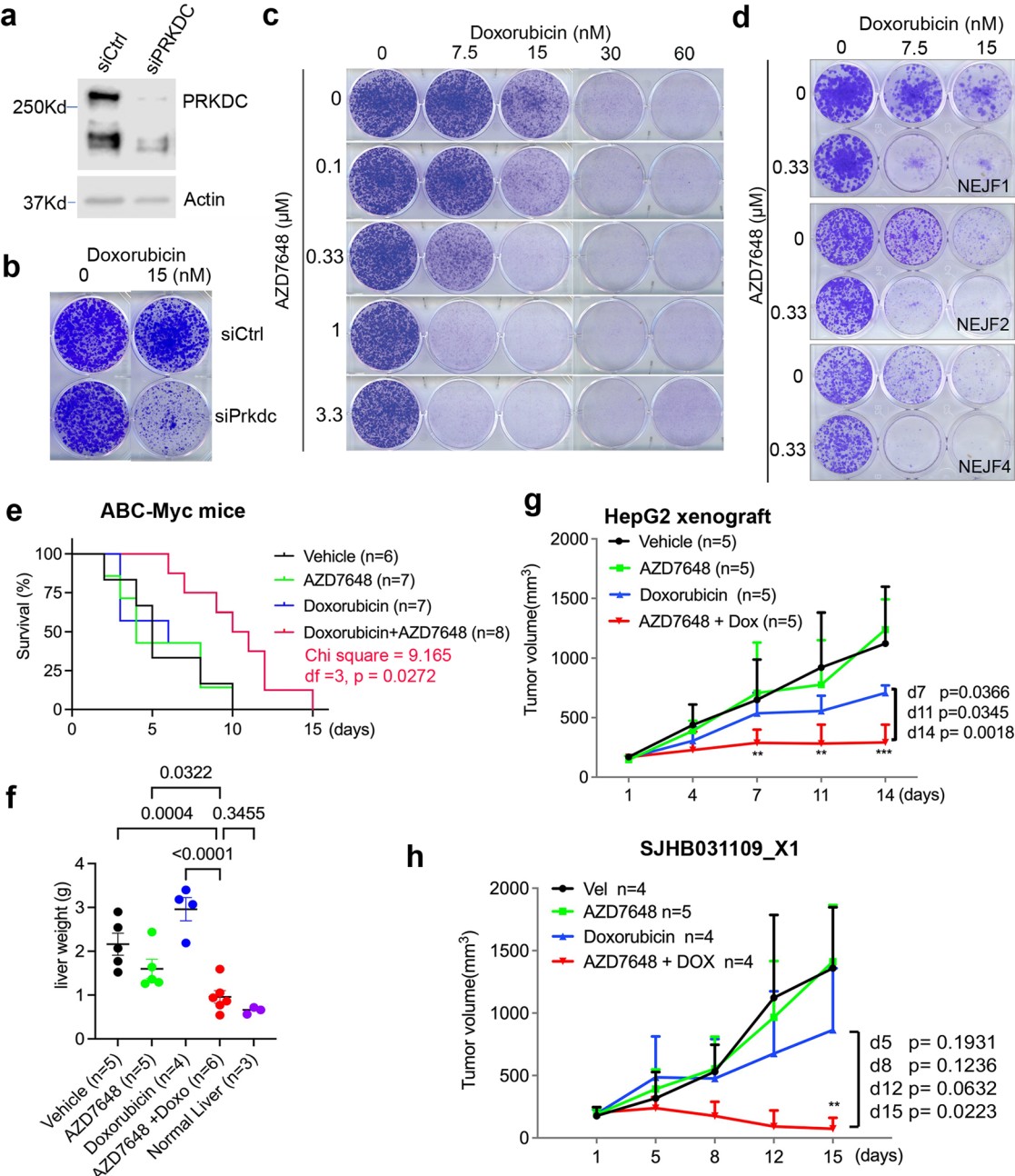

**Fig. 8 | Combination of doxorubicin and PRKDC inhibitor has a better anticancer efficacy. a** Western blot showing knockdown of PRKDC in NEJF10 cells after 72 h transfection of siRNA into NEJF10 cells. The blots are representative of three independent experiments. **b** Colony formation showing the effect of combination of *Prkdc* knockdown and doxorubicin treatment for 4 days. The images are representatives of 3 independent experiments. **c** Colony formation showing the synergistic effect of combination of different concentrations of doxorubicin and AZD7648 to treat NEJF10 for 5 days. The images are representatives of three independent experiments. **d** Colony formation for NEJF1, NEJF2, NEJF4 treated with doxorubicin and AZD7648 for 5 days. The images are representatives of two independent experiments. **e** Survival rate for ABC-Myc treated with vehicle (*n* = 6), doxorubicin (0.75 mg/kg, twice weekly; *n* = 7) and AZD7648 (50 mg/kg, twice daily; *n* = 7), and combination of doxorubicin and AZD7648 (*n* = 8). P value calculated by log-rank (Mantel-Cox) test method. **f** Liver weight after treatment in each group of

ABC-Myc mice (vehicle *n* = 5, AZD7648 *n* = 5, doxorubicin *n* = 5, combination of doxorubicin and AZD7648 *n* = 6) and normal liver (*n* = 3) in age matching mice. Data are presented as mean ± SD. *P* value calculated by two-sided student t test. **g** Tumor volume for each treatment group of HepG2 xenografts with vehicle (*n* = 5), doxorubicin (1.0 mg/kg, twice weekly; *n* = 5), AZD7648 (50 mg/kg, twice daily; *n* = 5) and combination of doxorubicin and AZD7648 (*n* = 5). *p* value calculated by two-sided student t test for two groups (doxorubicin vs doxorubicin/AZD7648) at each time point. **h** Tumor volume for each treatment group of SJHB031109_X1 PDX xenografts with vehicle (*n* = 4), doxorubicin (0.75 mg/kg, twice weekly; *n* = 4), AZD7648 (50 mg/kg, twice daily; *n* = 5) and combination of doxorubicin and AZD7648 (*n* = 4).Data are presented as mean ± SD. *p* value calculated by two-sided student t test for two groups (doxorubicin vs doxorubicin/AZD7648) at each time point. Source data are provided as a Source Data file.

immunohistochemical analyses of β-catenin are not always correlative. Nuclear localization/expression of β-catenin in human hepatoblastomas is not always diffuse and can be regional or focal and is an imperfect surrogate for molecular testing. The fact that nuclear

localization of β-catenin does not have to be diffuse in Wnt-driven human pediatric liver tumors is of interest and points to subpopulation heterogeneity that drives aspects of tumor initiation, growth, and biological aggressiveness in human pediatric HBs and HCCs. These

observations may help us understand how tumor heterogeneity correlates with biological aggressiveness. Interestingly the IHC staining pattern for β-catenin in our model is localized to the membrane and cytoplasm of all HB-like tumors and is comparable to staining patterns that have been observed in pediatric tumors classified as HBs with HCC features including HCN-NOS and HB-FPA. Our model may represent locally aggressive, spontaneously arising hepatoblastoma cases where a mutation driving constitutive WNT/β-catenin signaling is absent. Therefore, lack of strong nuclear translocation of the wild-type β-catenin may suggest that MYC overactivation relieves the selective pressure on β-catenin, since MYC is a key downstream effector of the Wnt-β-catenin pathway[27,30,119].

Molecular profiling demonstrates that the ABC-Myc hepatoblastoma resembles high-risk human disease, making this model helpful for understanding the mechanisms of tumorigenesis and for testing therapy response in high-risk hepatoblastoma. Based on our ABC-Myc hepatoblastoma-like model, we generated cell lines that can be readily passaged in vitro in either 2D or 3D format. Using three ABC-Myc cell lines, we mapped the genome-wide cancer dependency genes of ABC-Myc hepatoblastoma. Approximately 70% of genes essential to Huh6, the human hepatoblastoma cell line, are overlapped with ABC-Myc dependency genes, suggesting that our ABC-Myc model and human hepatoblastomas share common oncogenic pathways. While hepatoblastoma lacks targetable somatic mutations, our screen identified numerous targetable genes including those encoding AURKA, CDK1, CDK7, CKD9, PLK1, PRMT1, PRMT5, EGFR and mTOR. The anticancer activity of inhibitors against AURKA, CDK7, mTOR and EGFR was validated in vitro in this study. While the clinical utility of drugging these targets with selective inhibitors awaits future testing, a PLK1 inhibitor, Volasertib, has shown anticancer activity in high-risk hepatoblastoma models[120]. While genetic mutations in classical cancer pathways (e.g., CDKN2A-MDM2-p53, PTEN-PI3K-AKT-mTOR, Ras-Raf-MEK) are rare in primary hepatoblastomas at diagnosis, our screen results show that ABC-Myc tumor cells depend on them, suggesting that they may be dysregulated in other ways in hepatoblastoma and/or play important roles in cancer progression and relapsed disease. Particularly, we found that the Hippo signaling pathway is essential to ABC-Myc cell survival, in line with previous studies which demonstrate that YAP1 and TAZ promote and maintain tumorigenesis of hepatoblastoma[21,22,121], supporting that our ABC-Myc model can be faithfully applied to identify hepatoblastoma-related oncogenic pathways. The roles of several tumor suppressive genes in hepatoblastoma identified from CRISPR screen are largely unknown. *Myh9*, which encodes nonmuscle myosin IIa, is a tumor suppressor in squamous cell carcinoma by regulating p53 stabilization[122]. While the transcription factor SOX4 could be oncogenic, it also has tumor suppressive functions by modulating p53 function[123]. Loss of function of *DAPK3* has been observed in several cancers[124], and DAPK3 regulates p53 activity by phosphorylating S20 on p53 to block MDM2-p53 interaction[125]. DAPK3 also has kinase independent tumor suppressive function by driving tumor-intrinsic immunity through the STING–IFNβ pathway[126]. *Lzrt1* encodes leucine zipper–like transcriptional regulator 1 that is associated with RAS, functioning as an adaptor to promote RAS ubiquitination, thus inhibiting RAS oncogenic functions[127]. *Rasa2*, encoding a RasGAP to inhibit RAS activity, is a tumor-suppressor gene with loss-of-function in ≥30% of human melanomas[128]. DUSP9, a dual-specificity phosphatase, may exert its antitumor functions by suppressing mTOR pathway[129,130]. The role of PAWR (Pro-Apoptotic WT1 Regulator) in cancer remains unknown and its antiproliferative activity might be due to its anti-apoptosis function[131]. Interestingly, we identified genes responsible for heme biosynthesis pathway exert antiproliferative functions in some ABC-Myc cells.

Adjuvant and neoadjuvant chemotherapy are the mainstays of hepatoblastoma treatment. Although chemotherapy intensification has resulted in improved outcomes for high-risk disease, this comes at the expense of significant ototoxicity and cardiotoxicity associated with cisplatin and doxorubicin. New combination therapies are badly needed to improve survival of patients. Using a genome-wide CRISPR screen, we have identified pathways whose loss-of-function enhance and antagonize the anti-tumor activity of doxorubicin. This has led to the identification of a more effective combination therapy in which a PRKDC inhibitor greatly enhances the efficacy of doxorubicin in ABC-Myc mice and human hepatoblastoma xenograft models. Recent studies have shown that chromosomal instability, TERT promoter mutations or telomerase activation were correlated with poor outcomes of the HB patients[65,132,133], suggesting that tumors in these patients may have DNA repair defects and thus, PRKDC inhibitors may be effective in these tumors. The advantage of our system is that we can test many different combinations based on screening of ABC-Myc cell lines in vitro, then validate the results in ABC-Myc mice in vivo, an approach not practical to carry out in clinical trials.

## Methods

### Animals
All experiments that involved the use of mice were performed under the protocol 615 issued to Jun Yang in accordance with the guidelines outlined by the St Jude Children's Research Hospital Institutional Animal Care and Use Committee (IACUC). Mice were housed with ambient temperature and humidity with 12 h light /12 h dark cycle controlled under specific-pathogen-free conditions (SPF) at the St Jude Children's Research Hospital mouse facility. Mice were allowed to feed and drink ad libitum. The maximal tumor burden permitted was 20% of mouse body weight, and in our experiments, the maximal tumor burden was not exceeded. Transgenic mice were euthanized through $CO_2$ inhalation with 3 liters/min in the mouse cage and followed by cervical dislocation when moribund or determined by a veterinarian in Animal Research Center at St Jude. For therapy studies in subcutaneous xenograft mouse models, the mice were euthanized through $CO_2$ inhalation with 3 liters/min in the mouse cage and followed by cervical dislocation when the tumor volume reached 2000 mm³ or mice became moribund.

### Generation of Alb-Cre;CAG-Myc (ABC-Myc) mice, and Alb-Cre; CAG-Myc;TdTomato (ABC-Myc; TdTomato) mice
Albumin-Cre (Alb-Cre) (Strain #003574), R26StopFLMYC (CAG-MYC) (Strain #020458), and CAG-tdTomato (Strain #007914) mice were obtained from the Jackson Laboratory. ABC-Myc (Alb-Cre⁺/ʷᵗ::CAG-MYC⁺/ʷᵗ) mouse model was generated by crossbreeding Alb-Cre⁺/⁺ with CAG-MYC⁺/⁺ mouse (age between 2 months to one year old), or Alb-Cre⁺/ʷᵗ with CAG-MYC⁺/⁺, or Alb-Cre⁺/ʷᵗ with CAG-MYC⁺/ʷᵗ. The littermates with genotypes of Alb-Creʷᵗ/ʷᵗ::CAG-Myc⁺/ʷᵗ, or Alb-Cre⁺/ʷᵗ::CAG-MYCʷᵗ/ʷᵗ, or Alb-Creʷᵗ/ʷᵗ::CAG-MYCʷᵗ/ʷᵗ were served as normal controls. In order to generate ABC-MYC;TdTomato mice, Alb-Cre⁺/⁺ mice were first bred with CAG-Tdtomato⁺/⁺ mice to obtain the mice with genotypes of Alb-cre⁺/ʷᵗ::CAG-tdTomato⁺/ʷᵗ, which were then bred with CAG-Myc⁺/⁺ mice. For genotyping, the genomic DNA was extracted from tail biopsies, and PCR amplification assay was performed using KAPA Mouse Genotyping Kits (Roche Corporate, Cat#KK7352) according to The Jackson Laboratory genotyping PCR conditions for each mice strain. The primers 5′-TGC AAA CAT CAC ATG CAC AC, GAA GCA GAA GCT TAG GAA GAT GG-3′ and 5′-TTG GCC CCT TAC CAT AAC TG-3′ were used for Alb-Cre genotyping (AlbCre = 390 bp and WT = 351 bp). The primers 5′-CCA AAG TCG CTC TGA GTT GTT ATC-3′, 5′-GAG CGG GAG AAA TGG ATA TG, CCA AGA GGG TCA AGT TGG A-3′ and 5′-GCA ATA TGG TGG AAA ATA AC-3′ are used for CAG-Myc genotyping (MYC = 550 bp and WT = 604 bp). The primers 5′-AAG GGA GCT GCA GTG GAG TA, CCG AAA ATC TGT GGG AAG TC-3′, 5′-CTG TTC CTG TAC GGC ATG G-3′ and 5′-GGC ATT AAA GCA GCG TAT CC-3′ were used for CAG-tdTomato genotyping (tdTomato = 196 bp and WT = 297 bp). The genotyping PCR products were resolved in 2% agarose gel (Invitrogen,

Cat#16500-500) and imaged with Alphaimager HP (ProteinSimple, Alphaimager HP) or Li-COR D-Digit (Li-COR, 3500).

## Generation of ABC-Myc-derived hepatoblastoma cell lines (NEJF1, NEJF2, NEJF4, NEJF5, NEJF6)

The livers from ABC-Myc mice were excised and placed in a sterile tube containing phosphate-buffered saline (PBS) on wet ice during transport from the animal research facility to the research laboratory. Tumor nodules were excised using a sterile scalpel and underwent an enzymatic digestion with collagenase IV (2 mg/ml; in 25 ml of DMEM medium) for 1 h in a 37 °C rotor (Robbins Scientific Corporation, model 2000). After digestion, cells were filtered using a 70-µm sterile strainer and cultured in ultra-nonadherent cell culture plate with DMEM medium with 10% FBS and 1% penicillin and streptomycin. Liver cancer cells form spheroids and are propagated in ultra-nonadherent cell culture plate. In parallel, the spheroids were transferred to adherent plates in standard DMEM media and adherent cell lines were derived. Notably, the adherent cell lines form spheroids when culturing in non-adherent cell culture plate.

## Generation of NEJF10 from the ABC-Myc;TdTomato tumor

The liver from ABC-Myc; TdTomato mouse (#NEJF10) was excised and placed in a sterile tube containing cold phosphate-buffered saline (PBS). Tumor nodules were excised using a sterile scalpel and underwent an enzymatic digestion with collagenase IV (2 mg/ml; in 25 ml of DMEM medium) for 1 h in a 37 °C rotor (Robbins Scientific Corporation, model 2000). After digestion, cells were filtered using a 70-µm sterile strainer and cultured in 15 cm culture dish (Fisher Scientific, FB012925) with DMEM medium with 10% FBS and 1% penicillin and streptomycin. Next day, the NEJF10 hepatoblastoma cells formed spheroids with tdTomato red color under EVOS M7000 Imaging System (Invitrogen, EVOS M7000). The spheroids were transferred and cultured in ultra-low attachment microplates for propagation (Corning, Cat#3471). The NEJF10 spheroids were transferred to adherent plates and the adherent NEJF10 cells were consequently derived. The adherent NEJF10 cells also form spheroids when cells were cultured back to ultra-low attachment microplates.

## Human cell lines, reagents, and validation

HepG2 (ATCC, HB-8065) cells were cultured in 1× DMEM (Fisher Scientific, Cat#MT10013CM) supplemented with 10% FBS (Gibco, Cat#10437028), 1% Penicillin-Streptomycin solution (Gibco, Cat#15140122)) at 37 °C in 5% $CO_2$ in a humidified incubator. All human-derived cell lines were validated by short tandem repeat (STR) profiling using PowerPlex® 16 HS System (Promega) once a month. Additionally, a polymerase chain reaction (PCR)-based method was used to screen for mycoplasma once a month employing the LookOut® Mycoplasma PCR Detection Kit (MP0035, Sigma-Aldrich) and JumpStart™ Taq DNA Polymerase (D9307, Sigma-Aldrich) to ensure cells were free of mycoplasma contamination.

Doxorubicin, NU7441, Samuraciclib (ICEC0942), and EPZ015666 (GSK3235025) were purchased from Selleckchem. All other compounds used for screening were obtained from St Jude compounds deposit. AZD7648, Lot01, was purchased from Chemietek, and the quality was verified by Chemieteck by HPLC-MS and NMR, with purity >99.5%. The purity of AZD7648 was further verified in house by using Waters UPLC-MS system (Acquity PDA detector, SQ detector and UPLC BEH-C18 column). The mass spectrometer was acquired using MassLynx v. 4.1. The chromatographic conditions are as follows: flow rate: 1.0 mL/min, sample injection volume: 2 µL, column temperature: 55 °C, mobile phase: 0.1% formic acid in $CH_3CN$ and $H_2O$.

## Pathological assessment of ABC-Myc hepatoblastomas

Liver tumors were fixed in 10% neutral buffered formalin, embedded in paraffin, sectioned at 4 µm, mounted on positive charged glass slides (Superfrost Plus; 12-550-15, Thermo Fisher Scientific, Waltham, MA) that were dried at 60 °C for 20 min, and stained with hematoxylin and eosin (HE). The following immunohistochemistry protocols were used for the detection of AFP, ARG1, Beta-catenin, GS, and KRT19, respectively, on commercial autostainers: (1) anti-Alpha-1-fetoprotein, A0008, Agilent, 1:300, 32-min incubation. Heat-induced epitope retrieval, Cell conditioning media 2 (Ventana Medical Systems, Tucson, AZ), 32 min; Visualization with DISCOVERY OmniMap anti-Rb HRP (760-4311; Ventana Medical Systems), DISCOVERY ChromoMap DAB kit (760-159; Ventana Medical Systems). (2) anti-Arginase-1 (H-52), sc-20150, Santa Cruz, 1:75, 60-min incubation. Heat-induced epitope retrieval, Cell conditioning media 2 (Ventana Medical Systems, Tucson, AZ), 48 min; Visualization with DISCOVERY OmniMap anti-Rb HRP (760-4311; Ventana Medical Systems), DISCOVERY ChromoMap DAB kit (760-159; Ventana Medical Systems). (3) anti-Beta-catenin (Clone E247), RM-2101, ThermoFisher, 1:300, 60-min incubation, Heat-induced epitope retrieval, Cell conditioning media 1 (Ventana Medical Systems, Tucson, AZ), 48 min. Visualization with DISCOVERY OmniMap anti-Rb HRP (760-4311; Ventana Medical Systems), DISCOVERY ChromoMap DAB kit (760-159; Ventana Medical Systems). (4) anti-Glutamine synthase, ab73593, Abcam, 1:1000, 60-min incubation, Heat-induced epitope retrieval, Cell conditioning media 1 (Ventana Medical Systems, Tucson, AZ), 32 min. Visualization with DISCOVERY OmniMap anti-Rb HRP (760-4311; Ventana Medical Systems), DISCOVERY ChromoMap DAB kit (760-159; Ventana Medical Systems). (5) anti-Keratin19, TROMA-III, Developmental Studies Hybridoma bank, 1:1000, 15-min incubation. Heat-induced epitope retrieval, Epitope Retrieval solution 1 (ER2), 20 min. Visualization with rabbit anti-rat (712-4126; Rockland), Bond Polymer Refine Detection (DS9800, Leica Biosystems). All HEs and IHCs were reviewed by light microscopy and interpreted by a board-certified veterinary pathologist (HT).

## Clinical chemistry analysis

The plasma and serum samples were obtained at sacrifice time when mice became moribund. Once whole blood samples for chemistries are received in the Diagnostic Lab at St Jude, they are allowed to clot for 30 min, at which point the clot is removed and the serum separated by centrifuging at 5700 rpm for 10 min. Once separated, the serum is pipetted into a Horiba bio cup for processing. Data is processed on the Horiba Pentra 400 instrument and uploaded via the RSAS app directly into an excel spreadsheet for further analysis. The ABX Pentra chemistry panel reagents including ABX Pentra Albumin CP (REF# A11A01664), ABX Pentra ALP CP (REF# A11A01626), ABX Pentra ALT CP (REF# A11A01627), ABX Pentra Amylase CP (REF# A11A01628), ABX Pentra Urea CP (REF# A11A01641), ABX Pentra Calcium AS CP (REF# A11A01954), ABX Pentra Creatinine 120 CP (REF# A11A01933), ABX Pentra Glucose HK CP (REF# A11A01667), ABX Pentra Phosphorus CP (REF# A11A01665), ABX Pentra Potassium-E (REF# A11A01740), ABX Pentra Sodium-E (REF# A11A01738), ABX Pentra Bilirubin, Total CP (REF# A11A01639), ABX Pentra Total Protein 100 CP (REF# A11A01932), according to the manufacturer's instructions.

## AFP quantification

The control and ABC-Myc mice were anesthetized with isoflurane. Blood samples from control and ABC-Myc tumor bearing mice were collected using the cardiac puncture method. To separate serum, blood in collection tubes was allowed to clot at room temperature without disturbing for 1 h and centrifuged at 1000 g for 20 min using a 4 °C refrigerated centrifuge. Serum was aliquoted in 55 µl or 110 µl per tube with the animal ID and date of collection. Samples were stored at −80 °C until all controls and tumor mice samples were complete. AFP measurement was performed by following the instructions of the AFP Elisa kit (Mouse Alpha-Fetoprotein ELISA Kit, MyBioSource, MBS033826). Briefly, 50 µl of standards (S1, S2, S3, S4, S5, S6) were added in standard wells and 50 µL of samples were added to every

sample well. 100 µl of HRP-Conjugate Reagent were mixed with each standard and sample wells. The plate was covered with closure plate membrane and incubated for 60 min at 37 °C. All wells were washed 4 times and 50 µl Chromogen Solution A was added to every well and gently mixed with 50 µl Chromogen Solution B and incubated for 15 min at 37 °C avoid from light. After adding 50 µl Stop Solution for 5 mins, the optical density (O.D.) at 450 nm were measured with Bio-Tek Synergy H1 Microplate Reader. Standard curve was plotted with GraphPad Prism 9.3.1. Based on the Standard curve to calculate the AFP value of each sample.

### Complete blood counting

Once EDTA anti-coagulated samples for CBC's are received in the lab, they are immediately organized by ID number and processed on the Oxford Science hematology analyzer. The results are automatically downloaded onto an excel spreadsheet, reviewed by lab personnel, and sent to the investigator via email for further analysis. In brief, blood was collected in Eppendorf tubes containing 10 µl of 10% EDTA via retroorbital bleed using 200 µl heparinized capillary tubes (Cat# 22-362-566, Fisher brand). Blood samples were processed within 2 h to avoid hemolysis. The number of leucocytes (WBC), erythrocytes (RBC), lymphocytes (LY), neutrophil (NE), monocytes (MO), eosinophils (EO) and platelets (PLT) were counted. Proprietary lysing agent was added to liberate hemoglobin and ultimately convert it to cyanmethemoglobin to calculate the value.

### Western blot and antibodies

For western blotting, samples from normal livers and tumors excised from livers were homogenized with calculated volume of 2× sample buffer (1 M TRIS/HCl, 10% SDS, 0.1% bromophenol-blue, 10% β−mer-captoethanol, 10% glycerol) and heated for 15 min at 95 °C. Proteins were resolved on protein gels (Bio-Rad, Cat#4568083) and transferred onto PVDF membrane (Bio-Rad, Cat#170-4272) with Trans-blot Turbo transfer system (Bio-Rad, Cat#1704150). After being incubated with the primary antibody, horseradish peroxidase-(HRP) conjugated secondary antibody (Novex, Life technologies) at 1: 5000 was used for 1 h incubation. The signals were detected by chemiluminescence (ECL, Thermo scientific). Images were taken with Li-COR Odyssey FC (Li-COR, Cat#2800). C-MYC (Cell Signaling Technology, 5605 S, RRID:AB_1903938, 1:1000), PRKDC (DNA-PK))Novus, sc57-08, RRID: AB_2809479, 1:1000), β-actin (Sigma, A5441, RRID:AB_476744, 1:5000) and GAPDH-HRP (Cell Signaling Technology, 3683 S, RRID:AB_1642205, 1:1000) were used for western blot.

### Small interfering RNA Transfection and doxorubicin treatment

Small interfering RNAs (siRNA) were transfected into NEJF10 cells using Lipofectamine RNAiMax (Invitrogen, Cat#13778150) according to manufacturer's instructions. Non-Targeting siRNA#2 (Thermo Fisher Scientific, AM4637) used as siRNA control. The siRNA oligos for Prkdc was ordered from Thermo Fisher Scientific (Thermo Fisher Scientific, AssayID151238, https://www.thermofisher.com/order/genome-database/details/sirna/151238?CID=&ICID=&subtype=, siRNA for mouse *Prkdc*:

Sense: 5-GGAAUAUACUAUAGAUCCUTT-3; Antisense: 5-AGGAUCUAUAGUAUAUUCCTG-3). Post 72 h of transfection, cells were harvested for western blot. For doxorubicin treatment experiments, 24 h post siPrkdc transfection, cells were treated with doxorubicin with concentrations of 0, 15 nM for 4 days and fixed with formaldehyde for crystal violet staining.

### Bulk RNA-seq and analysis

Total stranded RNA sequencing data were processed by the internal AutoMapper pipeline. Briefly the raw reads were first trimmed (Trim-Galore version 0.60), mapped to mouse genome assembly (GRCm38, mm10) (STAR v2.7) and then the gene level values were quantified

(RSEM v1.31) based on GENCODE annotation (VM22). Low count genes were removed from analysis using a CPM cutoff corresponding to a count of 10 reads in at least one sample group and only confidently annotated (level 1 and 2 gene annotation) and protein-coding genes are used for differential expression analysis. Normalization factors were generated using the TMM method, counts were then transformed using voom and transformed counts were analyzed using the lmFit and eBayes functions (R limma package version 3.42.2). The significantly up- and down- regulated genes were defined by at least 2-fold changes and adjusted *p*-value < 0.05. Then Gene set enrichment analysis (GSEA) was carried out using gene-level log2 fold changes from differential expression results against gene sets in the Molecular Signatures Database (MSigDB 6.2) (gsea2 version 2.2.3). GSEA parameters (number of permutations =1000, permutation type = gene_set, metric for ranking genes = Signal2Noise, Enrichment statistic = Weighted).

### Cross-species transcriptomic comparison of human hepatoblastoma and ABC-Myc tumors

A cross-species dataset of RNA-seq reads was obtained for all autosomal genes with a 1:1 homology between human and mouse, as determined Ensemble BioMart version 75. Sex chromosome genes were omitted to avoid gender effects. The resulting cross-species data consisted of 11,393 genes (common stable gene symbol in both datasets). The counts from Mouse and Human datasets were transformed using variance stabilizing transformation (VST) available from the DESeq2 R package version 1.16.1[134] for each dataset. To further account for differences in expression levels across species, the VST subject-regressed data were quantile-normalized using normalizeQuantiles from limma R package version 3.46.0[135]. Finally, we adjust the differences between species using ComBat function from sva package version 3.38.0.[136]. One thousand cycles of bootstrapping using pvclust R version 2.2.0[137] were done in tumor samples from Mouse and Human and plotted their Pearson correlation in a heatmap using pheatmap R version 1.0.12. We applied Principal Component Analysis on the full ortholog data matrix using prcomp (stats R package v.3.4.1) to compute PC1 and second PC2 principal components to illustrate the relative ordination of the integrated group samples and species. We plotted the results incorporating confidence ellipses at the groups by means of the ggbiplot R package (v-0.55). Differential expression between tumor and non-tumor for mouse and human was assessed using moderated t-statistics (limma R package. version 3.46.0). Benjamini–Hochberg (BH) procedure were employed to compute FDR. Proportional VENN diagrams were performed using the DeepVen tool (arXiv:2210.04597).

### Single-cell RNA-seq and analysis

**Library preparation and sequencing.** The liver tumor was harvested from ABC-Myc mouse. Tumor mass was dissociated by using a modified two-step collagenase procedure[138]. Briefly, the mouse was perfused with PBS containing 0.5 mM EDTA and followed by perfusion with 2 mg/ml of collagenase type IV (Worthington Biochemical Corporation, CLS-4) in DMEM (Dulbecco's Modified Eagle Medium) (Corning, 10-013-CM). The tumor from liver was chopped with razor and digested in 2 mg/ml of collagenase type IV DMEM medium for 30 mins at 37 °C. The cell suspension was filtered through a 70 µm strainer and washed twice with DMEM. The dissociated cells were suspended in the DMEM medium. Before loaded into Chromium chips, cells were filtered again through a 40 µm strainer and the single cells were counted by using a Luna cell counter, and then loaded into Chromium Chips V3 (10× Genomics) with a target capture of 8000 cells. The cDNA library construction and quality control were performed by following the manufacture's protocol. The library was sequenced in Novaseq-V1 reagents. The sequenced data were processed by Cell Ranger Software (10× Genomics).

**Data generation and preprocessing.** We collected cells from four tumor samples and three healthy control mouse samples. The single-cell RNA-seq samples were processed according to the 10× Genomics protocol. Cell Ranger Single-Cell Software Suite (version 6.0.0, 10× Genomics, using the option –include-introns) was used to quality control and quantify the single-cell expression data to generate filtered gene-barcode matrices for 90,715 cells with an average of 3037 mRNA molecules (UMIs, median: 1963, range: 302–32,765). Cells with low (≤500) UMI counts or more than 20% UMI counts from mitochondrial genes were considered low quality as proportion of mitochondrial genes may indicate the quality of cells[139]. With this quality control criterion, 9.3% of cells were considered low-quality (9.7% of low-quality cells from the control group and 9.2% of low-quality cells in the tumor group).

**Clustering.** The whole dataset's subpopulation structure was inferred using Latent Cellular State Analysis[140]. An optimal number of clusters was selected from top models determined by the silhouette measure for solutions with different clusters (from 2 to 100). Clusters with more than 50% low-quality cells were excluded from further analysis. The identities of the cell clusters were analyzed by inspecting the assignment scores using reference expression datasets with curated cell types[70] from the R/Bioconductor package (version 1.8.0)[69]. We also used an independent marker gene list from Song et al.[65]. The assignment score was calculated using the R/Bioconductor package SingleR (version 2.0.0)[69] based on the reference samples with the highest Spearman rank correlations, using only the marker genes between pairs of labels to focus on the relevant differences between cell types.

**Differential gene expression analysis.** We estimated differential expressions between tumor and control groups or among cell types using the negative binomial with independent dispersions[67], using edgeR (version 3.40.1)[141], with batch effect corrected using SVA (3.46.0)[142]. Specifically, we used scran (version 1.26.1)[143] to extract the normalization factor. For edgeR, we set prior.df to zero to independently infer each gene's dispersion based on scRNA-seq data and used the likelihood ratio-based test. For the application of SVA, we first sorted cells by total UMI within each batch and then summed 20 cells into a new aggregated pseudo-cell. Then SVA was applied with ten iterations to extract the top 20 surrogate variables representing the latent batch effects. The method was implemented in the function DEAdjustForBatch in the NBID package (https://bitbucket.org/Wenan/nbid/src/master/R/DEAdjustForBatch.R).

**Pathway analysis.** Gene set enrichment and pathway analysis were performed using the R package ClusterProfiler (version 4.4.4)[144]. We used Molecular Signature Database[51,145] for gene annotations through R package msigdbr (Dolgalev I, 2022, version 7.5.1) for gene annotation. We also used Database Annotation Visualization Integrated Discovery (DAVID 2021) for functional annotation analysis (version 2021)[146].

**Data visualization.** High-dimensional scRNA-seq data were visualized on two-dimensional maps through t-distributed stochastic neighbor embedding (t-SNE)[147,148] using R package Rtsne[149] with default settings. Dot plot and heatmap for gene expression patterns for cell types and gene/gene groups, and Sankey plot for the composition of inferred cell types and experimental samples were visualized using R package SCpubr[150] (version 1.0.4.9000). Cell assignment score heatmap was plotted using R package pheatmap[151] (version 1.0.12). General visualizations, including feature plots were performed using packages ggplot2[152] (version 3.4.0) and ggpubr[153] (version 0.5.0).

## Spatiotranscriptomic analysis

**Tissue harvest.** The ABC-Myc mouse was anesthetized with avertin (0.4 ml/20 g of mouse body weight). The mouse chest cavity was opened to expose the heart with needles, tweezers, and dissecting scissors. The right auricle was incised, and the needle filled with PBS containing 0.5 mM EDTA was immediately inserted into the apex of the left ventricle for the perfusion, followed by perfusion with 2 mg/ml of collagenase type IV (Worthington Biochemical Corporation, CLS-4) in Dulbecco's Modified Eagle Medium (Fisher Scientific, Cat#MT10013CM). The tumor tissue was immediately isolated in the ice cold DMEM medium.

**Pathological assessment.** Fresh frozen tissues were sectioned and mounted on the ST Library preparation slides, HE stained, and scanned with a Zeiss Axioscan slide scanner to generate 20× digital whole slide images. CZI files were imported into HALO (v3.2.1851.351, Indica Labs) to annotate and classify bulky tissue regions as neoplasia, non-neoplastic hepatocytes and stroma, extramedullary hematopoiesis, glass/clear space, or tissue folds/artifacts based on morphology and tinctorial staining characteristics.

**Tissue processing and data generation for spatial transcriptomics.** Flash frozen samples were embedded in OCT (Tissue-Tek, Sakura) and cryosectioned as per Tissue preparation guide from Visium Spatial Gene expression Kit- 10× Genomics (Cat.1000184). Briefly, the tumor tissue was harvested from ABC-MYC mouse liver in the ice cold DMEM medium (Fisher Scientific, Cat#MT10013CM), excess liquid was removed from tissue and flash frozen immediately in the bath of Isopentane and Liquid nitrogen. The OCT embedded tissue block was sectioned (10 μm) and placed on the capture area of Visium Gene Expression Slide and stored at −80 °C overnight. The tissue sections on the Visium slide were fixed with Methanol by incubating 30 min at −20 °C. The tissue was H&E stained following Visium Gene expression kit procedure. The H&E-stained sections were imaged using AxioScan Z.1. Whole slide scanner with standardized imaging protocol for Visium kit. After image acquisition, the slide sections were permeabilized for 18 min at 37 °C and cDNA, library was generated according to the Visium Spatial Gene Expression User Guide. The libraries were loaded and sequenced (R1-28cy, i7-10cy, i5-10cy and R2 −120 cycle) on Novaseq 6000 (Illumina) following recommendation of Visium Gene expression kit. The raw data were converted into FastQ, and matrices of expression generated using the Space Range software V1.0 provided by 10× Genomics.

**Spatial transcriptomics analysis and integration.** The gene-spot matrices were analyzed with the Seurat package[154] (versions 4.3.0) in R. Normalization across spots was performed with the SCTransform function[155] with regression of replicate and number of genes per spot. Dimensionality reduction and clustering were performed with principal component analysis (PCA) using the default setting in function RunPCA. We integrated the spatial data with scRNA-seq data by using the cell clusters inferred by LCA from scRNA-seq dataset as a reference. Spatial feature expression plots were generated using Seurat's SpatialFeaturePlot.

## Annexin V/DAPI staining

Cells were seeded at a density of 100, 000 cells in each well in 6 well plates. Next day, cells were pretreated with AZD7648 for 1 h, before adding doxorubicin for further 48 h. Cells were trypsinized (0.05% trypsin for NEJF10 and 0.25% trypsin for HepG2) for 4 min and centrifuged at 1000 rpm for 5 min at 4 °C. Apoptosis was detected by dual staining of Annexin V-FITC and DAPI using apoptosis assay kit (TONBO biosciences, CA, USA) according to manufacturer's instructions. Annexin V-FITC/DAPI positive cells were Collected using log amplification, and 10,000 events were recorded (BD LSR-II, BD Biosciences, NJ, USA), and data was analyzed using BD FACSDiva™ Software.

## Crystal violet staining

ABC-Myc-derived hepatoblastoma cell lines (750 cells per well) and HepG2 (10,000 cells per well) cells were seeded in 6-well plates. After 24 h, cells were treated with AZD7648 (0, 0.1, 0.33, 1 and 3.3 μM). Doxorubicin (0, 7.5, 15, 30 and 60 nM for NEJF10 cell line and 0, 5, 25, 125, and 625 nM for HepG2) was added post AZD7648 treatment 1 h. NEJF10 cell was cultured with DMEM complete medium for 5 days and HepG2 was cultured with DMEM complete medium for 8 days. The culture medium and AZD7648 and doxorubicin were changed every 2–3 days. After removing media, cells were washed with Dulbecco's phosphate buffered saline without calcium or magnesium (Lonza, Cat#17-516Q) and fixed with 4% formaldehyde in PBS for 20 min. Once formaldehyde was removed, cells were stained with 0.1% crystal violet (Sigma-Aldrich, Cat#HT90132-1L) for 1 h. Plates were rinsed with water and imaged.

## PrestoBlue assay and Bliss score calculation

PrestoBlue assay and Bliss score calculation was described as previous report (Alexandra et al., 2021) with minor modification. Briefly, NEJF10 (100 cells per well) and HepG2 (1000 cells per well) cells were seeded in 96-well plates. After 24 h, cells were treated with AZD7648 (0, 0.1, 0.33, 1 and 3.3 μM) and doxorubicin (0, 7.5, 15, 30 and 60 nM for NEJF10 cell line and 0, 5, 25, 125 and 625 nM for HepG2) in an 8 × 5 matrix. Cells were treated for 5 days, and cell viability was determined using the PrestoBlue assay (Invitrogen, A-13262) according to manufacturer's instructions. Cell viability for each treatment was normalized against the control group. A Bliss independence model was used to evaluate combination effects. Percentage over the Bliss score index was calculated with the equation (A + B)-AxB, in which A and B are the percentage of growth inhibitions induced by agents A and B at a given dose, respectively. The difference between the Bliss expectation and the observed growth inhibition induced by the combination of agent A and B at the same dose is the Bliss excess.

## Cell viability assay for IC50 of CDK7 and AURKA inhibitors

Cell lines were plated in 384 well plates at 100 (NEJF1, NEJF10), 500 (NEJF2, NEJF4, NEJF6) or 1000 (CCLF_PEDS_0046_N) and treated with either samuraciclib or alisertib in technical quadruplicate at doses ranging from 2 nM to 20 uM using a Tecan D300e compound printer (Tecan Biosciences). All wells were normalized to 0.1% total DMSO input. Cells were incubated at 37 °C until timepoint of assay development, using the Cell-Titer Glo assay (Promega). Data was processed using Graphpad Prism 7.0. Cell Lines: CCLF_PEDS_0046_N normal fibroblasts were a kind gift of the Cancer Cell Line Factory (Broad Institute, Cambridge, MA).

## CRISPR screening for cancer dependency gene and genetic modifiers to doxorubicin

The Mouse CRISPR Knockout Pooled Library (Brie, lentiCRISPRv2) was obtained from Addgene (Addgene#73632), which includes 1000 control gRNAs and 78,637 gRNAs targeting 19,674 genes. The plasmid library was amplified and validated in the Center for Advanced Genome Engineering at St. Jude Children's Research Hospital as described in the Broad GPP protocol (https://portals.broadinstitute.org/gpp/public/resources/protocols) except Endura™ DUOs (Lucigen) electrocompetent cells were used for the transformation step. The workflow of this whole genome genetic screen is illustrated in Fig. 6a. We used NEJF1, NEJF6, and NEJF-10 cells; three mouse hepatoblastoma cell lines established in our laboratory by culturing dissociated liver mass cells from the ABC-Myc models. The cells were transduced with mouse CRISPR Knockout pooled library (Brie) which contains 78, 637 unique sgRNA sequences targeting 19,674 human genes (4 sgRNAs per gene, and 1000 non-targeting controls) at a low MOI (-0.3) to ensure effective barcoding of individual cells. Cells were replenished with fresh DMEM medium containing 2 μg/mL puromycin (Millipore Sigma) for

36 h. After puromycin selection, cells were washed to eliminate dead cell debris and maintained in complete DMEM medium, and $32 \times 10^6$ cells were collected for genomic DNA extraction to ensure over 400× coverage of Brie library. For genetic mapping, the transduced cells were cultured for 5 days for CRISPR editing to generate a mutant cell pool, which was then treated with vehicle (DMSO) and doxorubicin (IC20 - 5 nM, 14 days: IC90 - 30 nM 21 days), these concentrations were selected from colony formation assay that mimics similar experimental setup for actual experiment (IC20 - 11.84 nM, IC90 - 35.83 nM, for 4 days), since doxorubicin has a very narrow therapeutic window at the nM level. During the experiment, at least $32 \times 10^6$ cells were collected for genomic DNA extraction to ensure over 400× coverage of Brie library. The total genomic DNA was extracted using a DNeasy Blood & Tissue Kit (Qiagen) and quantified with a Nanodrop instrument. The sgRNA sequences were amplified using PCR method using NEB Q5 polymerase (New England Biolabs). PCR products were purified by AMPure XP SPRI beads (Beckman Coulter) and quantified by a Qubit dsDNA HS assay (Thermo Fisher Scientific). A total of 16 million reads were sequenced using an Illumina HiSeq sequencer, and the sequencing data were analyzed using MAGeCK-VISPR software. NGS sequencing was performed in the Hartwell Center Genome Sequencing Facility at St. Jude Children's Research Hospital. Single-end, 100-cycle sequencing was performed on a NovaSeq 6000 (Illumina). Validation to check gRNA presence and representation was performed using calc_auc_v1.1.py (https://github.com/mhegde/) and count_spacers.py. Network analysis performed using STRING program (https://string-db.org/).

## Drug response screen

For the screen, assay-ready plates were prepared by dispensing 50 nl small molecules in empty white 384-well plates (Corning) using Echo 555 Liquid Handler (Labcyte). A total of 50 μl ABC-Myc cells per well were seeded in the assay-ready plates. The cells were incubated at 37 °C, 5% $CO_2$ in a humidified cell culture incubator (LiCONiC) for five days. Prior to the cytotoxicity assay, 25 μl medium per well was removed by Apricot S2 (SPT Labtech). To quantify the cytotoxicity, the amount of intracellular ATP was measured by CellTiter Glo (Promega). Widget, an automated robot system in St. Jude Children's Research Hospital was utilized for the cytotoxicity assay. A total of 25 μl CellTiter Glo reagent (Promega) was added to each well by Multidrop Combi (ThermoFisher). After shaking plates, plates were incubated for 20 min at room temperature. Then, the luminescent signal was measured by EnVision (PerkinElmer). Luminescent signal results were analyzed by Genedata Screener (Genedata). All the results were normalized by the negative control (DMSO) and the positive control (5 μM 17-DMAG).

## In vivo therapy

(**1**) *Transgenic ABC-Myc mice mouse model*: All the animals are procured at Animal Resource Center (ARC) at St. Jude Children's Research Hospital and study was approved by Institutional Animal Care and Use Committee. Following genotyping, ABC-Myc mice were randomized and assigned to treatment groups. Inclusion criteria were the presence of the ABC-Myc allele, either in heterozygosity or both ABC-Myc alleles, age ranging 16–18 days after birth, both genders. Mice were treated with vehicle, doxorubicin (0.75 mg/kg, intraperitoneal, twice weekly) and AZD7648 (50 mg/kg/day, twice, oral gavage, everyday); either agent alone or in combination with doxorubicin and AZD7648 for 3 weeks. The mice weight and activity were monitored throughout the experiment. The humane end point was decided (notified by staff not directly involved in this study) to euthanize the mice. The livers from treatment groups of ABC-Myc mice and age matching normal mice were excised, weighed, and imaged. (**2** *HepG2 xenograft study*: 4–6-week-old female NSG mice (NOD.Cg-Prkdc scid Il2rg tm1Wjl /SzJ) were housed in pathogen-free conditions with food and water

provided ad libitum. HepG2 cells ($5 \times 10^6$/mouse) in 100 µl PBS were injected subcutaneously on the right flank of mice. When the tumor size reached up to ~100 mm³, the animals were randomized into four groups ($n = 5$ mice per group). Mice were treated with vehicle, doxorubicin (1 mg/kg, intraperitoneal, twice weekly) and AZD7648 (50 mg/kg/day, oral gavage everyday); either agent alone or in combination with doxorubicin and AZD7648 for three weeks. The tumor volume and mice weight were measured twice in a week. The volumes calculated with formula $\pi/6 \times d^3$, where d is the mean of two diameters taken at right angles. All the mice were euthanized, and subcutaneous tumors were collected, imaged, and weighed. The tumor volume and weight were presented as the means ± S.D ($n = 5$). (3) *AZD7648 and doxorubicin effects in SJHB031109_X1 PDX model*: To establish SJHB031109_X1-PDX model, around 5 weeks old female NSG mice (NOD.Cg-Prkdc scid Il2rg tm1Wjl /SzJ) were purchased from St Jude Children's Research Hospital Animal Research Resource and housed in pathogen-free conditions with food and water provided ad libitum. PDX SJHB031109_X1 tumor was finely minced with sterile scissors and blade in a sterile petri dish. Tumor tissue ~50 µl was subcutaneously engrafted on the right flank of NSG mice. When the tumor size reached up to ~100–200 mm³, the animals were randomized into four groups ($n = 4–5$ mice per group). Mice were treated with vehicle (HPMC/T), doxorubicin (0.75 mg/kg, intraperitoneal, twice weekly) and PRKDCi (AZD7648) (50 mg/kg, twice/day, oral gavage every day and the time between the morning and evening doses was 8 h); either agent alone or in combination with doxorubicin and AZD7648 for two weeks. In the day of doxorubicin, doxorubicin was dosed 1 h after the morning AZD7648 treatment. All the mice were euthanized, and subcutaneous tumors were collected.

In vivo studies were approved and conducted in accordance with Institutional Animal Care and Use Committee at St. Jude Children's Research Hospital.

## Statistical analysis and reproducibility

All measurements were biological replicate samples except tumor heterogeneity studies in that individual tumor was applied. Data are presented as mean ± SD from at least three biological replicates unless otherwise stated. In general, for two experimental comparisons, unpaired two-tailed Student's t-test was used. Kaplan-Meier survival curves were statistically compared by the log-rank test. Statistical significance is represented by asterisks corresponding to **$p < 0.05$ but actual $P$ values provided, ***$p < 0.0001$, and ****$p < 0.0001$. GraphPad Prism software (version 9.0) was used to generate graphs and perform statistical analyses. Enrichr program (https://maayanlab.cloud/Enrichr/) was used for gene ontology analysis.

## Reporting summary

Further information on research design is available in the Nature Portfolio Reporting Summary linked to this article.

## Data availability

The bulk RNA-seq data generated in this study have been deposited to GEO database with accession number: GSE193124. The scRNA-seq and spatiotranscriptomics data generated in this study have been deposited to SRA database with accession numbers: GSE223689, GSE194051, GSE195575. The publicly available datasets re-used in this study included GSE79084[29], GSE94858[53], GSE133039[15]. GSE131329[52], https://www.sciencerepository.org/gene-expression-profiling-in-hepatoblastoma-cases-of-the-japanese-study-group-for-pediatric-liver-tumors-2-jplt-2-trial_EJMC-2018-1-103]. All other data, including original western blot and statistical data generated in this study, are released in the Source Data file. Source data are provided with this paper. Academic researchers can request ABC-MYC cell lines without limitations once MTA is signed. Source data are provided with this paper.

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

## Acknowledgements

We thank the staff of the St. Jude Animal Resource Center, Hartwell Center and Applied Center for Bioinformatics for their dedication and expertize. We thank Asa Karlstrom and Developmental Biology and Solid Tumor Program at St Jude for providing PDX models. The work was supported by American Cancer Society-Research Scholar (130421-RSG-17-071-01-TBG, J.Y.), National Cancer Institute (1R01CA229739-01, J.Y., R01CA266600-01A1, J.Y.). The content is solely the responsibility of the authors and does not necessarily represent the official views of the National Institutes of Health. C.A. was partially supported by the Scientific Foundation of Spanish Association Against Cancer (ref PRYCO223102ARME) and the Fight Kids Cancer Funding Programme (BT4ChildLC), by Imagine for Margo, Foundation Kidcancer, Foundation Kriibskrank Kanner, Federazione italiana Associazioni Genitori e Guariti Oncoematologia Pediatrica, Cris Cancer Foundatinon.

## Author contributions

J.Y.: Conceptualization, data curation, formal analysis, supervision, validation, investigation, visualization, methodology, writing–original draft, writing–review and editing. X.C.: Conceptualization, data curation, formal analysis, supervision, validation, investigation, visualization, methodology, writing–review and editing. J.F., S.S., S.N., H.S., A.D., H.W.L., C.C., W.C., H.J., W.C., J.R., C.A., J.E., S.K., T.S.: Data curation, formal analysis, validation, investigation, visualization, methodology, writing–review and editing. A.A.-Z., J.S., J.C., S.C.: Formal analysis, methodology, writing–review and editing. Q.W.: validation, methodology. Y.F., S.P.-M., T.C., J.E.: Supervision, methodology, writing–review, and editing. R.W., E.G., A.M.: Conceptualization, writing–review, and editing. A.D.: Conceptualization, supervision, writing–review, and editing.

## Competing interests

The authors declare no competing interests.

## Additional information

[1]Department of Surgery, St. Jude Children's Research Hospital, Memphis, TN, USA. [2]Department of Computational Biology, St. Jude Children's Research Hospital, Memphis, TN, USA. [3]Comparative Pathology Core, St. Jude Children's Research Hospital, Memphis, TN, USA. [4]Department of Pathology, St. Jude Children's Research Hospital, Memphis, TN, USA. [5]Division of Molecular Oncology, Department of Oncology, St. Jude Children's Research Hospital, Memphis, TN, USA. [6]Department of Chemical Biology and Therapeutics, St. Jude Children's Research Hospital, Memphis, TN, USA. [7]Center for Advanced Genome Engineering (CAGE), St. Jude Children's Research Hospital, Memphis, TN, USA. [8]Center for Applied Bioinformatics, St. Jude Children's Research Hospital, Memphis, TN, USA. [9]VPC Diagnostic Laboratory, St. Jude Children's Research Hospital, Memphis, TN, USA. [10]Champions Oncology, 1330 Piccard dr, Rockville, MD, USA. [11]Center for Childhood Cancer and Blood Disease, Hematology/Oncology & BMT, Abigail Wexner Research Institute, Nationwide Children's Hospital, Columbus, OH, USA. [12]Department of Surgery, College of Medicine, The University of Tennessee Health Science Center, 910 Madison Ave., Suite 325, Memphis, TN, USA. [13]St Jude Graduate School of Biomedical Sciences, St Jude Children's Research Hospital, Memphis, TN, USA. [14]Department of Pathology, College of Medicine, The University of Tennessee Health Science Center, Memphis, TN, USA. [15]Childhood Liver Oncology Group, Germans Trias i Pujol Research Institute (IGTP), Translational Program in Cancer Research (CARE), Badalona, Spain. [16]CIBER, Hepatic and Digestive Diseases, Barcelona, Spain. [17]CIBERehd, Madrid, Spain. [18]These authors contributed equally: Jie Fang, Shivendra Singh, Changde Cheng, Sivaraman Natarajan, Heather Sheppard. ✉e-mail: Xiang.Chen@stjude.org; Jun.Yang2@stjude.org

