## [Peer Review File · Nature Communications]

Genome-wide mapping of cancer dependency genes and genetic modifiers of chemotherapy in high-risk hepatoblastomaReviewers' Comments:

Reviewer #1:

Remarks to the Author:

In this manuscript, Fang et al developed a liver-specific MYC-driven hepatoblastoma (HB) murine model (ABC-Myc mice) that drives rapid hepatic oncogenesis. Authors showed that pathological and transcriptomic analyses reveal traits and signaling pathways characteristics of human HB. Another interesting aspect of this manuscript is the CRISPRCas9 screening to identify oncogenes/tumor suppressor genes in HB as well as genes that synergizes/antagonizes with doxorubicin. Specifically, they found that PRKDC inhibition enhanced the efficacy of doxorubicin chemotherapy, improving its anticancer efficacy in vitro and in vivo using the ABC-Myc and human HepG2 models.

This tumor model seems to be an exceptional tool for future studies in the HB field, a, extremely rare disease from which few tumor models have been reported (PMIDs: 27734029, 24837480, 24848510). Despite the relevance and the useful opportunities of this disease models and the knowledge generated about the HB biology using the CRISPCas9 screening of ABC-Myc cell lines with and without doxorubicin, the paper needs to be improved. In that regard, certain points still need to be clarified, some analysis/figures (specifically those related to Hba-a1 and triptofan pathway) need to be eliminated or moved to supplementary data, some analysis need to be improved and additional experiments should be performed to get robust conclusions. In addition, the figure legends lack a lot of information and difficult the reading of the manuscript. Please find below a summary of the major and minor comments to the Fang et al manuscript.

Major comments

- Nomenclature of the ABC-Myc tumors. The authors along the paper stated that their model recapitulate an HB. However, due to the fact that these tumors don't harbor Wnt/beta-catenin pathway activation, a key hallmark constitutively activated in nearly all human HBs, it would be more appropriate to refer the ABC-Myc tumors as "HB-like tumors" along all manuscript.

- Histopathological analysis: HB is a complex disease at pathological level with usually a mixture of tumors cells with different degrees of hepatic differentiation ranging from highly pleomorphic HB to tumors cells resembling embryonal or fetal hepatoblasts (Lopez-Terrada et al, Modern pathology, 2014). In that regard, authors pointed out that the pathologist found embryonal and fetal epithelial HB components. In that regard, the histological study of tumor lesions in the ABC-MYC model needs to be deeply studied. Specifically: how is the heterogeneity among the different tumor nodules of the same mice? Are there any pre-neoplastic lesions? Which is the main epithelial component of the lesions? It is important to remark that the main epithelial component has been associated to HB subtypes and patient outcome (Cairo et al, Cancer Cell, 2008). How are the tumors of different mice? Is there any mesenchymal component? Authors also mentioned that a "648 board-certified veterinary pathologist (HT)" (lines 648-649) reviewed the HE slides of 4-8 models. In that regard, the pathological study should be performed by an expert pathologist of human HB to describe better the pathology of these tumors.

- Immunostaining of ABC-MYC tumors: Authors showed in figure 1E-J the images of some immunohistochemistry (IHC) markers in the tumors. In relation to the above comment, authors should have quantified the staining of the different IHCs to provide a better idea about the heterogeneity/homogeneity of the tumor nodules of the different mice. Authors should also provide a non-tumor liver tissue as a control of the IHC. To confirm the similarities of the ABC-MYC tumors with aggressive HBs, it will be necessary that authors complete the IHC analysis with stem cell markers (i.e. EpCAM, GPC3, etc) as well as with proliferating markers (i.e. Ki67, PCNA). Authors should also justify why ABC-MYC tumors have strong expression of GLUL (marker of C1 tumors) and low expression of AFP (marker of aggressive C2 tumors); as shown by Cairo et al (Cancer Cell, 2008). Finally, Supplementary Figure S1 c-H seems to provide identical information that Figure 1E-J. Authors should delete it or specify which are the differences.

- Wnt/beta-catenin activation in the ABC-Myc model. It is surprising that the ABC-Myc mice generate tumors without activating Wnt/beta-catenin pathway, a key hallmark of HB (>80% of the human tumors have point mutations or deletions of exon 3 of the CTNNB1 gene). In relation to beta-catenin, authors mentioned a "lack of strong nuclear translocation of the wild-type beta-catenin" (line 105-106). However, in the figure 1, it is difficult to appreciate nuclear staining instead there is a marked staining of CTNNB1 in the membrane (localization of beta-catenin when Wnt signaling is not active). In addition, the expression of Wnt/beta-catenin GSEA analysis is not highly convincing (Figure 2D) because the geneset used "FEVR CTNNB1 TARGETS DN" came from intestinal crypt cells upon deletion of CTNNB1. Because beta-catenin is activating different target genes depending on the cell type and differentiation cell degree, the study of beta-catenin target genes should be improved by using a gene set of beta-catenin target genes induced in liver hepatocytes/hepatoblasts.

-Biochemical analysis of plasma from ABC-Myc mice harboring HB-like tumors. Authors performed a comprehensive clinical chemistry analysis of serum from ABC-Myc. The results showed a clear liver damage with an increase levels of hepatic enzymes and renal dysfunction (at what day are obtained the plasma sample? At sacrifice? Please specify at M&M). These results are probably explained to the advanced stage of the tumors and not to the fact that they are harboring an HB-like tumor. Please specify this. In addition, since alpha-fetoprotein (AFP) is a classical marker of HB in patients, authors should complement the biochemical analysis of the plasma of this model by measuring the levels of AFP in ABC-Myc mice vs. controls.

-GSEA analysis (figures 2B, C, D and F): This analysis has major drawbacks. For one side, the authors stated that figures 2B-D with "GSEA showing gene signatures or signaling pathways highly upregulated in ABC-Myc hepatoblastoma". The main problem is that these signatures came from adult hepatocellular carcinoma or intestinal crypts. In that regard, authors should use geneset signatures from gene expression profiling studies of human HB. For another side, authors should specify in the figure legend if the p-value is the FDR (only FDR < 0.25 should be considered significant) as well as include the normalized enrichment score (NES). Moreover, in the M&M section, due to the limited number of samples included in the GSEA analysis and authors should provide details about the approach used (i.e. number of permutations, describe the enrichment statistic used and the metrics for gene ranking)

- Characterization of the degree of tumor immaturity. Authors mentioned (lines 171-173) that "Consistent with the embryonal histological features, GSEA results showed that ABC-Myc tumors had significant upregulation of cancer stem cell signatures including "liver cancer with upregulated EpCAM" (Figure 2F), and "undifferentiated cancer" (Figure 2G)." However, to complete the study of the degree of tumor immaturity it would be interesting to show the expression of stem cell or liver progenitor markers that are characteristic of aggressive HBs. In addition, the increased expression of stem cell markers will provide further support of the aggressive phenotype of ABC-Myc tumors.

- Gene expression profiling (figures 2E, G, H and I). Figure 2E is a heatmap showing the top differentially expressed genes in ABC-Myc tumors vs. age-matched normal livers and authors concluded that these genes found dysregulated in the ABC-Myc model are similar to the ones altered in human HB tumors. This analysis would be improved if the authors provide the % of the overlapping genes altered in human HB vs non-tumors and MYC-ABC tumors vs control livers at a specific FDR. In addition, authors showed that ABC-Myc tumors had C2 and progenitor and proliferation signatures of high-risk HB. However, the control sample of gene expression profiling used for the heatmap is "normal" (non-tumor tissue). Accordingly, we cannot assure that C1 tumors could also share some C2 tumor profiling since as tumors they are "per se" also more immature and more proliferative than non-tumor tissue. A possibility to confirm the C2 nature of the ABC-Myc tumors, authors should integrate the expression profiling of human and mice tumors and study its aggrupation in an unsupervised heatmap.

- Metabolic pathways dysregulation. This part of the study is not relevant, the analysis performed are poor and in somehow biased to the Tryptofan pathway that, as so many pathways involved in the liver function, is down-regulated in HB. In that regard, the message is confusing and contributes to disperse the main message of the manuscript. In addition, authors stated that the "tryptophan catabolism is one of the most significantly altered metabolic pathways in hepatoblastoma (...)" (from line 209), but there is no strong data on this sentence in the literature and authors neither provide the specific references. Therefore, to improve the quality of the manuscript, authors should delete this section in the manuscript and the corresponding figure or move it to supplementary data including an additional analysis of other liver specific pathways (not just tryptophan pathway).

-scRNA and spatial transcriptomics. In general, the analysis of scRNA is very poor for several reasons: 1) Authors should include a healthy mouse control liver (scRNAseq and spatial transcriptomics) in order to see differences in cell populations in comparison with the ABC-Myc tumor. In that regard, they cannot mention that "tumor associated macrophages were abundant, suggesting that the ABC-Myc environment is immunosuppressive" (lines 278) because there is no control tissue analyzed; 2) authors did not identify the tumor cell populations. They just were focused on the stromal (Suppl. Fig 4) and non-tumor (Figure 4G) cell populations and they did not characterize the heterogeneity of cell tumor content. To note that pathologically, human HB but also ABC-Myc tumors exhibit embryonal and fetal tumor cells... are these distinct populations found in scRNAseq? A comparison of spatial transcriptomics data with pathological review of the tumors would be needed. Accordingly, only AFP has been studied as a tumor marker (when its expression is not very strong by IHC) and authors should identify the tumor cell populations by looking at other tumor markers such as GS (this is very important because it has been demonstrated by IHC that tumor cells express high levels of glutamine synthetase), DLK1, GPC3, KRT19 or EpCAM among others. 3) Apparently, there is an incongruence among scRNAseq and spatial transcriptomics and the expression of Hba-a1 and Afp. In that regard, the number of cells expressing Hba-a1 is very high in Figure 4B but very low in Figure 4D and viceversa for Afp. Authors should provide an explanation. 4) Authors indicate that one subpopulation of cells has a high levels of hemoglobin cells (line 262-263) but in Figure 4B we can see that at least 4 subpopulations of cells have high levels of hemoglobin cells. Which type of cells are? In addition to hemoglobin genes, which genes do these subpopulations overexpress? This is a very important issue taking into account that HB is characterized by having extramedular hematopoiesis, we could think that these populations are precursors of blood cells. Authors studied the expression of Hba-a1 using HB RNAseq datasets but it would be also important to compare with RNAseq databases of hematopoiesis or red blood precursors. 5) The fact that there is a negative correlation between Afp-Hba is not consistent with the fact that it has been reported that high-risk patients have high Hba-a1 expression according to Sekiguchi et al (lines 266-267). In that regard, it has been clinically demonstrated that high levels of AFP are associated with a higher risk patients and high levels of AFP in plasma are included in the current clinical stratification CHIC-HS as a marker of poor prognosis (Meyer et al, Lancet Oncol, 2017). In addition, Figure 4F is not informative and should be deleted. The same for the sentence: "Although not statistically significant, the expression between AFP and HBD (one of the adult human globin genes) tended to be negatively correlated in human hepatoblastoma tumors (Figure 4F), similar to the spatial expression pattern of Afp and Hba-a1 in ABC-Myc tumors" (lines 267-270).

- Genome-wide CRISPR screen: This part of the study is very interesting and the results provided including essential and tumor suppressive genes are in agreement with the literature. However, the figure 5D is not easy to understand. In that regard, authors should provide more details about the lists/genesets used for this analysis (i.e. hepatoblastoma C2 subclass: what this include? The 8 C2-specific genes of the 16-gene signature reported in Cairo et al (Cancer Cell, 2008)? The chapter "Genome-wide screen of cancer dependency genes in an ABC-Myc-derived hepatoblastoma" should be integrated with the "Murine and human hepatoblastoma share common essential and targetable genes" since the later should be considered as a finding validation in human HB cells of the CRISPRCas9 experiment in the murine cancer cell line (Figures 5G and H) and the results are all included in the same figure 5.

- Huh6 gene mutations: authors should correct the following mistake when mentioned that "HuH6 bears TP53 and AXIN1 mutations (depmap.org)" because Huh6 has Gly34Val CTNNB1 mutation instead of AXIN1 mutation as it is specified in the depmap.org website.
- Chemotherapy response. The authors explained that they treated the cells with doxorubicin at two doses (5nM and 30nM) for 7 and 14 days. However, figure S7 did not justify this choice. Please, justify. In addition, the authors should include the information of the axes in the plot of figure S6B and split the X-axis in 2 parts (0-100 nM and >100nM) to appreciate better the results on colony intensity.
- PRKDC inhibition enhances efficacy of chemotherapy. This part of the study is highly interesting since authors demonstrated the synergistic effect of PRKDC (a gene found in their screening) to sensitize tumor cells derived from ABC-Myc mice to doxorubicin. Authors validate the study in vivo and in vitro but to get robust conclusions, they need to complement it with the use of another PRKDC inhibitor and an at least additional human HB cell line (i.e. Huh6 previously used by the authors).
- Statistical analysis in the "Genetic and Chemical Perturbation database" should be checked because the log10FD is too high ranging from 5 to 200. The CRISPRCas9 screening for positive selection genes is not robust if we take into account the FDR; authors should justify this.
- All Figure legends have very limited information that difficult the comprehension of the results. Therefore, authors should expand them to provide much more information such as number of samples, statistical methods used, GEO code, database references, meaning of color codes, etc

Minor comments

- Figure 1C. Authors should specify in the figure the number of mice and specify the time units (days?)
- Figure 1D. There is a lack of control of MYC expression in P5 and P10. Please add or justify why it is not necessary.
- Figures 2H-I and 3F: Authors should add a color scale
- Supplementary tables should be numbered according to the text.
- Line 172: typo error "caner" instead of "cancer"
- Fig. 3A: Why some genes are highlighted in green?
- Fig. 3B: Why are some genes highlighted with a red arrow?
- Figure 5D: not clear the legend because the FDR is not indicated.
- All heatmaps should include color score index bar.
- Figure 4D, 4E and S4B: the colors are not clear (the quality of the images are too low).
- HB datasets: authors use different datasets in different analysis (i.e. Hooks et al for the study of Hba-a1 in HB, Carrillo-Reixach et al and Sumazin et al for the triptofan pathway study). Authors should use all datasets to validate the different findings.
- Please change "NEGLOG10FD" for in Suppl Figure 5 for $-\text{Log}_{10}(\text{FDR})$ as it is mentioned in Figure 5
- Figure 5G: authors should delete "Figure 5"
- Figure 6: delete the 2 sentences "Network analysis performed using STRING program (<https://string-db.org/>)" and include this in M&M
- Figure 7: order the panels in a homogenous way in all paper, especially in this figure.

Reviewer #2:

Remarks to the Author:

Fang and colleagues describe studies using a mouse model of hepatoblastoma, aiming at identifying new therapeutic targets. The authors describe (i) the characterization of the mouse model, which is based on hepatocyte-specific expression of c-Myc, (ii) molecular and histopathologic characterization of tumors, thereby showing that the model predominantly reflects high-risk human medulloblastoma, (iii) derivation of mouse hepatoblastoma cell lines and their use for genome-wide CRISPR/Cas9

screens to identify new vulnerabilities, (iv) genetic screens to identify modifiers of standard chemotherapy and (v) validation of one such combinatorial therapy in vitro/in vivo using mouse and human hepatoblastoma models.

High-risk hepatoblastoma is not only incompletely understood at the molecular level, but there is also an urgent need for new therapies. This study contributes some valuable cellular resources, gives some new insights into the genetic dependencies and also validates a potential new therapeutic approach. I do have however several major concerns, including the limited novelty of some parts of the work, the insufficient depth of analyses in other parts or overstatements stemming from .

1. The mouse model. Previous work reported Myc-induced aggressive/poorly differentiated hepatoblastomas in mice. This includes the dox-inducible Tet-o-Myc/LAP-tTA model (Goga et al, Nat med 2007 Jul;13(7):820; characterized in detail in comparison to the bCat-mutant/Myc mice by Comerford et al, JCI Insight. 2016 Oct 6;1(16):e88549.) or models based on hepatic Myc delivery by hydrodynamic tail vein injection (e.g. Mol Ther 2022 Apr 6;30(4):1645). Thus, although the Alb-Cre;LSL-Myc model reported here is a useful model for childhood hepatoblastoma, the conceptual advance is relatively limited.

2. The CRISPR screens have been performed in one mouse hepatoblastoma cell line only. All conclusions are drawn from this line. A similar screen has already been performed in the HuH6 human hepatoblastoma line (ref 68), and those human data are now being used by the authors to support findings in mice. Whilst this comparative approach is valid, the screen performed by the authors adds only limited new information to the existing human data. Screening much larger cell numbers would permit corroboration of findings and enable further analyses, such as mapping of genetic context-dependencies.

3. Mapping of synergies with chemotherapy. Identification of positive and negative selection trends specific for the doxorubicin-treatment setting requires more scrutiny, confirmation in multiple cell lines and experiments, as well as rigorous genetic and pharmacologic validation in separate experiments (e.g. using competition assays). In their current form, the analyses lack required depth.

4. Several claims made in the manuscript are not supported adequately by data. For example, "mapping of oncogenic pathways" is overstated, given that that (i) cancer genomes have not been characterized and that (ii) the functional studies to map dependencies are derived from one cell line only.

Reviewer #3:

Remarks to the Author:

Fang, Singh, Natarajan, Tillman et al present a study involving many different technologies spanning bulk RNA-seq, scRNA-seq, spatially-resolved transcriptomics (Visium by 10x Genomics), as well as CRISPR-Cas9, that were used to investigate the pathogenesis of hepatoblastoma. The authors link mouse in vivo experiments with human cell lines to find similarities between them in order to identify potential therapeutic targets in experimental conditions that could later be evaluated in clinical trials. Ultimately, this project is a great proof of concept and future studies could expand it with larger sample sizes.

While I'm not an expert in hepatoblastoma, I was asked to help review this manuscript given my experience with the different RNA-seq technologies. Overall, my impression is that this project and manuscript is well written. However, there is still some room for improvement as some methods are only superficially described and the justification for some thresholds or analytical decisions is omitted in this manuscript.

Here are some main comments:

* Am I understanding correctly that only 1 sample with 11,696 cells was used to generate the scRNA-

seq data? That seems to be the case looking at

<https://www.ncbi.nlm.nih.gov/geo/query/acc.cgi?acc=GSE194051>. I would emphasize this as a limitation in the discussion to warn readers about over interpreting the results.

* The methods for the scRNA-seq analysis are incomplete as right now it would be very challenging to try to reproduce the results with the provided information. Also, it seems like the authors are either choosing some arbitrary thresholds or they are using thresholds that they found useful for other datasets but that they didn't describe in the methods for this manuscript.

* The three GEO accession links don't show any FASTQ files. Is this a limitation of the reviewer visibility settings or are these files missing? The SRA links don't work for me right now.

<https://www.ncbi.nlm.nih.gov/Traces/study/?acc=PRJNA413799> from

<https://www.ncbi.nlm.nih.gov/geo/query/acc.cgi?acc=GSE104766>, which is used in Figure 4E does work.

* GSE195575 contains 4 Visium samples, but why is only 1 of them shown in the main and supplementary figures?

* Some Excel files don't have named Sheets. Some have colors that are not described like Table S7, Table S8, "IC20 gene (negative)" (what does the blue font mean for some rows?). None of the tables explain what each column is nor do they include a caption.

* I did not find a link to the code used for this project. You should share your code on a permanent repository like Zenodo or Figshare (both give you a DOI you can cite) or another option. You could also share it via GitHub, and well, GitHub repositories can easily be linked to Zenodo or Figshare. Code is the ultimate documentation on how an analysis was carried out and it's invaluable in order to reproduce an analysis.

Here are some smaller comments and requests:

* Line 203: cite KEGG

* Line 291: cite "Genetic and Chemical Perturbation database"

* Line 293: cite Biocarta

* Would it make sense to include a supplementary file with an example excel spreadsheet that was downloaded for "complete blood counting"? I don't know if these are standard Excel files in your field that everyone knows, or if not, then it'd be useful for others to see them.

* Include a public link for AutoMapper and version number (Line 698). I understand that it's not peer-reviewed, but it would still be useful to see this code in case someone wants to reproduce your analysis or investigate some parameters used among all the software included in AutoMapper.

* Can you clarify the CPM 10 reads cutoff? Is that a cutoff for the mean number of reads across all samples? Or if one sample has < 10 reads, then it's dropped?

* Cite limma, voom, STAR, Trim-Glaore, RSEM, Gencode, edgeR for the TMM method, GSEA, MSigDB, and other methods I might have missed.

* Include an equation or R formula for the model used for the bulk RNA-seq differential expression analysis (Lines 697 to 709). You could also explain why some variables were included in the model.

* Line 724: what version of Cell Ranger was used? If you used versions prior to 7.0.0, did you use --include-introns?

* How was the 40% cutoff chosen for the ribosomal/mitochondrial genes/proteins chosen? Did you try using automatic outliers methods like `scater::isOutlier()` or equivalents?

* How were the cutoffs for low and high UMI counts chosen?

* You might want to use <https://bioconductor.org/packages/release/bioc/html/scDbtFinder.html> for detecting doublets instead of the $\geq 32,768$ UMI threshold you are currently using.

* Why did you normalize the UMI counts to 10,000 UMIs per cell?

* What were the parameters used for the tSNE and what software/method was used to compute it? Without that information the tSNE described is not reproducible.

* What parameters were used for the differential expression analysis with <https://doi.org/10.1186/s13059-018-1438-9>

* Line 775: what version of Seurat was used?

- * Line 779: what version of ggplot2 was used?
- * Line 776: what is the "Seurat's dimension reduction" functionality? Can you be more precise and mention the specific function used as Seurat has several options.
- * Cite Seurat and ggplot2.
- * <https://www.ncbi.nlm.nih.gov/geo/query/acc.cgi?acc=GSE195575> does not include the high resolution images. For example at <https://www.ncbi.nlm.nih.gov/geo/query/acc.cgi?acc=GSM5840728> there is a 18.1 Mb JPG file, which is not useful for methods that require access to the high quality images.
- * How was Figure 3B generated (the heatmap)? In general, the figure captions for the heatmaps don't mention what we are looking at: normalized log counts? raw counts? Some seem to be Z scores, like Figure 2E.
- * Can you comment on the 3 outlier samples in Figure 2E and Figure 3B? I see 2 blue and 1 red mixed with the other color. Should those samples be excluded or is there some reason why you are still including them in the analyses? Are those 3 samples outliers when you visualize the top principal components or MDS for this data?
- * Why did you change the colors for Figure 4E and 4F? Blue used to be normal, now it's hepatoblastoma. This is confusing. This also affects Figure S2 panel D and Figure S3 as the colors for normal hepatoblastoma or heptoblastoma (is that a typo in Figure S3's color legend?) are not consistent.
- * Figure 4C: I can't see yellow although it's described in the caption. I also can't differentiate the light and dark blue. I highly recommend choosing other colors and to use colors that are color blind friendly. If you have ggplot2 objects, you could use <https://github.com/clauswilke/colorblindr> for example.
- * Figure 4D and G: what are we looking at? log counts? counts?
- * Figure S4 B: we can't see the B-cells with that color scale.

Here are some typos I noted:

- * Page 8 line 156: "associated downregulated" -> "associated with downregulated"
- * Page 8 line 165: "levels" -> "level"
- * Page 9 line 170: "caner" -> "cancer"
- * Line 653: "manufacture's" -> "manufacturer's"
- * Line 699: "firs" -> "first"

REVIEWER COMMENTS

Reviewer #1, expertise in hepatoblastoma and liver cancer models (Remarks to the Author):

In this manuscript, Fang et al developed a liver-specific MYC-driven hepatoblastoma (HB) murine model (ABC-Myc mice) that drives rapid hepatic oncogenesis. Authors showed that pathological and transcriptomic analyses reveal traits and signaling pathways characteristics of human HB. Another interesting aspect of this manuscript is the CRISPRCas9 screening to identify oncogenes/tumor suppressor genes in HB as well as genes that synergizes/antagonizes with doxorubicin. Specifically, they found that PRKDC inhibition enhanced the efficacy of doxorubicin chemotherapy, improving its anticancer efficacy in vitro and in vivo using the ABC-Myc and human HepG2 models. This tumor model seems to be an exceptional tool for future studies in the HB field, a, extremely rare disease from which few tumor models have been reported (PMIDs: 27734029, 24837480, 24848510).

We highly thank this reviewer's positive comments on our study.

Despite the relevance and the useful opportunities of this disease models and the knowledge generated about the HB biology using the CRISPRCas9 screening of ABC-Myc cell lines with and without doxorubicin, the paper needs to be improved. In that regard, certain points still need to be clarified, some analysis/figures (specifically those related to Hba-a1 and triptofan pathway) need to be eliminated or moved to supplementary data, some analysis need to be improved and additional experiments should be performed to get robust conclusions. In addition, the figure legends lack a lot of information and difficult the reading of the manuscript. Please find below a summary of the major and minor comments to the Fang et al manuscript.

We appreciate this reviewer's constructive comments and suggestions, which has led to a great improvement of our study. Now we have made extensive revisions to address each point as highlighted below.

Major comments

- Nomenclature of the ABC-Myc tumors. The authors along the paper stated that their model recapitulate an HB. However, due to the fact that these tumors don't harbor Wnt/beta-catenin pathway activation, a key hallmark constitutively activated in nearly all human HBs, it would be more appropriate to refer the ABC-Myc tumors as "HB-like tumors" along all manuscript.

We agree with this reviewer. We have changed to "HB-like tumors" along all manuscript. Nevertheless, Wnt/beta-catenin pathway seems to be active in this model as indicted by a strong cytoplasmic staining of beta-catenin (Figure 1f) and GSEA analysis (Figure 3c).

- Histopathological analysis: HB is a complex disease at pathological level with usually a mixture of tumor cells with different degrees of hepatic differentiation ranging from highly pleomorphic HB to tumor cells resembling embryonal or fetal hepatoblasts (Lopez-Terrada et al, Modern pathology, 2014). In that regard, authors pointed out that the pathologist found embryonal and fetal epithelial HB components. In that regard, the histological study of tumor lesions in the ABC-MYC model needs to be deeply studied. Specifically: how is the heterogeneity among the different tumor nodules of the same mice? Are there any pre-neoplastic lesions? Which is the main epithelial component of the lesions? It is important to remark that the main epithelial component has been associated to HB subtypes and patient outcome (Cairo et al, Cancer Cell, 2008). How are the tumors of different mice? Is there any mesenchymal component? Authors also mentioned that a "648 board-certified veterinary pathologist (HT)" (lines 648-649) reviewed the HE slides of 4-8 models. In that regard, the pathological study should be performed by an expert pathologist of human HB to describe better the pathology of these tumors.

To understand the disease features as this reviewer asked above, hepatoblastoma-like tumors were re-evaluated by both a board certified veterinary anatomic pathologist (Heather Sheppard) and a MD solid tumor pathologist (Selene Koo, Teresa Santiago). We have addressed all comments in our revised manuscript.

Specifically speaking, additional samples at time points E14.5, E 17.5, and P7 were

evaluated to address the presence of pre-neoplastic lesions. Neoplastic transformation was first observed in E17.5 livers in low numbers of scattered developing hepatocytes with abnormal nuclear morphologies. Nuclear changes consisted of karyomegaly, marginalization of chromatin, and a single, centralized, and prominent nucleolus that is consistent with other cancers where constitutive

MYC activation is present (**Supplementary Figure 1c**). These dysplastic cells were interpreted as pre-neoplastic lesions based on the biological time course of the ABC-MYC mouse model as described in this paper.

Neoplastic nodules were grossly visible in all liver sections of ABC-Myc mice starting at P7. Multifocal to coalescing neoplastic foci with an embryonal morphology could be observed in the livers of P7 ABC-Myc mice (**Figure 1e**), consistent with the hypothesis that hepatoblastoma-like neoplasia may arise from epithelial-lineage committed hepatic stem progenitor cells with the introduction of human oncogenic Myc signaling resulting in impaired differentiation. Further

evaluation of the hepatoblastoma-like tumors from time points P25 to P67 showed a coexistence of distinct subpopulations of neoplastic cells with embryonal, fetal, and rarer cholangioblastic-like morphologies (**Figure 1e**). The co-existence of these morphologies in advanced hepatoblastoma-like tumors is most consistent with human pediatric hepatoblastoma with a mixed epithelial phenotype. Small cell undifferentiated, rhabdoid, teratoid, and mesenchymal morphologies were not observed. There were no definitive well-differentiated fetal morphologies identified in the sections except within P67 tumors (**Figure 1e**). All tumors had combinations of primitive morphologies comparable to the previously described C2 morphologic phenotype as described by Cairo et al. Hepatoblastomas characterized as C2 are documented to have aggressive biological behavior and an unfavorable prognosis, which is observed in this model. (**Figure 1e, and Table 1**).

Figure 1e

- Immunostaining of ABC-MYC tumors: Authors showed in figure 1E-J the images of some immunohistochemistry (IHC) markers in the tumors. In relation to the above comment, authors should have quantified the staining of the different IHCs to provide a better idea about the heterogeneity/homogeneity of the tumor nodules of the different mice. Authors should also provide a non-tumor liver tissue as a control of the IHC. To confirm the similarities of the ABC-MYC tumors with aggressive HBs, it will be necessary that authors complete the IHC analysis with stem cell markers (i.e. EpCAM, GPC3, etc) as well as with proliferating markers (i.e. Ki67, PCNA). Authors should also justify why ABC-MYC tumors have strong expression of GLUL (marker of C1 tumors) and low expression of AFP (marker of aggressive C2 tumors); as shown by Cairo et al (Cancer Cell, 2008). Finally, Supplementary Figure S1 c-H seems to provide identical information that Figure 1E-J. Authors should delete it or specify which are the differences.

Table 1. Pathological characterization of ABC-Myc liver tumors

Accession Number	Fetal	Embryonal	Crowded	Macrotrabecular	Cholangioblastic	Mesenchymal	INI1	SALL4	GPC3	AFP	GLUL	ARG	Bcat	Grouping
RS19-2057	+	+	-	-	+/15-20%	-	+/100%; retained	+/90%	+/100%	+/100%	+/100%	+	+/100%; membranous	C2
RS22-578	-	+	-	-	+/<5%	-	+/100%; retained	+	+	+	+/patchy & strong	+	+/100%; membranous	C2
RS22-575	+	+	-	+	+/<5%	-	+/100%; retained	+/15-20%	+	+	+/87%	+	+/100%; membranous	C2
AP21-532	+	+	+	+	+/<5%	-	+/100%; retained	+	+	+	+	+	+/100%; membranous	C2
RS22-576	-	+	-	-	-	-	+/100%; retained	+	+	+	+	+	+/100%; membranous	C2
RS22-577	+	+	-	-	+/<5%	-	+/100%; retained	+	+	+	+	+	+/100%; membranous	C2

We further determined the pathological features of this hepatoblastoma-like malignancy using immunohistochemical markers of human pediatric hepatoblastoma, and observed overexpression of hepatic stem/progenitor cell markers documented in C1 and C2 human pediatric hepatoblastomas (**Figure 1f**). Murine hepatoblastoma-like neoplasms had diffuse immunopositivity for alpha fetoprotein (AFP) and glypican 3 (GPC3), two stem cell markers

used to distinguish neoplastic hepatocellular cells, as well as immunoreactivity for glutamine synthetase (GLUL or named as GS), a beta-catenin target and a marker of beta-catenin activated hepatocytes, SALL4, another embryonal hepatoblastoma marker, and Arginase-1 (ARG-1), a marker used to distinguish primary hepatocellular tumors from metastatic tumors (**Figure 1f**). Immunoreactivity was visually observed in greater than 75% of the bulky hepatoblastoma-like neoplasms and staining intensity for all markers was visually graded as moderate to strong in staining intensity for all markers (**Table 1**). Rare subpopulations of poorly differentiated neoplastic cells, visually quantified at less than 1% of the neoplasm, had immunoreactivity for cytokeratin 19 (KRT19), a marker for biliary cancer or small-cell undifferentiated type hepatoblastoma, as well as non-neoplastic, entrapped bile ducts. INI1 (SMARCB1) was retained in all neoplasms, further demonstrating the hepatocellular origin of these cells. The strong cytoplasmic staining of beta-catenin may suggest an activation of the Wnt/beta-catenin signaling pathway in these tumors (**Figure 1f**). In summary, the ABC-Myc hepatoblastoma-like model recapitulates the embryonal or mixed fetal and embryonal histologic features of human hepatoblastoma and has anatomic and molecular characteristics of human disease highly associated with the high-risk C2 subtype¹¹.

Figure 1f

KI67 immunoreactivity was examined for a subset of tumors from P25-P36, and ranged from 2-6% (**Supplementary Figure 1d, 1e**), primarily observed in foci of extramedullary hematopoiesis. Rare subpopulations of neoplastic cells expressed KI67 in IHC staining suggesting that HB-like tumors had a low proliferation rate. However, the positivity of KI67 in IHC seemed to be much lower than what we observed in our scRNA-seq (**please see Figure 4e, 4f**) and spatial transcriptomics (**please see Figure 5b, 5c**). We tentatively attributed this discrepancy to technical issue. The human anti-KI67 antibody probably has a lower binding affinity to mouse KI67.

We have no definitive answer to the question why ABC-MYC mice expressed high levels of GLUL. Interestingly, we observed that *GLUL* expression is not higher in tumor samples than the control livers in our scRNA-seq and spatial transcriptomics (**please see Figure 4e, 5a**), and the *GLUL* expression seemed to be negatively correlated with the expression of stem cell markers (**DLK1, GPC3, and EPCAM**) (**please see Figure 5a**). These data suggest that *GLUL* expression pattern in ABC-Myc tumors is consistent with Cario reported. We do not think AFP

level is low in ABC-Myc tumors. IHC staining showed strong positivity of AFP in all assessed tumors, which is consistent with the results from our bulk RNA-seq, scRNA-seq and Spatial transcriptomics. Some retention of both C1 and C2 characteristics by our murine hepatoblastoma-like neoplasms may result from differences in transgene copy number expression in the embryonal liver; this difference may affect the timing of malignant transformation in susceptible hepatocyte specified stem-progenitor populations. We have added one paragraph to discuss this possibility (line 139-156)

We have deleted Supplementary Figure S1 C-H.

- Wnt/beta-catenin activation in the ABC-Myc model. It is surprising that the ABC-Myc mice generate tumors without activating Wnt/beta-catenin pathway, a key hallmark of HB (>80% of the human tumors have point mutations or deletions of exon 3 of the CTNNB1 gene). In relation to beta-catenin, authors mentioned a "lack of strong nuclear translocation of the wild-type beta-catenin" (line 105-106). However, in the figure 1, it is difficult to appreciate nuclear staining instead there is a marked staining of CTNNB1 in the membrane (localization of beta-catenin when Wnt signaling is not active). In addition, the expression of Wnt/beta-catenin GSEA analysis is not highly convincing (Figure 2D) because the geneset used "FEVR CTNNB1 TARGETS DN" came from intestinal crypt cells upon deletion of CTNNB1. Because beta-catenin is activating different target genes depending on the cell type and differentiation cell degree, the study of beta-catenin target genes should be improved by using a gene set of beta-catenin target genes induced in liver hepatocytes/hepatoblasts. As this reviewer pointed out, it is difficult to appreciate nuclear staining of CTNNB1. However, we noticed strong cytoplasmic staining of CTNNB1 (**Figure 1f**), which suggests CTNNB1 might be active.

To further verify whether Wnt/beta-catenin pathway is activated in ABC-Myc tumors, we have used two gene sets including (1) Top 500 genes downregulated by CTNNB1 knockdown in HEPG2 cells (GSE94858, Biotechnol Bioeng 2017 Dec;114(12):2868-2882) , (2) Top 500 genes upregulated in liver tumors transformed by CTNNB1 mutant (GSE79084, JCI Insight 2016 Oct 6;1(16):e88549). The results support our conclusion that Wnt/beta-catenin pathway is active (at least to some degree) in ABC-Myc HB-like tumors (**Figure 3c**).

Figure 3c

-Biochemical analysis of plasma from ABC-Myc mice harboring HB-like tumors. Authors performed a comprehensive clinical chemistry analysis of serum from ABC-Myc. The results showed a clear liver damage with an increase levels of hepatic enzymes and renal dysfunction (at what day are obtained the plasma sample? At sacrifice? Please specify at M&M). These results are probably explained to the advanced stage of the tumors and not to the fact that they are harboring an HB-like tumor. Please specify this. In addition, since alpha-fetoprotein (AFP) is a classical marker of HB in patients, authors should complement the biochemical analysis of the plasma of this model by measuring the levels of AFP in ABC-Myc mice vs. controls.

The plasma and serum samples were obtained at sacrifice time when mice became moribund. As this reviewer pointed out, the results represent the advanced stage of the tumors. We have clarified this in M&M.

We have also examined the levels of AFP in serum in ABC-Myc and control mice. Although the AFP levels were significantly higher in ABC-Myc mice than the normal controls (**Figure 2a**). However, *Afp* mRNA levels in tumor samples were even much higher based on our RNA-seq and spatial gene expression.

-GSEA analysis (figures 2B, C, D and F): This analysis has major drawbacks. For one side, the authors stated that figures 2B-D with "GSEA showing gene signatures or signaling pathways highly upregulated in ABC-Myc hepatoblastoma". The main problem is that these signatures came from adult hepatocellular carcinoma or intestinal crypts. In that regard, authors should use geneset signatures from gene expression profiling studies of human HB. For another side, authors should specify in the figure legend if the p-value is the FDR (only FDR < 0.25 should be considered significant) as well as include the normalized enrichment score (NES). Moreover, in the M&M section, due to the limited number of samples included in the GSEA analysis and authors should provide details about the approach used (i.e. number of permutations, describe the enrichment statistic used and the metrics for gene ranking).

We thank the reviewer for this suggestion. Now we have included the gene sets from Cario paper (Cancer Cell, 2008) (Genes down in HB vs normal, and genes down in C2 class; Genes up in HB vs normal, and genes up in C2 class). (**Figure 3b**)

However, we cannot find EpCAM gene sets specifically upregulated in HB; therefore, we have kept the original GSEA for EpCAM.

We have added the NES and FDR in GSEA results.

In M&M, we have added more information of GSEA analysis (number of permutations = 1000, permutation type = gene_set, metric for ranking genes = Signal2Noise, Enrichment statistic = Weighted).

Figure 3b

- Characterization of the degree of tumor immaturity. Authors mentioned (lines 171-173) that “Consistent with the embryonal histological features, GSEA results showed that ABC-Myc tumors had significant upregulation of cancer stem cell signatures including “liver cancer with upregulated EpCAM” (Figure 2F), and “undifferentiated cancer” (Figure 2G).” However, to complete the study of the degree of tumor immaturity it would be interesting to show the expression of stem cell or liver progenitor markers that are characteristic of aggressive HBs. In addition, the increased expression of stem cell markers will provide further support of the aggressive phenotype of ABC-Myc tumors.

We have examined the stem cell markers including SALL4 and GPC3 in our IHC in Figure 1 and Table 1. Please see our response above.

- Gene expression profiling (figures 2E, G, H and I). Figure 2E is a heatmap showing the top differentially expressed genes in ABC-Myc tumors vs. age-matched normal livers and authors concluded that these genes found dysregulated in the ABC-Myc model are similar to the ones altered in human HB tumors. This analysis would be improved if the authors provide the % of the overlapping genes altered in human HB vs non-tumors and MYC-ABC tumors vs control livers at a specific FDR.

To address the reviewer’s comments, we have collaborated with Dr. Carolina Armengol, performed analysis using the VENN diagram that shows the number of deregulated genes

(and their %) in the comparison between tumor vs. non-tumor liver samples obtained from patients with HB and from the ABC-Myc model at FDR<0.05. Briefly, we integrated the RNAseq database from MYC-ABC tumors (n=3) and control livers (n=3) with the RNAseq database from Carrillo-Reixach et al study including a total of 66 tumor and non-tumor samples from 32 patients with HB (n=34 tumors including 3 recurrences; n= 32 non-tumors). As a result, we obtained a matrix of 11,393 ortholog genes. Then, we performed the supervised analysis comparing tumor vs. non-tumor samples using human and mouse samples. The resulting significant lists indicated a strong significant overlapping in which 50.1% and 42.5% of the up- and down-regulated genes in the MYC-ABC tumors vs. control liver (CL) samples were also deregulated in human HB as compared with non-tumor (NL) samples, respectively. This result clearly supports the high similarity (>50%) of the MYC-ABC tumor model with human HB (**Figure 3d**).

To confirm the high similarity between human and mouse samples, we used the integrative human and mouse RNAseq database of ortholog genes to perform a Principal Component Analysis (**Figure 3e**). This analysis showed that tumor and non-tumor samples were clearly different between them independently on the species from which samples were obtained. Specifically, mouse tumor samples were grouped with human tumor samples and control mice liver samples were grouped with adjacent non-tumor samples from patients with HB. Altogether our new data clearly indicates the high similarity of our MYC-ABC model and the human HB and support its use as an experimental model for this extremely rare disease.

In addition, authors showed that ABC-Myc tumors had C2 and progenitor and proliferation signatures of high-risk HB. However, the control sample of gene expression profiling used for the heatmap is "normal" (non-tumor tissue). Accordingly, we cannot assure that C1 tumors could also share some C2 tumor profiling since as tumors they are "per se" also more immature and more proliferative than non-tumor tissue. A possibility to confirm the C2 nature of the ABC-Myc tumors, authors should integrate the expression profiling of human and mice tumors and study its aggrupation in an unsupervised heatmap.

As this reviewer suggested, we have performed additional analysis to integrate expression profiles in ABC-Myc tumors with human HB data.

The correlation between the gene expression of human and mouse samples showed four main group of tumor samples (**Figure 3g**). Interestingly, the gene expression profile of mouse ABC-Myc tumor samples were highly correlated with that of the human primary HBs and specifically, with tumors of the proliferative C2- Pure subclass ($p=0.019$) and with a strong

14q32-gene signature overexpression ($p=0.027$). These molecular features have been already reported to be associated with clinical features of poor prognosis (see Carrillo- Reixach et al, J Hepatol, 2022).

* the second cluster of additional 5 C2 tumors (cluster at the left side from the cluster with mouse samples) was a cluster mainly enriched with tumor recurrences (3/5 tumors).

Figure 3g

Finally, the study of the expression profile of the 11 ortholog genes of the 16-gene signature confirmed that the mouse ABC-Myc tumors had a similar profile to the human C2 HBs (**Figure 3h**); thereby confirming this model as a model of molecular aggressive tumors.

- Metabolic pathways dysregulation. This part of the study is not relevant, the analysis performed are poor and in somehow biased to the Tryptofan pathway that, as so many pathways involved in the liver function, is down-regulated in HB. In that regard, the message is confusing and contributes to disperse the main message of the manuscript. In addition, authors stated that the "tryptophan catabolism is one of the most significantly altered metabolic pathways in hepatoblastoma (...)" (from line 209), but there is no strong data on this sentence in the literature and authors neither provide the specific references. Therefore, to improve the quality of the manuscript, authors should delete this section in the manuscript and the corresponding figure or move it to supplementary data including an additional analysis of other liver specific pathways (not just tryptophan pathway).

We agree with this reviewer that the metabolic section is not relevant. We have deleted in our revision.

-scRNA and spatial transcriptomics. In general, the analysis of scRNA is very poor for several reasons: 1) Authors should include a healthy mouse control liver (scrape and spatial transcriptomics) in order to see differences in cell populations in comparison with the ABC-Myc tumor. In that regard, they cannot mention that "tumor associated macrophages were abundant, suggesting that the ABC-Myc environment is immunosuppressive" (lines 278) because there is no control tissue analyzed; 2) authors did not identify the tumor cell populations. They just were focused on the stromal (Suppl. Fig 4) and non-tumor (Figure 4G) cell populations and they did not characterize the heterogeneity of cell tumor content. To note that pathologically, human HB but also ABC-Myc tumors exhibit embryonal and fetal tumor cells... are these distinct populations found in scRNAseq? A comparison of spatial transcriptomics data with pathological review of the tumors would be needed. Accordingly, only AFP has been studied as a tumor marker (when its expression is not very strong by IHC) and authors should identify the tumor cell populations by looking at other tumor markers such as GS (this is very important because it has been demonstrated by IHC that tumor cells express high levels of glutamine synthetase), DLK1, GPC3, KRT19 or EpCAM among others. 3) Apparently, there is an incongruence among scRNAseq and spatial transcriptomics and the expression of Hba-a1 and Afp. In that regard, the number of cells expressing Hba-a1 is very high in Figure 4B but very low in Figure 4D and viceversa for Afp. Authors should provide an explanation. 4) Authors indicate that one subpopulation of cells has a high levels of hemoglobin cells (line 262-263) but in Figure 4B we can see that at least 4 subpopulations of cells have high levels of hemoglobin cells. Which type of cells are? In addition to hemoglobin genes, which genes do these subpopulations overexpress? This is a very important issue taking into account that HB is characterized by having extramedular hematopoiesis, we could think that these populations are precursors of blood cells. Authors studied the expression of Hba-a1 using HB RNAseq datasets but it would be also important to compare with RNAseq databases of hematopoiesis or red blood precursors. 5) The fact that there is a negative

correlation between Afp-Hba is not consistent with the fact that it has been reported that high-risk patients have high Hba-a1 expression according to Sekiguchi et al (lines 266-267). In that regard, it has been clinically demonstrated that high levels of AFP are associated with a higher risk patients and high levels of AFP in plasma are included in the current clinical stratification CHIC-HS as a marker of poor prognosis (Meyer et al, Lancet Oncol, 2017).

In addition, Figure 4F is not informative and should be deleted. The same for the sentence: "Although not statistically significant, the expression between AFP and HBD (one of the adult human globin genes) tended to be negatively correlated in human hepatoblastoma tumors (Figure 4F), similar to the spatial expression pattern of Afp and Hba-a1 in ABC-Myc tumors" (lines 267-270).

To answer these important questions from this reviewer, we have performed additional scRNA-seq in 4 more ABC-Myc tumors and 3 normal controls, as well as spatial transcriptomics in 3 more tumors and 3 normal controls. We have made extensive revisions to understand the heterogeneity of ABC-Myc tumors and discussed the erythroid markers **(please see details in Figure 4 and 5 in our revised manuscript)**

- Genome-wide CRISPR screen: This part of the study is very interesting and the results provided including essential and tumor suppressive genes are in agreement with the literature. However, the figure 5D is not easy to understand. In that regard, authors should provide more details about the lists/genesets used for this analysis (i.e. hepatoblastoma C2 subclass: what this include? The 8 C2-specific genes of the 16-gene signature reported in Cairo et al (Cancer Cell, 2008)?

Sorry for the confusion. This is not the 16-gene signature. We have added the information (CAIRO_HEPATOBLASTOMA_CLASSES_UP, n=612).

The chapter "Genome-wide screen of cancer dependency genes in an ABC-Myc-derived hepatoblastoma" should be integrated with the "Murine and human hepatoblastoma share common essential and targetable genes" since the later should be considered as a finding validation in human HB cells of the CRISPRCas9 experiment in the murine cancer cell line (Figures 5G and H) and the results are all included in the same figure 5.

We have merged the two parts as "Murine and human hepatoblastoma share common essential and targetable genes".

- Huh6 gene mutations: authors should correct the following mistake when mentioned that "HuH6 bears TP53 and AXIN1 mutations (depmap.org)" because Huh6 has Gly34Val CTNNB1 mutation instead of AXIN1 mutation as it is specified in the demap.org website.

We thank this reviewer for the information. We have now corrected it.

- Chemotherapy response. The authors explained that they treated the cells with doxorubicin at two doses (5nM and 30nM) for 7 and 14 days. However, figure S7 did not justify this choice. Please, justify. In addition, the authors should include the information of the axes in the plot of figure S6B and split the X-axis in 2 parts (0-100 nM and >100nM) to appreciate better the results on colony intensity.

We have plotted the killing curve by performing a Prestoblu e assay and determined the IC₂₀, IC₅₀, and IC₉₀ (**Supplementary Figure 9**). We have justified the choice in figure legend.

Plot showing the cell viability curve assessed by Prestoblu e assay. NEJF10 cells were treated with different concentrations (n=8 for each concentration) of doxorubicin in 96-well plate for 4 days. Since doxorubicin has a very narrow therapeutic window at the nM level, we decided to reduce the doses for IC₂₀ and IC₉₀ by 5 nM to become ~ 5 nM for IC₂₀ and ~ 30 nM for IC₉₀ for a 2-3 week treatment in CRISPR screening.

- PRKDC inhibition enhances efficacy of chemotherapy. This part of the study is highly interesting since authors demonstrated the synergistic effect of PRKDC (a gene found in their screening) to sensitize tumor cells derived from ABC-Myc mice to doxorubicin. Authors validate the study in vivo and in vitro but to get robut conclusions, they need to complement it with the use of another PRKDC inhibitor and an at least additional human HB cell line (i.e. Huh6 previously used by the authors).

We have tested another PRKDC inhibitor, NU7441 (Bioorg Med Chem Lett 2004;14:6083-7), in several ABC-Myc cell lines and HepG2, and obtained similar results as AZD7648 (**Supplementary Figure 13**)

Supplementary Figure 13

Additionally, we have tested the combination therapy in one HB PDX model. The results showed a significant antitumor effect induced by the combination therapy (**Figure 8h**). These results further validated our conclusion that combination of doxorubicin with PRKDC inhibitors will enhance the efficacy.

Figure 8h

- Statistical analysis in the “Genetic and Chemical Perturbation database” should be checked because the log₁₀FD is too high ranging from 5 to 200. The CRISPRCas9 screening for positive selection genes is not robust if we take into account the FDR; authors should justify this.

We are sorry for the typo. Log₁₀FD should be log₁₀(FDR). We have corrected this mistake.

- All Figure legends have very limited information that difficult the comprehension of the results. Therefore, authors should expand them to provide much more information such as number of samples, statistical methods used, GEO code, database references, meaning of color codes, etc

We apologize for this. In our revision, we have added detailed information in Figure Legend.

Minor comments

-Figure 1C. Authors should specify in the figure the number of mice and specify the time units (days?)

We have specified the mouse numbers in the figure legend.

-Figure 1D. There is a lack of control of MYC expression in P5 and P10. Please add or justify why it is not necessary.

We have added the control MYC expression.

-Figures 2H-I and 3F: Authors should add a color scale

We have added the color scale.

-Supplementary tables should be numbered according to the text.

We have numbered the tables.

-Line 172: typo error "caner" instead of "cancer"

Sorry for the typo. We have corrected it.

-Fig. 3A: Why some genes are highlighted in green?

Figure 3 has been removed according to reviewer's suggestion.

-Fig. 3B: Why are some genes highlighted with a red arrow?

Figure 3 has been removed according to reviewer's suggestion.

-Figure 5D: not clear the legend because the FDR is not indicated.

We have included FDR.

-All heatmaps should include color score index bar.

We have added the color score index bar.

-Figure 4D, 4E and S4B: the colors are not clear (the quality of the images are too low).

The whole panel of Figure 4 (scRNA-seq and spatial gene expression) has been replaced with new data.

-HB datasets: authors use different datasets in different analysis (i.e. Hooks et al for the study of Hba-a1 in HB, Carrillo-Reixach et al and Sumazin et al for the triptofan pathway study). Authors should use all datasets to validate the different findings.

We have deleted the triptofan pathway as the reviewer suggested.

-Please change "NEGLOG10FD" for in Suppl Figure 5 for $-\log_{10}(\text{FDR})$ as it is mentioned in Figure 5

We have removed this Figure and replaced with new figures since we have expanded CRISPR screening by including more cell lines.

-Figure 5G: authors should delete "Figure 5"

We have deleted it.

-Figure 6: delete the 2 sentences "Network analysis performed using STRING program (<https://string-db.org/>)" and include this in M&M

We have moved it M&M as suggested.

-Figure 7: order the panels in a homogenous way in all paper, especially in this figure.

We have tried to re-order the panels, but it was difficult to fit so we just left as it was. Sorry about this.

Reviewer #2, expertise in CRISPR-Cas9 screens and functional genomics (Remarks to the Author):

Fang and colleagues describe studies using a mouse model of hepatoblastoma, aiming at identifying new therapeutic targets. The authors describe (i) the characterization of the mouse model, which is based on hepatocyte-specific expression of c-Myc, (ii) molecular and histopathologic characterization of tumors, thereby showing that the model predominantly reflects high-risk human medulloblastoma, (iii) derivation of mouse hepatoblastoma cell lines and their use for genome-wide CRISPR/Cas9 screens to identify new vulnerabilities, (iv) genetic screens to identify modifiers of standard chemotherapy and (v) validation of one such combinatorial therapy in vitro/in vivo using mouse and human hepatoblastoma models.

High-risk hepatoblastoma is not only incompletely understood at the molecular level, but there is also an urgent need for new therapies. This study contributes some valuable cellular resources, gives some new insights into the genetic dependencies and also validates a potential new therapeutic approach. I do have however several major concerns, including the limited novelty of some parts of the work, the insufficient depth of analyses in other parts or overstatements stemming from .

We highly thank for this reviewer's constructive comments and suggestions. In our revision, in addition to in depth pathological characterization of ABC-Myc tumors, we have performed scRNA-seq and spatial transcriptomics in 4 additional tumors and 3 normal livers, and for the first time investigated the heterogeneity and molecular features of murine HB at single cell levels and spatial levels; we have performed CRISPR screening in additional two cell lines and had a robust validation of screening results. We have validated the combination therapy in multiple cell lines and included one PDX model for in vivo validation. We have also changed our statement about "mapping oncogenic pathways" to "mapping cancer dependency". With these extensive revisions, we believe we have made great improvement to our study.

Please see new data including Figure 1e, 1f; Figure 3d, 3e, 3g, 3h; Figure 4; Figure 5; Figure 6g-6j; Figure 8h, and relevant Supplementary data.

1. The mouse model. Previous work reported Myc-induced aggressive/poorly differentiated hepatoblastomas in mice. This includes the dox-inducible Tet-o-Myc/LAP-tTA model (Goga et al, Nat med 2007 Jul;13(7):820; characterized in detail in comparison to the bCat-mutant/Myc mice by Comerford et al, JCI Insight. 2016 Oct 6;1(16):e88549.) or models based on hepatic Myc delivery by hydrodynamic tail vein injection (e.g. Mol Ther 2022 Apr 6;30(4):1645). Thus, although the Alb-Cre;LSL-Myc model reported here is a useful model for childhood hepatoblastoma, the conceptual advance is relatively limited.

We thank this reviewer for referencing the previous reports regarding to hepatoblastoma mouse models. In our manuscript, we acknowledged that our model was not the first mouse model of hepatoblastoma. However, we believe our study has made conceptual advance in understanding the MYC-driven hepatoblastoma at single cell and spatial gene transcriptomics levels, and substantiated the clinical relevance of our model to high-risk disease. To further strengthen this, we have performed single cell RNA-seq and spatial gene transcriptomic analyses in additional three normal livers and four hepatoblastoma tissues in our revision.

As this reviewer mentioned, the Tet-o-Myc/LAP-tTA model is a great liver cancer model published by Dr. Dean Felsher (Nature 2004; 431:1112–1117). However, our ABC-MYC mouse model has its advantages. First of all, MYC expression is not controlled by doxycycline, which is known to disturb mitochondrial function (*Cell Rep*, 10(10), 1681-1691) cause mouse embryonic lethality (Cancer Cell. 2014;26(2):248-61). Second, the genetic background of ABC-MYC mouse is C57BL/6J, which has been well characterized for studies in immunotherapy. The genetic background of Tet-o-Myc (FVB/N-Tg(tetO-MYC)36aBop/J) is FVB/NJ. Although the genetic background of another transgenic Tet-o-Myc mouse strain (B6.FVB-Tg(tetO-MYC)36Bop/DwfJ) generated by Dr. Dean Felsher is C57BL/6J, the MYC transgene is inserted in Y chromosome, and therefore only male could develop phenotype. Thus, our ABC-MYC model could have a broader application including immunotherapy studies.

Comerford et al indeed used the Tet-o-Myc/LAP-tTA model to compare the pathological features with their model (β -cat Δ Ex3:MyC) (JCI Insight. 2016;1:e88549), and confirmed that Tet-o-Myc/LAP-tTA mice developed embryonic hepatoblastoma. However, this study did not characterize the molecular features of hepatoblastoma generated from the Tet-o-Myc/LAP-tTA mice.

The recent observation by Wang et al that Myc delivery by hydrodynamic tail vein injection led to development of hepatoblastoma-like liver cancer is interesting (Mol Ther 2022; 30:1645). However, this study did not characterize the tumors by using either specific hepatoblastoma markers or transcriptomics approach. They performed HDTV in 6 week-old male mice (at this age, mice are mature), which did not align the developmental program of liver. Additionally, one major limitation of HDTV for gene delivery is that genes are predominantly taken by the hepatocytes in the pericentral region (zone 3 of the liver acinus). Therefore, this approach cannot be applied to study tumors originating from hepatic stem cells which reside in periportal region (zone 1) (Am J Pathol. 2014; 184: 912–923. Hepatology 2011;53:1035-45). It is well accepted that the embryonic hepatoblastoma is transformed from hepatic stem cells.

2. The CRISPR screens have been performed in one mouse hepatoblastoma cell line only. All conclusions are drawn from this line. A similar screen has already been performed in the HuH6 human hepatoblastoma line (ref 68), and those human data are now being used by the authors to support findings in mice. Whilst this comparative approach is valid, the screen

performed by the authors adds only limited new information to the existing human data. Screening much larger cell numbers would permit corroboration of findings and enable further analyses, such as mapping of genetic context-dependencies.

We thank this reviewer for this insightful comment. In our original submission, we only focused on the common genes shared by NEJF10 and Huh6. We now have performed CRISPR screening in two additional ABC-MYC lines (NEJF1 and NEJF6), and compared with Huh6 with detailed analysis, which again showed a large number of dependency genes shared by all. We also identified individual genes converged to the common cancer signaling pathways, and we also identified distinct cancer dependency genes in each cell line. Particularly, we have identified heme biosynthesis pathway is specifically selected in NEJF1, and genetic mutations of heme synthesis genes have been recently identified in liver cancers (*J Hepatol.* 2022;77:1038-1046). We have included the results in our revision (**Please see Figure 6 and Supplementary Figures 6-8**).

3. Mapping of synergies with chemotherapy. Identification of positive and negative selection trends specific for the doxorubicin-treatment setting requires more scrutiny, confirmation in multiple cell lines and experiments, as well as rigorous genetic and pharmacologic validation in separate experiments (e.g. using competition assays). In their current form, the analyses lack required depth.

We have further confirmed our results by using another PRKDC inhibitor, NU7441, which has a distinct chemotype from AZD7648, in multiple cell lines, and obtained similar results (**Please see Supplementary Figure 13**).

We have included one hepatoblastoma PDX model and further verified the efficacy of combination of doxorubicin and PRKDC inhibition (**Please see Figure 8h**).

We have also tried competition assay but unfortunately, we encountered technical issues after we transduced BFP and GFP into ABC-Myc cell lines. The cells seemed to be very sensitive to PRKDC knockdown after transduction of fluorescent proteins. We therefore gave up and hopefully this reviewer could understand our frustration.

4. Several claims made in the manuscript are not supported adequately by data. For example, "mapping of oncogenic pathways" is overstated, given that that (i) cancer genomes have not been characterized and that (ii) the functional studies to map dependencies are derived from one cell line only.

After expanding our CRISPR screening in more cell lines, now we feel it is appropriate by saying so as (1) the genes are indeed classical oncogenes or tumor suppressors, (2) they are conservedly important in all the cell lines based on our CRISPR screen. However, as this reviewer suggested, we have changed it in our title as "cancer dependency genes".

Reviewer #3, expertise in sc-RNAseq and spatial transcriptomics (Remarks to the Author):

Fang, Singh, Natarajan, Tillman et al present a study involving many different technologies spanning bulk RNA-seq, scRNA-seq, spatially-resolved transcriptomics (Visium by 10x Genomics), as well as CRISPR-Cas9, that were used to investigate the pathogenesis of hepatoblastoma. The authors link mouse in vivo experiments with human cell lines to find similarities between them in order to identify potential therapeutic targets in experimental conditions that could later be evaluated in clinical trials. Ultimately, this project is a great proof of concept and future studies could expand it with larger sample sizes.

We highly appreciate this reviewer for the positive comments on our study.

While I'm not an expert in hepatoblastoma, I was asked to help review this manuscript given my experience with the different RNA-seq technologies. Overall, my impression is that this project and manuscript is well written. However, there is still some room for improvement as some methods are only superficially described and the justification for some thresholds or analytical decisions is omitted in this manuscript.

We thank this reviewer for the constructive comments. We have made extensive revisions and added details in methods and the justification for thresholds.

Here are some main comments:

* Am I understanding correctly that only 1 sample with 11,696 cells was used to generate the scRNA-seq data? That seems to be the case looking at <https://www.ncbi.nlm.nih.gov/geo/query/acc.cgi?acc=GSE194051>. I would emphasize this as a limitation in the discussion to warn readers about over interpreting the results.

We have performed scRNA-seq and spatial gene transcription in three normal livers and four additional hepatoblastoma samples. Please see detailed results in Figure 4 and 5, and Methods in our revised manuscript. We believe we have generated robust results now.

* The methods for the scRNA-seq analysis are incomplete as right now it would be very challenging to try to reproduce the results with the provided information. Also, it seems like the authors are either choosing some arbitrary thresholds or they are using thresholds that they found useful for other datasets but that they didn't describe in the methods for this manuscript.

We are sorry for missing important information in our original submission. In our revision, we have included all need information.

* The three GEO accession links don't show any FASTQ files. Is this a limitation of the reviewer visibility settings or are these files missing? The SRA links don't work for me right now. <https://www.ncbi.nlm.nih.gov/Traces/study/?acc=PRJNA413799> from <https://www.ncbi.nlm.nih.gov/geo/query/acc.cgi?acc=GSE104766>, which is used in Figure 4E does work.

We apologize for this, and we have no idea why it did not work. We have included the link in our revision, and we believe it works.

* GSE195575 contains 4 Visium samples, but why is only 1 of them shown in the main and supplementary figures?

We apologize for this confusion. This 4 Visium samples were from one sample and that why we only showed one. Now the Figure has been replaced by a new one that includes 4 HB and 3 normal livers (**Figure 5**).

* Some Excel files don't have named Sheets. Some have colors that are not described like Table S7, Table S8, "IC20 gene (negative)" (what does the blue font mean for some rows?). None of the tables explain what each column is nor do they include a caption.

We are sorry for this. We have added information. We have also classified most of the tables as Supplementary Data.

* I did not find a link to the code used for this project. You should share your code on a permanent repository like Zenodo or Figshare (both give you a DOI you can cite) or another option. You could also share it via GitHub, and well, GitHub repositories can easily be linked to Zenodo or Figshare. Code is the ultimate documentation on how an analysis was carried out and it's invaluable in order to reproduce an analysis.

The package used for single cell clustering and differential expression analysis were previous of publications from Chen Lab. Other analysis and visualization were performed with R/Bioconductor packages. The codes were documented at <https://github.com/chenlab-sj>. We included the link in the manuscript.

Here are some smaller comments and requests:

* Line 203: cite KEGG

We have removed this part after revision.

* Line 291: cite "Genetic and Chemical Perturbation database"

We have cited.

* Line 293: cite Biocarta

We have removed this part after revision.

* Would it make sense to include a supplementary file with an example excel spreadsheet that was downloaded for "complete blood counting"? I don't know if these are standard Excel files in your field that everyone knows, or if not, then it'd be useful for others to see them.

We have presented the data in Figure 2 instead of using a table.

* Include a public link for AutoMapper and version number (Line 698). I understand that it's not peer-reviewed, but it would still be useful to see this code in case someone wants to reproduce your analysis or investigate some parameters used among all the software included in AutoMapper.

Currently there is no public repository for AutoMapper pipeline that remain internally maintained. AutoMapper implements a set of public tools (Trim-Galore, STAR, RSEM, GENCODE etc.) and their versions are described in the manuscript (lines 851-854). Default parameters are used for these tools if no further declaration.

* Can you clarify the CPM 10 reads cutoff? Is that a cutoff for the mean number of reads across all samples? Or if one sample has < 10 reads, then it's dropped?

This is a cutoff for removing lowly-expressed genes. Only genes with greater than a CPM cutoff (corresponding to a count of 10 reads) in at least one sample group were kept for differential expression analysis". R code snippets are

```
cutoff_count <- 10
groupLabel<- factor(c("Tumor", "Tumor", "Tumor", "Normal", "Normal", "Normal"))
cutoff_CPM<- cpm(cutoff_count, median(colSums(counts)))
keep <- rowSums(cpm(counts) > cutoff_CPM) >= min(table(groupLabel))
```

* Cite limma, voom, STAR, Trim-Glaore, RSEM, Gencode, edgeR for the TMM method, GSEA, MSigDB, and other methods I might have missed.

We have added these references in manuscript.

* Include an equation or R formula for the model used for the bulk RNA-seq differential expression analysis (Lines 697 to 709). You could also explain why some variables were included in the model.

No additional variable was used beyond group label and batch correction was unnecessary based on our experimental design. R code snippets are

```
groupLabel <- factor(c("Tumor", "Tumor", "Tumor", "Normal", "Normal", "Normal" ),
```

```
levels = c("Tumor", "Normal")
design <- model.matrix(~ 0 + groupLabel )
```

* Line 724: what version of Cell Ranger was used? If you used versions prior to 7.0.0, did you use --include-introns?

We used Cell Ranger version 6.0.0 and we used the option which includes introns. We therefore updated the manuscript by adding the information.

* How was the 40% cutoff chosen for the ribosomal/mitochondrial genes/proteins chosen? Did you try using automatic outliers methods like `scater::isOutlier()` or equivalents?

Proportion of ribosomal/mitochondrial genes may indicate the quality of cells (for example, see Lun, McCarthy & Marioni, 2016). The cutoff was chosen to remove low quality cells by inspecting the distribution of proportions of ribosomal and mitochondrial genes across cells.

Reference: Lun AT, McCarthy DJ, Marioni JC. A step-by-step workflow for low-level analysis of single-cell RNA-seq data with Bioconductor. *F1000Res*. 2016 Aug 31;5:2122. doi: 10.12688/f1000research.9501.2. PMID: 27909575; PMCID: PMC5112579.

* How were the cutoffs for low and high UMI counts chosen?

The low and high cutoffs were chosen by inspecting the distribution of UMI across cells.

* You might want to

use <https://bioconductor.org/packages/release/bioc/html/scDblFinder.html> for detecting doublets instead of the $\geq 32,768$ UMI threshold you are currently using.

We thank the reviewer for suggesting scDblFinder for doublet detection. We used scDblFinder (version 1.12.0) as an alternative way to detect doublets and updated our manuscript respectively. Following the scDblFinder documentation on processing multiple samples, we ran scDblFinder on the count matrix and identified a doublet rate of 3%. There is no strong enrichment of doublets in any of the inferred 16 clusters, with a percentage of called doublets ranging from 0.3% to 4.7%. The clusters with high doublet rates are mostly stromal components (myeloid, endothelial) or replicating cells. Using a different doublet detection method did not affect our major results' interpretation and conclusions.

* Why did you normalize the UMI counts to 10,000 UMIs per cell?

It is an arbitrary scale factor, as also has been set as the default scale factor in popular softwares like Seurat (version 4.3.0).

* What were the parameters used for the tSNE and what software/method was used to compute it? Without that information the tSNE described is not reproducible.

For tSNE plots, we used R package Rtsne (version 0.16). We used the function Rtsne with its default parameters. We updated the manuscript by providing the information on parameters used.

* What parameters were used for the differential expression analysis with <https://doi.org/10.1186/s13059-018-1438-9?>

We performed scRNA-seq DE analysis using edgeR and adjusted for latent batch effects using SVA (PMID: 32322368). Specifically, we used scran (PMID: 27122128) to extract the normalization factor. For edgeR, we set prior.df to zero to independently infer each gene's dispersion based on scRNA-seq data and used the likelihood ratio based test. For the application of SVA, we first sorted cells by total UMI within each batch and then summed 20 cells into a new aggregated pseudo-cell. Then SVA was applied with ten iterations to extract the top 20 surrogate variables representing the latent batch effects. The method was implemented in the function DEAdjustForBatch in the NBID package (<https://bitbucket.org/Wenan/nbid/src/master/R/DEAdjustForBatch.R>). We updated the manuscript by providing the information on parameters used.

* Line 775: what version of Seurat was used?

We used Seurat, version 4.3.0. We updated the manuscript by providing the information on parameters used.

* Line 779: what version of ggplot2 was used?

We used ggplot2, version 3.4.0. We updated the manuscript by providing the information on parameters used.

* Line 776: what is the "Seurat's dimension reduction" functionality? Can you be more precise and mention the specific function used as Seurat has several options.

We used Seurat's RunPCA function with its default parameters. We updated the manuscript by providing the information on parameters used.

* Cite Seurat and ggplot2.

We added citation to Seurat and ggplot2.

* <https://www.ncbi.nlm.nih.gov/geo/query/acc.cgi?acc=GSE195575> does not include the high resolution images. For example at <https://www.ncbi.nlm.nih.gov/geo/query/acc.cgi?acc=GSM5840728> there is a 18.1 Mb JPG file, which is not useful for methods that require access to the high quality images.

We have submitted all raw data.

* How was Figure 3B generated (the heatmap)? In general, the figure captions for the

heatmaps don't mention what we are looking at: normalized log counts? raw counts? Some seem to be Z scores, like Figure 2E.

We have removed this figure according R1's suggestion.

* Can you comment on the 3 outlier samples in Figure 2E and Figure 3B? I see 2 blue and 1 red mixed with the other color. Should those samples be excluded or is there some reason why you are still including them in the analyses? Are those 3 samples outliers when you visualize the top principal components or MDS for this data?

We thank this reviewer to pointed out this odd. We also noticed this, but we have no idea why. We used the online program R2: Genomics Analysis and Visualization Platform to generate the heatmap. Now we have moved the Figure 2E as Supplementary Figure 2A and replaced Figure 2 panels with a more powerful and robust analysis.

* Why did you change the colors for Figure 4E and 4F? Blue used to be normal, now it's hepatoblastoma. This is confusing. This also affects Figure S2 panel D and Figure S3 as the colors for normal hepatoblastoma or heptoblastoma (is that a typo in Figure S3's color legend?) are not consistent.

We are sorry for the confusion. We now have replaced the whole figure with a new one since we have expanded the experiment by including more samples.

* Figure 4C: I can't see yellow although it's described in the caption. I also can't differentiate the light and dark blue. I highly recommend choosing other colors and to use colors that are color blind friendly. If you have ggplot2 objects, you could use <https://github.com/clauswilke/colorblindr> for example.

* Figure 4D and G: what are we looking at? log counts? counts?

We now have replaced the whole figure with a new one since we have expanded the experiment by including more samples.

* Figure S4 B: we can't see the B-cells with that color scale.

We have removed these data and because we have expanded the experiment by including more samples.

Here are some typos I noted:

* Page 8 line 156: "associated downregulated" -> "associated with downregulated"

* Page 8 line 165: "levels" -> "level"

* Page 9 line 170: "caner" -> "cancer"

* Line 653: "manufacture's" -> "manufacturer's"

* Line 699: "firs" -> "first"

We are sorry for these typos. We have corrected them all.

Reviewers' Comments:

Reviewer #4:

Remarks to the Author:

Fang et al described a liver-specific MYC-driven hepatoblastoma (HB) murine model (ABC-Myc mice) with pathological and transcriptomic analyses and revealed signaling pathways in this tumor in the comparison with human HB. And they described the CRISPRCas9 screening to identify candidate genes and genes correlating with the efficacy of doxorubicin chemotherapy. This tumor model seems to be an exceptional tool for future studies in the HB field. This paper contains a lot of data from mice model to the clinical application of this model.

As mentioned by the Reviewer #1, this tumor should be called as "HB-like tumors" because MYC activation is not a common event in human HB and this pathway is special signaling in HB.

And as mentioned by the Reviewer #1, the authors should describe the existing of the heterogeneity among in tumor nodules of the same mice simultaneously. Clinically, the most important point is the presence of heterogeneity in the tumor. The authors explained the development of tumor in mice during ontogenesis. They should describe the existence of simultaneous heterogeneity of these tumor. As described by the response of the reviewers' comments, the authors showed the IHC figures by several antibodies. Therefore, the authors should show us the heterogeneity of tumor content in this model.

In addition, in high risk hepatoblastoma in human, distant metastasis, especially lung metastasis, often occurs. How many distant metastases did this HB-like tumors occur? If not, this model is considered as a special model for HB which represents multifocal aggressive tumors in liver without metastasis.

The authors explained that this mice model is C2 morphologic phenotype as described by Cairo et al. which has aggressive biological behavior and an unfavorable prognosis. The authors should identify the possibility that this model represents HCN-NOS or pediatric HCC. In the clinical chemistry data (Figure 2), the levels of AFP were not so high as the AFP data of human HB and other abnormal data are considered as the result of tumor progression in liver. These data are similar to those of HCC and HCN-NOS.

As mentioned by the previous review, it is amazing that the ABC-Myc mice generate tumors without activating Wnt/beta-catenin pathway because almost all human HBs have this pathway activation due to the mutation/deletion of CTNNB1 gene or mutation of APC genes.

As MYC is one of the target genes of this Wnt/beta-catenin pathway, the authors should clarify the correlation between MYC activation and Wnt/beta-catenin pathway in this model and then explain the probability of this ABC-Myc mice generate tumors as a model of human HB. The IHC of beta-catenin is usual detected at cell membrane and when beta-catenin is accumulated due to inhibition of degradation by the exon 3 mutation/deletion it stained in nuclei and cytoplasm. If authors show the strong

-GSEA analysis (figures 2B, C, D and F): As mentioned by the Reviewer 1, the authors have to use the other resources from HB samples. In the revised manuscript, the authors included the gene sets from Cario paper (Cancer Cell, 2008). The data of this gene sets consisted of HB tissue RNA samples including those resected after preoperative chemotherapy. The authors should use the gene expression data of HB obtained before any treatments. (f.e.: Nagae, G et al. Nature Commun. 2021)

The revised paper was added the data from the reviewer's comment "This analysis would be improved if the authors provide the % of the overlapping genes altered in human HB vs non-tumors and MYC-ABC tumors vs control livers at a specific FDR." This VENN diagrams shows 50.1% and 42.5% of the up- and down-regulated genes in the MYC-ABC tumors vs. control liver (CL) samples were also deregulated in human HB as compared with non-tumor (NL) samples using 11,393 ortholog genes. respectively. This result clearly supports the high similarity (>50%) of the MYC-ABC tumor model with human HB. However, the gene numbers used in these VENN diagram are too many to be evaluated the similarity because approximately half of ortholog genes were included in VENN diagram. The authors should be more restricted genes (f.e. FDR <0.01 or less) to be used this analysis. In addition, PCA plot of the RNAseq database might be nonsense because the mouse liver and tumor samples were obtained from same clones but other human samples were not. Therefore, it is natural that PCA of these mouse samples were concentrated in the small area and other human samples were scattered.

As mentioned by the previous comments, the analysis of scRNA and spatial transcriptomics remains very poor, because these analyses were performed in only 3 or 4 samples. As shown in the Figures 4 and 5, the authors described "heterogeneity of hepatoblastoma-like cells" which is inappropriate expression, because these data were derived from tumor samples not cells. These data only represent many kinds of cells which form the tumor samples. Therefore, these data do not show the heterogeneity of tumor cells. The authors should reconsider whether to add these data or not in this manuscript.

The date of spatial transcriptomics are fine but these data do not show the heterogeneity of tumor cells.

- Genome-wide CRISPR screen: The data of this screen method is impressive. Temsirolimus, mTOR inhibitor has been used in clinical trial for high risk hepatoblastoma. If possible, the authors added the clinical outcome of this trial.

And as you know, cisplatin is the first chemotherapeutic agent for hepatoblastoma and recent clinical trials have focused into dose escalation of cisplatin. In this screen method, the data of cisplatin efficacy for these tumor cell should be added.

PRKDC inhibition enhances efficacy of chemotherapy. This part of the study is also highly interesting. The mechanism of the efficacy of this combination with DOR might be discussed. The recent papers reported that chromosomal instability, TRET promoter mutations or telomerase activation were correlated with poor outcomes of the HB patients. The PRKDC inhibitors might be effective into these tumors.

Reviewer #5:

Remarks to the Author:

The authors have significantly improved the manuscript in the revision. In particular, the authors performed scRNA-seq and spatial transcriptomics in 4 additional mouse tumors and 3 normal livers. The CRISPR screening data in additional two mouse cell lines strengthened the conclusion of this study. The authors have also adopted reviewer's suggestion to use the term "HB-like tumors" for Myc-driven HBs. Most of reviewers' questions have been adequately addressed.

Minor points:

- A VENN diagram of negative or positive selection genes between human Huh6 and one of the mouse HB cell line will help compare CRISPR screen hits in human vs mouse cells.
- Fig.6b,c, the genes are mouse but the names indicate human genes. For example, TRP53 should be Trp53.

Reviewer #6:

Remarks to the Author:

The authors have addressed most of questions raised by Reviewer #3, while there are still some questions remained or further introduced by including more samples.

1. The three GEO accession links still don't show any FASTQ files.
2. In the revision, four tumor samples and three normal controls were used for scRNA-seq. Did the author perform read depth normalization using cellranger aggr function to avoid artifact introduced by sequencing depth?
3. The authors performed analysis for both low-quality and high-quality cells considering UMI counts and mitochondrial genes. Only one cluster of cells with > 50% low-quality cells was removed from further analysis. If all low-quality cells are removed at the beginning (which is the way people usually do in scRNA-seq data analysis), do the results look similar?
4. Lines 387-391, the observation of tumor cells with high level of erythroid genes is interesting. Is that due to contamination during single-cell library preparation?

RESPONSES TO REVIEWER COMMENTS

We greatly appreciate the time and constructive comments/suggestions from all three reviewers. We now have addressed or clarified all comments point-by-point.

Reviewer #4 (Remarks to the Author):

Fang et al described a liver-specific MYC-driven hepatoblastoma (HB) murine model (ABC-Myc mice) with pathological and transcriptomic analyses and revealed signaling pathways in this tumor in the comparison with human HB. And they described the CRISPRCas9 screening to identify candidate genes and genes correlating with the efficacy of doxorubicin chemotherapy. This tumor model seems to be an exceptional tool for future studies in the HB field. This paper contains a lot of data from mice model to the clinical application of this model.

Response: We highly thank this reviewer for his/her positive comments on our study.

As mentioned by the Reviewer #1, this tumor should be called as “HB-like tumors” because MYC activation is not a common event in human HB and this pathway is special signaling in HB.

Response: As both reviewers suggested, we have called the model “HB-like tumors” throughout the manuscript.

And as mentioned by the Reviewer #1, the authors should describe the existing of the heterogeneity among in tumor nodules of the same mice simultaneously. Clinically, the most important point is the presence of heterogeneity in the tumor. The authors explained the development of tumor in mice during ontogenesis. They should describe the existence of simultaneous heterogeneity of these tumor. As described by the response of the reviewers’ comments, the authors showed the IHC figures by several antibodies. Therefore, the authors should show us the heterogeneity of tumor content in this model.

Response: This model demonstrates histologic diversity. As this reviewer knows, HB consists of multiple different histologic patterns that may occur in pure form but most commonly exist in various proportions within the same tumor; however, currently the prognostic significance of the presence of or relative proportions of different morphologic patterns remains unclear; histology can be variable within a single tumor and limited sampling may restrict the ability to observe the variations in morphologic patterns that can be observed within one tumor. One exception is that the presence of neoplastic cells with a small cell undifferentiated morphology is associated with poor survival. However, small cell undifferentiated morphology was not seen in this model. In addition, INI-1 staining was retained in all sampled HB-like tumors, indicating that the diagnosis of an INI1-lost rhabdoid-like tumor is not appropriate for this model.

Therefore, here we focused on heterogeneity within one tumor. We have included a revised figure of HE images of murine tumor histology side by side with human tumor histology to show comparisons and differences to better show the histologic phenotypes and how those phenotypes correlate with tumor samples that are fetal, embryonal, mixed, or HCN-NOS (please see **Fig. 1f, 1g**).

In addition, in high risk hepatoblastoma in human, distant metastasis, especially lung metastasis, often occurs. How many distant metastases did this HB-like tumors occur? If not, this model is considered as a special model for HB which represents multifocal aggressive tumors in liver without metastasis.

Response: Neither regionally invasive nor metastatic disease is a feature of this model, and this biological behavior is consistent with the known role for MYC to drive bulky tumor growth within the liver microenvironment (Wang, H., Lu, J., Edmunds, L.R., Kulkarni, S., Dolezal, J., Tao, J., Ranganathan, S., Jackson, L., Fromherz, M., Beer-Stolz, D., et al. (2016). Coordinated Activities of Multiple Myc-dependent and Myc-independent Biosynthetic Pathways in Hepatoblastoma. *J Biol Chem* 291, 26241-26251. 10.1074/jbc.M116.754218.). Metastatic disease is not observed in this model from several reasons including genetics or reduced survival time from localized disease. Alternatively, it may be an extremely rare event in the model. In any of these scenarios it is reasonable to hypothesize that additional genetic or non-genetic drivers are probably important for the invasive and metastatic potential in this model and from what is known in the literature about other HB model systems and pediatric HBs with demonstrated metastatic potential. Therefore, this model is considered as a special model for hepatoblastoma which represents multifocal aggressive tumors in liver without metastasis, as this reviewer suggested. We have included this explanation in Discussion.

The authors explained that this mice model is C2 morphologic phenotype as described by Cairo et al. which has aggressive biological behavior and an unfavorable prognosis. The authors should identify the possibility that this model represents HCN-NOS or pediatric HCC. In the clinical chemistry data (Figure 2), the levels of AFP were not so high as the AFP data of human HB and other abnormal data are considered as the result of tumor progression in liver. These data are similar to those of HCC and HCN-NOS.

Response: We highly thank this reviewer for this important question. We totally agree with this reviewer that the AFP levels in serum were not a high as observed in human HB. Interestingly, the data from bulk RNA-seq and scRNA-seq as well as the spatial transcriptomics showed that *Afp* was one of the top genes highly expressed in tumors. Nevertheless, we have re-examined the HE to seek the HCN-NOS or HCC components. Indeed, we do see some tumor areas bear HCN-NOS components (please see the revised **Fig. 1f, 1g**) since the MYC-driven murine HB-like tumors demonstrate phenotypic plasticity of hepatocyte lineage committed stem/progenitor cells. We agree with the reviewer that HCN-NOS may be a better way to describe the phenotypic plasticity of the model, keeping in mind that this is currently a provisional entity often applied to cases where a consensus diagnosis cannot be reached.

Now, we have added one paragraph in Results to describe this observation by saying:

“The Myc-driven murine hepatoblastoma-like tumors demonstrate phenotypic plasticity of hepatocyte lineage committed stem/progenitor cells. While the co-existence of embryonic and fetal histological features of ABC-Myc tumors resemble the human hepatoblastoma (**Fig. 1f**), it is important to differentiate hepatoblastoma from hepatocellular carcinoma in pediatric patients, because of differing treatment and prognosis. While the poorly differentiated histology is consistent with the pediatric C2 phenotype, some tumor areas also contain histologic features of the subclassification of pediatric hepatoblastomas with hepatocellular carcinoma features that were previously called transitional liver cell tumors (TLCT) (**Fig. 1g**), indicating that some ABC-Myc tumor cells have features of HCN-NOS (Hepatocellular Malignant Neoplasm, Not Otherwise Specified) that frequently presents phenotypic plasticity.”

We have also made an extensive discussion in our revised MS for pathological features of this model by saying:

“the poorly differentiated histology is consistent with the pediatric C2 phenotype. Nevertheless, this murine model also contains histologic features of the subclassification of pediatric hepatoblastomas with hepatocellular carcinoma features that were previously called transitional liver cell tumors (TLCT). The phenotypic plasticity that is observed in this Myc-driven murine model of hepatoblastoma is documented in pediatric hepatoblastoma where some hepatoblastomas can be classed into the transcriptomic subgroup “liver progenitor” differentiation state that appears to correlate with the C2A molecular for human hepatoblastomas. This differentiation state is highly proliferative, immune cold, composed of embryonal histologies, enriched for self-renewal and pluripotency transcription factors including MYCN and may represent a model for relapse or hepatoblastomas with metastatic potential, which fits with the histology and behavior of ABC-Myc hepatoblastoma-like tumors. The phenotype and molecular characterization of our model may also match with the C2A phenotype of which HCN-NOS can be included as well as a “liver progenitor” subgrouping, and which also has been correlated with high-risk MRS-3B subgrouping.

Currently the prognostic significance of the presence of or relative proportions of different morphologic patterns that may arise in pediatric and adolescent hepatoblastomas remains unclear and there is still variability in the subclassifications of pediatric liver tumors because histology can be variable within a liver tumor and sampling may limit the ability to observe the variations in morphologic patterning that can be observed within one tumor. One exception is the presence of foci of neoplastic cells having a small-cell-undifferentiated morphology, which is not observed in this model. Additionally, INI-1 staining was retained in all sampled hepatoblastoma-like tumors indicating that the diagnosis of rhabdoid-like tumor is not appropriate for this model. In pediatric patients it is important to differentiate hepatoblastoma from hepatocellular carcinoma because of treatment and prognosis. The Myc-driven murine hepatoblastoma-like tumors demonstrate phenotypic plasticity of hepatocyte lineage committed stem/progenitor cells, suggesting that some tumor components have HCN-NOS features.”

As mentioned by the previous review, it is amazing that the ABC-Myc mice generate tumors without activating Wnt/beta-catenin pathway because almost all human HBs have this pathway activation due to the mutation/deletion of CTNNB1 gene or mutation of APC genes.

As MYC is one of the target genes of this Wnt/beta-catenin pathway, the authors should clarify the correlation between MYC activation and Wnt/beta-catenin pathway in this model and then explain the probability of this ABC-Myc mice generate tumors as a model of human HB. The IHC of beta-catenin is usual detected at cell membrane and when beta-catenin is accumulated due to inhibition of degradation by the exon 3 mutation/deletion it stained in nuclei and cytoplasm. If authors show the strong (this reviewer stopped here)

Response: We thank this reviewer for the comment regarding to the relationship of MYC and CTNNB1. Now, we have added one paragraph to discuss their relationship by saying:

“Mutation of CTNNB1 occurs in about 48-67% of pediatric HB cases, which is different from how liver tumors arise in our model. MYC overexpression is a significant genetic event in pediatric liver tumors including pediatric HCCs. In humans mutational and immunohistochemical analyses of β -catenin are not always correlative. Nuclear localization/expression of β -catenin in human HBs is not always diffuse and can be regional or focal and is an imperfect surrogate for molecular testing. The fact that nuclear localization of β -catenin does not have to be diffuse in Wnt-driven human pediatric liver tumors is of interest and points to subpopulation heterogeneity that drives aspects of tumor initiation, growth, and biological aggressiveness in human pediatric HBs and HCCs making our model very

useful for understanding how tumor heterogeneity correlates with biological aggressiveness. Interestingly the IHC staining pattern for β -catenin in our model is localized to the membrane and cytoplasm of all HB-like tumors and is comparable to staining patterns that have been observed in pediatric tumors classified as HBs with HCC features including HCN-NOS and HB-FPA. Our model may represent locally aggressive, spontaneously arising HB cases where a mutation driving constitutive WNT/ β -catenin signaling is absent. Therefore, lack of strong nuclear translocation of the wild-type β -catenin may suggest that MYC overactivation relieves the selective pressure on β -catenin, since MYC is a key downstream effector of the Wnt- β -catenin pathway.”

-GSEA analysis (figures 2B, C, D and F): As mentioned by the Reviewer 1, the authors have to use the other resources from HB samples. In the revised manuscript, the authors included the gene sets from Cario paper (Cancer Cell, 2008). The data of this gene sets consisted of HB tissue RNA samples including those resected after preoperative chemotherapy. The authors should use the gene expression data of HB obtained before any treatments. (f.e.: Nagae, G et al. Nature Commun. 2021)

Response: We thank this reviewer for this insightful suggestion. In our 1st round revision, we included Cario gene sets in response to the request from reviewer 1. Actually, we also included several other HB gene sets in our study (Fig. 3 and Fig. S3). The results from each geneset all support our conclusion. We agree with this reviewer, Cario gene sets consisted of HB tissue RNA samples including those resected after preoperative chemotherapy. Unfortunately, the datasets published by Nagae are under restricted access (please see their declaration in their publication, and the link to dataset). Fortunately, this group performed Affymetrix microarray analysis for 53 hepatoblastoma tissues (prior to any chemotherapy) and 14 noncancerous liver tissue samples (<https://www.ncbi.nlm.nih.gov/geo/query/acc.cgi?acc=GSE131329>). We therefore used this dataset to compare with our murine model by GSEA. Again, we obtained very similar results, and included in our Results by saying:

“Since Cario gene sets consisted of hepatoblastoma tissue RNA samples including those resected after preoperative chemotherapy, we compared ABC-Myc gene expression with the gene datasets generated from biopsy or surgery prior to any chemotherapy (Ikeda dataset, GSE131329), which included 14 noncancerous liver tissues and 53 tumor tissues. We used the top 200 differentially expressed genes from Ikeda genset for GSEA analysis, and again, we obtained very similar results (Fig. S3a), which further strengthened our conclusion.”

The revised paper was added the data from the reviewer’s comment “This analysis would be improved if the authors provide the % of the overlapping genes altered in human HB vs non-tumors and MYC-ABC tumors vs control livers at a specific FDR.” This VENN diagrams shows 50.1% and 42.5% of the up- and down-regulated genes in the MYC-ABC tumors vs. control liver (CL) samples were also deregulated in human HB as compared with non-tumor (NL) samples using 11,393 ortholog genes. respectively. This result clearly supports the high similarity (>50%) of the MYC-ABC tumor model with human HB. However, the gene numbers used in these VENN diagram are too many to be evaluated the similarity because approximately half of ortholog genes were included in VENN diagram. The authors should be more restricted genes (f.e. FDR < 0.01 or less) to be used this analysis.

Response: Oncogenic transformation by a master transcription factor like MYC usually leads to genome-wide transcriptomics reprogramming. It is not surprising that many genes exhibit expression changes in cancer cells in comparison with the differentiated normal cells. Actually, in the Nagae, G

et al. study (Nature Commun. 2021), they selected the 5000 most variable genes for analysis (please see Method in their publication), with the number similar to what we obtained.

As suggested by the reviewer, we repeated the VENN diagram using a more restrictive gene list ($FDR < 0.01$). The results are similar to those previously obtained (Fig. 3d). Please note that the upregulated $p = 1.6 \times 10^{-96}$, downregulated $p = 2.1 \times 10^{-153}$ in our original VENN analysis (Fig. 3d). Therefore, we believe the $FDR < 0.05$ threshold was already stringent enough. We worried that if taking $FDR < 0.01$ as a threshold, we may filter out the important bona fide liver cancer related genes. We decided to keep the $FDR < 0.05$.

In addition, PCA plot of the RNAseq database might be nonsense because the mouse liver and tumor samples were obtained from same clones but other human samples were not. Therefore, it is natural that PCA of these mouse samples were concentrated in the small area and other human samples were scattered.

Response: We appreciated that this reviewer has noticed that mouse samples were concentrated in the small area while human samples were more scattered. Our purpose of PCA analysis is to demonstrate that mouse liver tumors were grouped with human HB samples, while mouse normal livers were grouped with human normal livers. It is true that mouse samples have less heterogeneity and are concentrated in a small area because they came from the same mouse strain whereas patient samples have much more heterogeneity because all of them have different genetic backgrounds.

As mentioned by the previous comments, the analysis of scRNA and spatial transcriptomics remains very poor, because these analyses were performed in only 3 or 4 samples. As shown in the Figures 4 and 5, the authors described “heterogeneity of hepatoblastoma-like cells” which is inappropriate expression, because these data were derived from tumor samples not cells. These data only represent many kinds of cells which form the tumor samples. Therefore, these data do not show the heterogeneity of tumor cells. The authors should reconsider whether to add these data or not in this manuscript.

Response: In response to the previous critiques from reviewer 1, we included 4 additional tumor samples and 3 normal liver samples for scRNA and Spatial transcriptomics analyses in our 1st round revision. On the financial end, scRNA and Spatial experiments are expensive (estimated at more than \$7,000 for each sample), and therefore it is challenging to include additional samples. Moreover, the statistical analysis from these samples is highly robust. Clear evidence of strong tumor heterogeneity at both inter-tumoral and intra-tumoral levels has been documented in Figure 4 and 5 through our scRNA-seq analysis and spatial transcriptomics analysis. We offer our sincere apologies if our initial presentation in Figures 4 and 5 has not clearly demonstrated the level of tumor heterogeneities. To enhance the presentation of tumor heterogeneity, we have included additional data analysis as described below, as **supplementary figure 5**. We included this in Results by saying:

“To demonstrate the heterogeneity of tumor cells, we have highlighted the tumor cell specific clusters (Cluster 2, 3, 7, 9, 12, 16) in each tumor sample (**Fig. S5a**). The results showed that each tumor consisted of these clusters with different percentages (**Fig. S5b**), which demonstrated both intra-tumoral and inter-tumoral heterogeneity. For example, NEJ723 and NEJ634 were dominated by Clusters 7 and 16, respectively; while NEJ709 and NEJ687 showed multiple tumor clusters co-existed

at substantial fractions. To further characterize the tumor cell heterogeneity, we only focused on the clusters 7 and 16, which expressed highest levels of *Afp* and *Igf2* and thus these clusters presumably represent bona fide tumor cells. We were able to partition these strong *Afp*+*Igf2*+ clusters 7 and 16 into several subclusters (Fig. S5c). For each of them, we found significant variation in sub-cluster proportion across tumor cells ($P < 0.0005$). In addition, we determined the composition of different types of tumor cells in each tumor sample by mapping ABC-Myc tumor cells with the annotated human hepatoblastoma cluster genes. Again, the murine tumors showed intra-tumoral heterogeneity (different tumor classes in individual tumors) and inter-tumoral heterogeneity (different composition of various tumor cell types) (Fig. S5d). There is a significant variation of proportion of tumor cell types among the four samples (Chi square test: $P = 0.0005$.)”

The date of spatial transcriptomics are fine but these data do not show the heterogeneity of tumor cells.

Response: Again, we apologize if we did not explain the data clearly. The spatial transcriptomics demonstrated the heterogeneity of tumor cells. For example, the *Afp* and *Igf2* expression in NEJ634 was nearly uniformly expressed across the whole tissue section; however, the expression of stem cell markers (*Dlk*, *Epcam* and *Gpc3*) were heterogeneous (Fig. 5a). Now, we have added one sentence by saying:

“The intra-tumor heterogeneity was also demonstrated by spatial transcriptomics analysis. For example, the *Afp* and *Igf2* expression in NEJ634 was nearly uniformly expressed across the whole tissue section; however, the expression of stem cell markers (*Dlk*, *Epcam* and *Gpc3*) were heterogeneous (Fig. 5a).”

We hope this reviewer can agree with us at this point.

- Genome-wide CRISPR screen: The data of this screen method is impressive. Temsirolimus, mTOR inhibitor has been used in clinical trial for high risk hepatoblastoma. If possible, the authors added the clinical outcome of this trial.

Response: We highly appreciate for this reviewer's positive comment on our CRISPR screen. We agree with this reviewer that mTOR is interesting. The following clinical trial resulting from this study recently reported that 10 patients with liver malignancy received everolimus after liver transplantation, and none of these patients developed recurrence by the endpoint of the study (*Children* **2023**, *10*(2), 367). We have included this in our revision.

And as you know, cisplatin is the first chemotherapeutic agent for hepatoblastoma and recent clinical trials have focused into dose escalation of cisplatin. In this screen method, the data of cisplatin efficacy for these tumor cell should be added.

Response: We highly thank this reviewer for this insightful suggestion. In this study, we used doxorubicin as an anchor, for genome-wide CRISPR screen in 3 ABC-MYC lines. As this reviewer knows, doxorubicin is the second commonly used agent in HB treatment. We totally agree with this reviewer that it would be interesting to have such information for cisplatin in large scale genome-wide studies as performed for doxorubicin. As this reviewer may understand, such studies take long time and large effort to accomplish. We have made plans to screen genetic modulators of cisplatin using our established tools. Hopefully, we could report our results to this reviewer in the near future.

PRKDC inhibition enhances efficacy of chemotherapy. This part of the study is also highly interesting. The mechanism of the efficacy of this combination with DOR might be discussed. The recent papers reported that chromosomal instability, TRET promoter mutations or telomerase activation were correlated with poor outcomes of the HB patients. The PRKDC inhibitors might be effective into these tumors.

Response: Again, we thank this reviewer for this thoughtful comment. Now, we have added one sentence in Discussion by saying:

"Recent studies have shown that chromosomal instability, TERT promoter mutations or telomerase activation were correlated with poor outcomes of the HB patients, suggesting that tumors in these patients may have DNA repair defects and thus, PRKDC inhibitors may be effective in these tumors."

Reviewer #5 (Remarks to the Author):

The authors have significantly improved the manuscript in the revision. In particular, the authors performed scRNA-seq and spatial transcriptomics in 4 additional mouse tumors and 3 normal livers. The CRISPR screening data in additional two mouse cell lines strengthened the conclusion of this study. The authors have also adopted reviewer's suggestion to use the term "HB-like tumors" for Myc-driven HBs. Most of reviewers' questions have been adequately addressed.

Response: We highly thank this reviewer for his/her positive comments on our revision efforts.

Minor points:

- A VENN diagram of negative or positive selection genes between human Huh6 and one of the mouse HB cell line will help compare CRISPR screen hits in human vs mouse cells.

Response: we have particularly added VENN analysis for Huh6 and NEJF10. We included the results by saying:

"We particularly compared the Huh6 with NEJF10 cell line by VENN analysis. 61.7% of the essential genes in Huh6 cells were shared by NEJF10 (Fig. S9e). Among the tumor suppressors, we found five (NF2, PTEN, PAWR, RASA2, STK40) were commonly shared between NEJF10 and HuH6 cells (Fig. S9f)."

- Fig.6b,c, the genes are mouse but the names indicate human genes. For example, TRP53 should be Trp53.

Response: We thank this reviewer for correcting this. We have changed in now in Figure 6b and Figure 7b.

Reviewer #6 (Remarks to the Author):

The authors have addressed most of questions raised by Reviewer #3, while there are still some questions remained or further introduced by including more samples.

Response: We highly thank this reviewer for his/her positive comments on our revision efforts.

1. The three GEO accession links still don't show any FASTQ files.

Sample type: RNA

Source name: liver tissue
Organism: Mus musculus
Characteristics: tissue: liver
disease state: hepatoblastoma murine model
Extracted molecule: polyA RNA
Extraction protocol: cDNA and Library was prepared following 3' Single cell Expression V3 from 10X Genomics.

Library strategy: RNA-Seq
Library source: transcriptomic single cell
Library selection: cDNA
Instrument model: Illumina NovaSeq 6000

Description: tissue-NEJ638
Single Cells_NEJ638
Single cells was captured and cDNA processed by 3' Single Gene expression kit V3.1- 10X Genomics

Data processing: Cell Ranger 6.0 used to generate Fastq, end count and matrix files that used further analyzes.

Submission date: Jan 25, 2023
Last update date: Jan 25, 2023
Contact name: Jun Yang
Organization name: St.Jude Children's Research Hospital
Department: Surgery
Street address: 262 Danny Thomas Place
City: Memphis
State/province: TN
ZIP/Postal code: 38103
Country: USA

Platform ID: GPL24247
Series (1): GSE223689 Mousemodel_Heptablastoma_ScRNAseq

Relations
BioSample: SAMN32909076
SRA: SRX1916657

External data have been provided but are not accessible for review while status is private.

Supplementary file	Size	Download	File type/resource
GSM6973187_Sc_NEJ638_barcodes.tsv.gz	6.4 Mb	(http)	TSV
GSM6973187_Sc_NEJ638_features.tsv.gz	254.1 Kb	(http)	TSV
GSM6973187_Sc_NEJ638_matrix.mtx.gz	66.0 Mb	(http)	MTX

SRA Run Selector ⓘ

Raw data are available in SRA
Processed data provided as supplementary file

Response: We thank the reviewer for this question and now we have figured it out for the reason. We submitted FASTQ files to GEO, and GEO then put the data to SRA. Unfortunately, the SRA files are not accessible to public until the time of data publication.

The following screenshot from GEO website shows that SRA accession along with GEO accession. Under the highlighted SRA number, there is an italic sentence line explaining the file is not accessible.

To see if we can solve this issue, Dr. Natarajan, the co-first author of this paper, reached out to SRA and got the feedback from SRA (please see the thread of emails below). We have asked SRA to generate a reviewer metadata link for the confirmation, which is included in the email below from SRA. Nevertheless, the reviewers are still unable to access the sequencing data (including us).

From: NLM Support <nlm-support@nlm.nih.gov>

Sent: Friday, April 14, 2023 2:49 PM

To: Natarajan, Sivaraman <Sivaraman.Natarajan@STJUDE.ORG>

Subject: RE: [EXTERNAL] RE: case #CAS-1094522-N7F3V0: Reg: Reviewer access to Accession SRA SRX13951112... TRACKING:000337000011772

Dear Siva,
I have included metadata links for these 2 studies below. Please let us know if you have any questions.
Best,

Jon Trow, Ph.D [C]
SRA Curator, Information Engineering Branch, NCBI/NLM/NIH

Reviewer / collaborator link to metadata:

ftp://ftp-trace.ncbi.nlm.nih.gov/sra/review/SRP419035_20230414_154811_901a3c0401285faf93efb3928de9949e

NOTE: The above URL is valid for a minimum of 3 months, but may be removed any time thereafter.

If you require access to the metadata after 3 months, please email sra@ncbi.nlm.nih.gov for an updated link.

Study SRP419035: GEO accession GSE223689 is currently private and is scheduled to be released on Jan 01, 2025.

Sample SRS16578999: NEJ634

Experiment SRX19166656: GSM6973186: NEJ634; Mus musculus; RNA-Seq

Run SRR23218885 with 32733995805 bases and 275075595 spots

Run SRR23218886 with 31458171934 bases and 264354386 spots

Sample SRS16579000: NEJ638

Experiment SRX19166657: GSM6973187: NEJ638; Mus musculus; RNA-Seq

Run SRR23218883 with 21301312256 bases and 179002624 spots

Run SRR23218884 with 20635445138 bases and 173407102 spots

Sample SRS16579001: NEJ709

Experiment SRX19166658: GSM6973188: NEJ709; Mus musculus; RNA-Seq

Run SRR23218881 with 27229570361 bases and 228819919 spots

Run SRR23218882 with 26974065223 bases and 226672817 spots

Sample SRS16579002: NEJ723

Experiment SRX19166659: GSM6973189: NEJ723; Mus musculus; RNA-Seq

Run SRR23218879 with 26715533677 bases and 224500283 spots

Run SRR23218880 with 26400142741 bases and 221849939 spots

Sample SRS16579003: NEJ654

Experiment SRX19166660: GSM6973191: NEJ654; Mus musculus; RNA-Seq

Run SRR23218877 with 15145684897 bases and 127274663 spots

Run SRR23218878 with 15464171142 bases and 129951018 spots

Sample SRS16579004: NEJ677

Experiment SRX19166661: GSM6973192: NEJ677; Mus musculus; RNA-Seq

Run SRR23218875 with 18202644481 bases and 152963399 spots

Run SRR23218876 with 18597398183 bases and 156280657 spots

Sample SRS16579005: NEJ687

Experiment SRX19166662: GSM6973193: NEJ687; Mus musculus; RNA-Seq

Run SRR23218873 with 11939831085 bases and 100334715 spots

Run SRR23218874 with 12252363998 bases and 102961042 spots

Reviewer / collaborator link to metadata:

ftp://ftp-trace.ncbi.nlm.nih.gov/sra/review/SRP357078_20230414_154734_6d18ce17d10bf194639d05e84940729c

NOTE: The above URL is valid for a minimum of 3 months, but may be removed any time thereafter.

If you require access to the metadata after 3 months, please email sra@ncbi.nlm.nih.gov for an updated link.

Study SRP357078: GEO accession GSE195575 is currently private and is scheduled to be released on Jan 31, 2025.

Sample SRS11792063: NEJ146-A

Experiment SRX13951112: GSM5840728: NEJ146-A; Mus musculus; OTHER

Run SRR17788913 with 15998860088 bases and 108100406 spots

Run SRR17788914 with 16059258444 bases and 108508503 spots

Sample SRS11792064: NEJ146-B

Experiment SRX13951113: GSM5840729: NEJ146-B; Mus musculus; OTHER

Run SRR17788911 with 16349412444 bases and 110469003 spots

Run SRR17788912 with 16449216836 bases and 111143357 spots

Sample SRS11792065: NEJ146-D

Experiment SRX13951115: GSM5840731: NEJ146-D; Mus musculus; OTHER

Run SRR17788907 with 14819157712 bases and 100129444 spots

Run SRR17788908 with 14868526664 bases and 100463018 spots

Sample SRS11792066: NEJ146-C

Experiment SRX13951114: GSM5840730: NEJ146-C; Mus musculus; OTHER

Run SRR17788909 with 19404440580 bases and 131111085 spots

Run SRR17788910 with 19475813284 bases and 131593333 spots

Sample SRS16639056: NEJ634_A

Experiment SRX19234615: GSM7016921: NEJ634_A; Mus musculus; OTHER

Run SRR23291368 with 15884348278 bases and 134613121 spots

Run SRR23291369 with 15384774510 bases and 130379445 spots

Sample SRS16639057: NEJ638_C

Experiment SRX19234616: GSM7016922: NEJ638_C; Mus musculus; OTHER

Run SRR23291366 with 13989680452 bases and 118556614 spots

Run SRR23291367 with 13631439178 bases and 115520671 spots

Sample SRS16639058: NEJ723_A

Experiment SRX19234617: GSM7016923: NEJ723_A; Mus musculus; OTHER

Run SRR23291364 with 17584697480 bases and 149022860 spots

Run SRR23291365 with 17068278622 bases and 144646429 spots

Sample SRS16639059: NEJ654_D

Experiment SRX19234618: GSM7016924: NEJ654_D; Mus musculus; OTHER

Run SRR23291362 with 14384995320 bases and 121906740 spots

Run SRR23291363 with 13982550420 bases and 118496190 spots

Sample SRS16639060: NEJ677_D

Experiment SRX19234619: GSM7016925: NEJ677_D; Mus musculus; OTHER

Run SRR23291360 with 14671616494 bases and 124335733 spots

Run SRR23291361 with 14256433612 bases and 120817234 spots

Sample SRS16639061: NEJ687_C

Experiment SRX19234620: GSM7016926: NEJ687_C; Mus musculus; OTHER

Run SRR23291358 with 16681357094 bases and 141367433 spots

Run SRR23291359 with 16178920764 bases and 137109498 spots

2. In the revision, four tumor samples and three normal controls were used for scRNA-seq. Did the author perform read depth normalization using cellranger aggr function to avoid artifact introduced by sequencing depth?

Response: To correct for the batch effect associated with individual library preparation and processing, we applied corresponding batch correction steps from scLCA and differentiation expression analysis. This correction process has removed possible artifacts including sequencing depth variation, and therefore, we didn't explicitly use cellranger aggr function.

3. The authors performed analysis for both low-quality and high-quality cells considering UMI counts and mitochondrial genes. Only one cluster of cells with > 50% low-quality cells was removed from further analysis. If all low-quality cells are removed at the beginning (which is the way people usually do in scRNA-seq data analysis), do the results look similar?

Response: As this reviewer knows, some types of cells such as lymphocytes tend to have low total UMI and high mitochondrial content. We therefore applied a less biased way to include all types of cells for clustering. In our preliminary analysis (not included in our manuscript), we indeed evaluated the sensitivity of clustering by either removal or inclusion of low-quality cells at the beginning. After comparing the two clustering results with filtered and unfiltered, we obtained the Rand index 0.93 (0 indicating the two clustering do not agree on any pair and 1 indicating that the two clustering are exactly the same.), which demonstrated that our clustering method is robust.

4. Lines 387-391, the observation of tumor cells with high level of erythroid genes is interesting. Is that due to contamination during single-cell library preparation?

Response: As shown in the accompanying table, the specific clusters from normal livers have low fraction of UMIs mapping to hemoglobin genes (except for Cluster 14, a small cluster of erythroid cells). On the contrary, tumor specific clusters have overall elevated fractions of UMIs mapping to hemoglobin genes, especially for the clusters of HB-associated Erythroid that showed 33-83% of UMI from hemoglobin genes per cell. These results indicate that the high expression levels of erythroid genes are unlikely due to contamination. Our results are also consistent with the recent human HB data published recently in Nature Comm by Song et al. We have added this as Supplementary Table 4 and explained in Results by saying:

Source	Cluster	Average percent of HB UMI
Shared	cluster1 (NK/T cells)	2.86
	cluster6 (B cells)	1.86
	cluster8 (Macrophage)	3.26
	cluster10 (Neutrophils)	3.04
	cluster11 (Monocytes)	2.26
Normal	cluster4 (Hepatocytes I)	1.30
	cluster13 (Endothelial)	1.15
	cluster14 (Erythroid)	87.58
	cluster15 (Hepatocytes II)	0.68
Tumor	cluster2 (HB associated Pro-myelocyte)	2.23
	cluster3 (HB associated proliferating Erythroid)	33.65
	cluster7 (Afp ^{weak} Tumor Cluster)	30.73
	cluster9 (HB associated Erythroid II)	83.57
	cluster12 (HB associated Erythroid III)	62.50
	cluster16 (Afp ^{strong} Tumor Cluster)	6.12

“The specific clusters from normal livers have low fraction of UMIs mapping to hemoglobin genes (except for Cluster 14, a small cluster of erythroid cells) (**Supplementary Table 4**). On the contrary, tumor specific clusters have overall elevated fractions of UMIs mapping to hemoglobin genes, especially for the clusters of HB-associated Erythroid that showed 33-

83% of UMI from hemoglobin genes per cell. These results indicate that the high expression levels of erythroid genes are unlikely due to contamination.”

Reviewers' Comments:

Reviewer #4:

Remarks to the Author:

The authors revised this paper promptly according to the reviewers' comments. It is preferable to call this tumor as "HB-like tumor" because CTNNB1 mutations, the common gene aberration in HB, is not seen in this model. However, upregulation of MYC, one of the downstream of Wnt signal pathway, derives HB-like tumor in mouse.

On this standpoint, the authors should clarify the followings:

1. The authors describe this tumor as C2 morphologic phenotype in this paper. However, in other site (lines 118-167), this tumor consists of different types of HB, such as C1, C2, and HCN-NOS. In human HB, these types are considered as different subtypes correlating with patient outcome. If this tumor consists of these different subtypes of HB, as shown in histological data, the authors described the relationship of these different subtypes in this model. C2 aggressive type may be derived from stem/progenitor cells in hepatoblast as described in lines 114-117. The histological findings in Fig. 1 show that this tumor consists of the different subtypes of HB: fetal (\cong C1), embryonal (\cong C2), macrotrabecular, cholangioblastic subtypes or HCN-NOS. The authors should describe the relationship of these subtypes. These subtypes may occur sequentially or randomly.

2. The correlation of these different subtypes should be explained from single-cell sequencing and spatial transcriptome.

3. In serum chemistry, abnormal elevation of AFP was not so high, the levels of AFP in this model were at most twice the normal. In human HB, serum levels of AFP increased more than 1000 times of normal ones. The immunostaining of AFP showed the strong signals in this model. Please describe the reason of this discrepancy. Is it possible that the livers of normal control mice are also positive of AFP?

4. There are minor corrections required

In Abstract: "embryonal hepatoblastoma" might be confused as "embryonal type of hepatoblastoma"

The term of "embryonal" should be deleted.

In Fig. 1 e-h, the scale bars are not seen.

In Line 1733, "f" is changed to #h".

Reviewer #5:

Remarks to the Author:

In this revised manuscript the authors have significantly improved their manuscript. All my questions have been adequately addressed.

Reviewer #6:

Remarks to the Author:

Most of the comments have been addressed.

Response to Reviewer 4

Again, we thank this reviewer for his/her insightful comments and suggestions. We have performed additional analysis to address these comments.

1. The authors describe this tumor as C2 morphologic phenotype in this paper. However, in other site (lines 118-167), this tumor consists of different types of HB, such as C1, C2, and HCN-NOS. In human HB, these types are considered as different subtypes correlating with patient outcome. If this tumor consists of these different subtypes of HB, as shown in histological data, the authors described the relationship of these different subtypes in this model. C2 aggressive type may be derived from stem/progenitor cells in hepatoblast as described in lines 114-117. The histological findings in Fig. 1 show that this tumor consists of the different subtypes of HB: fetal (\cong C1), embryonal (\cong C2), macrotrabecular, cholangioblastic subtypes or HCN-NOS, The authors should describe the relationship of these subtypes. These subtypes may occur sequentially or randomly.

This reviewer could be right about the sequence of tumorigenesis of these tumors. Namely, these subtypes may occur sequentially or randomly. Nevertheless, we only observed well differentiated HB in P67 while embryonal and cholangioblastic subtypes occur at an early time (P7, P25) (Figure 1e), suggesting that there could be a sequential event during MYC-mediated cellular transformation that coopts with liver developmental program. As this reviewer pointed out, C2 aggressive type may be derived from stem/progenitor cells in hepatoblast, and thus appeared at an early developmental stage, while the C1 type may be derived from a more differentiated cells at late developmental stage.

We have added this discussion in page 9, line 167-173.

2. The correlation of these different subtypes should be explained from single-cell sequencing and spatial transcriptome.

We have used the 16-gene signature that differentiate C1 and C2 types to interrogate the subtype heterogeneity from our scRNA-seq and spatial transcriptome (Supplementary Figure 6, and Supplementary Figure 7c, 7d).

Basically, in tumor samples, the expression levels of C2 signature were greatly higher than the C1 gene signature. However, in normal liver samples, the expression of C2 signature is negligible and the C1 signature was dominantly high (Fig. S6a). Next, we examined the expression of C1 and C2 in each cluster of all samples, and found that cluster 16 and cluster 7 expressed high levels of C2 while the hepatocytes (cluster 15) expressed highest levels of C1 (Fig. S6b). We then specifically determined the C1/C2 expression in tumor-specific clusters of each tumor sample. Again, in contrast to C1 signature expression, we found that C2 signature was highly expressed in cluster 16 and cluster 7 in tumor samples (NEJ634, NEJ687 and NEJ723) (Fig. S6c). However, we noticed that tumor sample NEJ709 expressed comparable levels of C1 and C2 signatures (Fig. S6c). Taken together, these data further support the tumor heterogeneity of ABC-Myc tumors.

In our spatial expression analysis, we also found that C1 expression was dominantly high in normal livers. However, the C1 expression in normal livers were not evenly distributed and heterogeneity was observed across the whole section. While C1 expression in tumor samples were greatly lower than in normal livers, spatial heterogeneity was present and samples NEJ687 and NEJ146 expressed higher levels of C1 than NEJ634 and NEJ723 (Fig. S7c). Correspondingly, C2 signature was highly expressed in all tumors present with heterogenous expression across the tumor sections. NEJ634 and NEJ723 expressed higher levels of C2 in comparison with NEJ687 and NEJ146, indicative of intra- and inter-tumoral heterogeneity.

Please note that, while we were successfully obtained scRNA-seq data for sample NEJ709, unfortunately we were failed to obtain high quality data for spatial gene expression.

Supplementary Figure 6. Co-existence of C1 and C2 signatures in ABC-Myc tumors.

a. Bubble plot of expression of C1 and C2 gene signatures in tumor and normal liver samples. **b.** Bubble plot of expression of C1 and C2 gene signatures in each cluster that is tumor-specific, normal liver-specific and shared in both. **c.** Bubble plot of expression of C1 and C2 gene signatures in tumor-specific clusters.

Supplementary Figure 7c,d. Spatial expression of C1 and C2 signatures in ABC-Myc tumors.

c. The spatial expression of C1 gene signature in tumor (top) and normal liver (bottom) samples. **d.** The spatial expression of C2 gene signature in tumor (top) and normal liver (bottom) samples.

In serum chemistry, abnormal elevation of AFP was not so high, the levels of AFP in this model were at most twice the normal. In human HB, serum levels of AFP increased more than 1000 times of normal ones. The immunostaining of AFP showed the strong signals in this model. Please describe the reason of this discrepancy. Is it possibly that the livers of normal control mice are also positive of AFP?

We are also puzzled by this discrepancy. As this reviewer pointed out, one possibility is that normal liver control mice are also AFP positive. Indeed, we do see some low levels of AFP in normal livers from control mice by looking at the spatial gene expression data (Figure 5a), however, this level was still too low in comparison with the tumor samples. Our bulk RNA-seq showed remarkable expression difference of AFP in tumors vs normal livers (log2 fold =9.73, close to 1000-fold). Another possibility is that there are fewer tumor cells undergoing necrosis than the human tumors, therefore AFP release to circulation is modest in this mouse model. We cannot exclude other possibilities such as AFP modifications or AFP amino acid difference between species may affect the ELISA results leading to discrepancy. Due to many possibilities, we therefore decided not to include these speculations.

There are minor corrections required

In Abstract: “embryonal hepatoblastoma” might be confused as “embryonal type of hepatoblastoma” The term of “embryonal” should be deleted.

We have changed it as “embryonal type of hepatoblastoma”.

In Fig. 1 e-h, the scale bars are not seen.

The line was too thin in our original scale bars. Now we have made it clear.

In Line 1733, “f” is changes to #h”.

We have corrected it.